# Vancomycin-resistant enterococci utilise antibiotic-enriched nutrients for intestinal colonisation

Olivia G. King[1], Alexander Y. G. Yip [1], Victoria Horrocks [1], Jesús Miguéns Blanco [2], Julian R. Marchesi [2], Benjamin H. Mullish [2,3], Thomas B. Clarke [4] & Julie A. K. McDonald [1] ✉

Antibiotic treatment significantly disrupts the gut microbiome and promotes vancomycin-resistant enterococci (VRE) intestinal colonisation. These disruptions cause the intestine to act as a reservoir for VRE that seed difficult-to-treat infections. Here we show that antibiotics that promote VRE intestinal colonisation increase the concentration of a wide range of nutrients and decrease the concentration of a wide range of microbial metabolites. We show significant but incomplete suppression of VRE growth by individual short chain fatty acids that were decreased in antibiotic-treated faecal microbiomes. However, mixtures of short chain fatty acids provide complete or near complete suppression of VRE growth. We show that VRE use most nutrients increased in antibiotic-treated faecal microbiomes as carbon or nitrogen sources to support their growth, where *Enterococcus faecium* and *Enterococcus faecalis* have some common and some distinct preferences for the use of these specific nutrients. Finally, we show that *E. faecium* and *E. faecalis* occupy overlapping but distinct nutrient-defined intestinal niches that promote high growth when cultured with each other and when cultured with carbapenem-resistant *Enterobacteriaceae*. Our results demonstrate that VRE occupy distinct intestinal niches in the antibiotic-treated intestine, defined by their abilities to utilise specific enriched nutrients and their abilities to grow with reduced concentrations of inhibitory microbial metabolites.

Vancomycin-resistant enterococci (VRE) are multidrug-resistant pathogens and a major cause of nosocomial infections worldwide[1,2]. These hospital-acquired pathogens pose a major therapeutic challenge as they have intrinsic resistance to several commonly prescribed antibiotics and also have acquired resistance to many available antibiotics[3].

*E. faecium* and *E. faecalis* are responsible for 75% of enterococcal infections and are commonly associated with hospital outbreaks of invasive infections such as bloodstream infections, urinary tract infections, and endocarditis[2,4,5]. *E. faecalis* was more virulent and more prevalent in healthcare-associated infections than *E. faecium*, however, the prevalence of *E. faecium* is increasing due to the rise of vancomycin-resistant and lactam-resistant *E. faecium* strains[6]. *E. faecium* has higher levels of intrinsic and acquired resistance than *E. faecalis*, where 80% of *E. faecium* isolates are vancomycin-resistant compared to 10% of *E. faecalis* isolates[4]. Vancomycin-resistant *E. faecium* has gained attention

[1]Department of Life Sciences, Centre for Bacterial Resistance Biology, Imperial College London, London, UK. [2]Division of Digestive Diseases, Department of Metabolism, Digestion and Reproduction, Faculty of Medicine, St Mary's Hospital Campus, Imperial College London, London, UK. [3]Departments of Gastroenterology and Hepatology, St Mary's Hospital, Imperial College Healthcare NHS Trust, Paddington, London, UK. [4]Department of Infectious Disease, Centre for Bacterial Resistance Biology, Imperial College London, London, UK. ✉e-mail: julie.mcdonald@imperial.ac.uk

as the World Health Organisation classified it as a high priority pathogen due to its unfavourable rankings in several criteria, including mortality, trends of resistance, transmissibility, preventability, treatability, and pipelines for new medicines and diagnostics[7].

The intestine is a major reservoir for VRE that seed difficult-to-treat invasive infections[7,8]. VRE can be acquired from other patients, hospital rooms, or healthcare workers in acute and long-term health-care facilities[9-11]. Patients exposed to broad-spectrum antibiotics (that significantly disrupt the gut microbiome) are highly susceptible to intestinal colonisation and domination by VRE[12-16]. Conversely, healthy individuals have intact colonisation resistance, where their gut microbiomes protect them from VRE intestinal colonisation. To develop targeted and effective therapies to restrict VRE intestinal colonisation we must first understand what drives VRE colonisation in an antibiotic-disrupted gut microbiome.

Currently, we have an incomplete understanding of the mechanisms of colonisation resistance that protect the host against VRE intestinal colonisation. However, a few studies have provided some insights. Previous work demonstrated that antibiotic treatment downregulated the expression of RegIIIγ, a C-type lectin that kills VRE in the small intestine[17]. Another study demonstrated that a gut commensal strain of *Blautia producta* can produce a lantibiotic that can reduce VRE growth[18]. Finally, another study demonstrated that lactulose treatment in patients with severe liver disease increased *Bifidobacterium* growth, increased the production of microbial metabolites, acidified the gut lumen, and reduced the incidence of systemic infections and mortality[19]. However, we lack information on how broad-spectrum antibiotics that promote VRE intestinal colonisation impact nutrient competition and inhibitory metabolite production by the gut microbiome, and how this in turn impacts VRE growth.

Pathogens must have access to nutrients that they can utilise to support their growth and colonise the intestine. In a healthy gut microbiome, commensal microbes compete intensely for nutrients, resulting in nutrient depletion[20]. However, antibiotic-induced killing of gut commensals reduces nutrient competition, making the nutrients available to support pathogen growth. We have a very poor understanding of VRE nutrient utilisation in the antibiotic-treated gut. A previous study demonstrated that vancomycin-treated mice had increased fructose in their caecal contents compared to antibiotic-naïve mice, and vancomycin-resistant *E. faecium* could use fructose as a carbon source[15]. However, this study used a targeted metabolomics approach that likely missed changes in other nutrients that were increased with vancomycin treatment. Moreover, this study only tested one antibiotic, whereas there are multiple broad-spectrum antibiotics used clinically that can promote VRE intestinal colonisation[12-16]. It is important to understand how other broad-spectrum antibiotics that promote VRE intestinal colonisation affect the availability of nutrients that support VRE growth.

Microbiome therapeutics designed to restrict VRE growth based on nutrient utilisation profiles will not be effective unless they deplete all available carbon and nitrogen sources that support VRE growth. Otherwise, VRE can switch to using other available nutrients to grow. We previously demonstrated that broad-spectrum antibiotics that promote the intestinal colonisation of carbapenem-resistant *Enterobacteriaceae* (CRE) caused an increase in the concentration of a wide range of nutrients[21]. As our previous study demonstrated that broad-spectrum antibiotics had a severe impact on the gut microbiota, in this study we hypothesised that broad-spectrum antibiotics that promote VRE intestinal colonisation would also increase the concentration of a wide range of nutrients in the intestine, increasing the availability of many nutrients that could support VRE growth.

It is also not clear how VRE occupy intestinal niches in antibiotic-treated gut microbiomes. A previous study by Caballero and colleagues proposed that vancomycin-resistant *E. faecium* and carbapenem-resistant *Klebsiella pneumoniae* occupied distinct but overlapping intestinal niches, as they showed that growth of each strain was not impaired when they colonised the mouse intestine simultaneously[22]. Shared bacterial niches have also been observed in humans, as patients can be co-colonised with both CRE and VRE simultaneously[23]. However, the mechanism of how VRE and CRE occupied the antibiotic-treated intestine was not shown. Therefore, we hypothesised that VRE and CRE occupy distinct but overlapping nutrient-defined intestinal niches, where VRE share the utilisation of some nutrients with CRE, but also has its own distinct nutrient utilisation abilities.

In the healthy gut microbiome, nutrient utilisation results in the production of microbial metabolites, some of which can inhibit pathogen growth[20]. Therefore, VRE must be able to grow in the presence of microbial metabolites to successfully colonise the intestine. We previously demonstrated that broad-spectrum antibiotics that promote CRE intestinal colonisation decreased the concentration of a wide range of microbial metabolites in the gut microbiome, and several of these metabolites highly inhibited CRE growth[21]. Another study from our group demonstrated that antibiotic treatment promoted *Clostridioides difficile* growth and decreased the concentration of valerate in the gut microbiome, and that valerate inhibited *C. difficile* growth[24]. However, we do not understand how microbial metabolites may impact VRE growth in the intestine. Therefore, we also wanted to determine whether microbial metabolites that are found in healthy gut microbiomes and decreased with antibiotic treatment could suppress VRE growth.

VRE are high-priority pathogens that are included in the ESKAPE group of pathogens, and urgently require new treatments, including new therapies to prevent VRE intestinal colonisation[25]. These therapies would remove a reservoir of VRE, thereby significantly reducing the subsequent development of VRE invasive infections and patient-to-patient VRE transmission[26]. Therefore, this study aimed to measure the nutrient and metabolite landscapes encountered by VRE in a gut microbiome treated with multiple different antibiotics that are known to promote VRE intestinal colonisation. Moreover, we wanted to define the niche that VRE occupied in an antibiotic-treated gut microbiome, how distinct these niches were between *E. faecium* and *E. faecalis*, and how different these niches were compared to niches occupied by other multidrug-resistant pathogens. The results from this study will inform the rational design of a new microbiome therapeutic that can occupy the same niche as VRE to displace them from the intestine.

Here, we show that VRE intestinal colonisation is promoted by antibiotics that increase the availability of a wide range of nutrients and decrease the production of a wide range of microbial metabolites. We demonstrate that individual short chain fatty acids (SCFAs; that are decreased in antibiotic-treated faecal microbiomes) provide significant but incomplete suppression of VRE growth, but mixtures of SCFAs provide complete or near complete suppression of VRE growth. We also show that VRE can use most nutrients that are increased in antibiotic-treated faecal microbiomes as carbon or nitrogen sources to support their growth, where vancomycin-resistant *E. faecium* and *E. faecalis* have some common and some distinct preferences for the use of these nutrients. Finally, we demonstrate that vancomycin-resistant *E. faecium* and *E. faecalis* achieve high growth when co-cultured with each other and when co-cultured with other multidrug-resistant pathogens. Together, these results demonstrate that VRE occupy overlapping but distinct intestinal niches in an antibiotic-treated faecal microbiome, defined by their abilities to utilise specific enriched nutrients and their abilities to grow with reduced concentrations of inhibitory microbial metabolites.

## Results
### Antibiotics altered the nutrient- and metabolite-defined intestinal niches in a manner that supported VRE growth
We performed ex vivo faecal culture experiments to determine how antibiotics that promote VRE intestinal colonisation killed gut

commensals, altered nutrient availability, and altered the presence of microbial metabolites. In these experiments, faeces from healthy human donors were cultured in a gut growth medium that was supplemented with water or an antibiotic known to promote VRE intestinal colonisation: metronidazole (MTZ), clindamycin (CLI), vancomycin (VAN), ceftriaxone (CRO), or piperacillin/tazobactam (TZP), at concentrations mimicking those found in human faeces[27–32]. Samples were analysed by 16S rRNA gene sequencing to measure changes in bacterial taxa and by [1]H-NMR spectroscopy to measure changes in nutrients and metabolites.

Antibiotic-treated faecal culture samples were compared to water-treated faecal culture samples. Principal component analysis plots showed separation between antibiotic-treated and water-treated faecal culture samples along the first principal component, which corresponded to antibiotic treatment (Fig. S1). Faecal cultures treated with antibiotics that promote VRE intestinal colonisation had significant decreases in many bacterial families, as shown in Fig. 1a. *Ruminococcaceae* and *Bifidobacteriaceae* were decreased in all the antibiotic-treated groups, while decreases in other bacterial families varied depending on the antibiotic tested. We found that there was some heterogeneity in the responses of specific bacterial families to antibiotic treatment in faecal cultures inoculated with faeces from different donors (Figs. S2 and S3). Antibiotic-treated faecal cultures also had significant decreases in taxa at the phylum, class, and order levels (Fig. S4).

Antibiotic treatment also significantly altered the abundances of nutrients and metabolites that would be encountered by VRE in an antibiotic treated faecal microbiome. Antibiotic treatment resulted in a significant increase in a wide range of nutrients, including monosaccharides, disaccharides, amino acids, uracil, and succinate (Fig. 1b). Monosaccharides increased with antibiotics included arabinose, fructose, fucose, galactose, glucose, mannose, ribose, xylose, and N-acetylglucosamine. Disaccharides increased with antibiotics included maltose, sucrose, and trehalose. Amino acids increased with antibiotics included alanine, arginine, aspartate, glutamate, glycine, isoleucine, leucine, lysine, methionine, phenylalanine, proline, threonine, tryptophan, tyrosine, and valine. Antibiotic treatment also resulted in a decrease in a wide range of microbial metabolites, including SCFAs (formate, acetate, propionate, butyrate, and valerate), carboxylic acids (lactate and 5-aminovalerate), isobutyrate, and ethanol (Fig. 1b). In contrast, isovalerate increased in CLI-treated and VAN-treated faecal cultures. We found that there was less heterogeneity in the responses of nutrients and metabolites to antibiotic treatment in faecal cultures inoculated with faeces from different donors (Fig. S5).

We performed another set of faecal culture experiments to confirm whether vancomycin-resistant *E. faecium* and *E. faecalis* were able to grow in these antibiotic-treated faecal microbiomes that had increased nutrient concentrations and decreased metabolite concentrations. First, MIC assays confirmed that the *E. faecium* and *E. faecalis* strains were resistant to all the antibiotics tested (Table S1). *E. faecium* and *E. faecalis* were also grown in pure cultures in the presence of the antibiotics at concentrations found in human faeces. Both *E. faecium* and *E. faecalis* were able to grow axenically in faecal concentrations of MTZ, CRO, CLI, and VAN, however, these strains were killed with faecal concentrations of TZP (Fig. S6). Therefore, TZP was not tested further in the faecal culture experiments.

Faecal culture experiments were performed by spiking vancomycin-resistant *E. faecium* or *E. faecalis* into antibiotic-treated faecal cultures as described above. We demonstrated that MTZ, CLI, VAN, and CRO-treated faecal cultures promoted the growth of *E. faecium* and *E. faecalis*, while water-treated faecal cultures restricted *E. faecium* and *E. faecalis* growth (Fig. 1c, d). We also found that there was a significant difference in VRE growth in some of the antibiotic-treated faecal cultures compared to others (Table S2).

In summary, we demonstrated that antibiotic treatment killed gut commensals and promoted VRE growth. This reduction in gut commensal abundance resulted in an increase in the concentration of nutrients (including monosaccharides, disaccharides, amino acids, uracil, and succinate) and a decrease in the concentration of microbial metabolites (including SCFAs, lactate, 5-aminovalerate, isobutyrate, and ethanol).

## Vancomycin treatment caused an increase in nutrients and a decrease in metabolites in mouse faeces and promoted vancomycin-resistant *E. faecium* growth

We used a mouse model to demonstrate that antibiotic treatment altered nutrient availability and metabolite production in vivo. Mice were treated with oral vancomycin as it has been previously demonstrated to promote VRE intestinal colonisation in both humans and mice[12,13,15,16]. Mice were colonised with *E. faecium* to confirm that vancomycin disrupts colonisation resistance and creates an intestinal environment that promoted *E. faecium* growth.

We measured changes in nutrient availability and metabolite production in the faeces of vancomycin-treated mice compared to water-treated mice using [1]H-NMR spectroscopy. Faeces from vancomycin-treated mice had an increase in the concentration of several nutrients compared to water-treated mice, including arabinose, glucose, trehalose, arginine, aspartate, glycine, phenylalanine, and tyrosine (Fig. 2a). Moreover, faeces from vancomycin-treated mice had a decrease in the concentration of several metabolites compared to water-treated mice, including propionate, butyrate, isobutyrate, lactate, and ethanol (Fig. 2b). Fucose and alanine concentrations were decreased in the faeces of vancomycin-treated mice compared to water-treated mice (Fig. S7).

Vancomycin-treated and water-treated mice were gavaged with vancomycin-resistant *E. faecium*, and *E. faecium* growth was quantified in mouse faecal samples (Fig. 2c). Mice treated with vancomycin had high *E. faecium* counts in their faeces, while mice treated with water had low *E. faecium* counts which were at or below the limit of detection.

In summary, the results from our mouse experiment were supportive of the results we observed in our ex vivo human faecal culture experiments, where we showed that VAN-treated faeces were enriched in nutrients (including sugars and amino acids) and depleted in microbial metabolites (including SCFAs, isobutyrate, lactate, and ethanol). We also confirmed that vancomycin-treated mice had disrupted colonisation resistance that promoted high levels of vancomycin-resistant *E. faecium* intestinal colonisation.

## Short chain fatty acids that were decreased in antibiotic-treated human faecal microbiomes significantly suppressed VRE growth

We demonstrated that antibiotics that promote VRE intestinal colonisation reduced the concentration of microbial metabolites in faecal microbiomes (Figs. 1, 2). Previous work demonstrated that microbial metabolites can inhibit the growth of pathogens in the intestine, including CRE and *C. difficile*[21,24,33]. Therefore, we were interested in whether microbial metabolites (that decreased in concentration with antibiotics) could also suppress VRE growth at physiologically relevant concentrations. Previous studies also demonstrated that antibiotic treatment increased the pH in the intestine[33,34]. Moreover, previous studies demonstrated that short chain fatty acids were more inhibitory towards *Enterobacteriaceae* growth at more acidic physiological pH values[21,33]. Therefore, we measured the effects of microbial metabolites at physiological pH values found in the healthy large intestine (pH 6–7)[35,36].

First, we tested whether a mixture of 10 metabolites (containing formate, acetate, propionate, butyrate, valerate, isobutyrate, isovalerate, lactate, 5-aminovalerate, ethanol) were able to suppress *E. faecium* and *E. faecalis* growth. The mixture was tested at three concentrations that represented the average, low, and high concentrations of these metabolites measured in healthy human faeces[21].

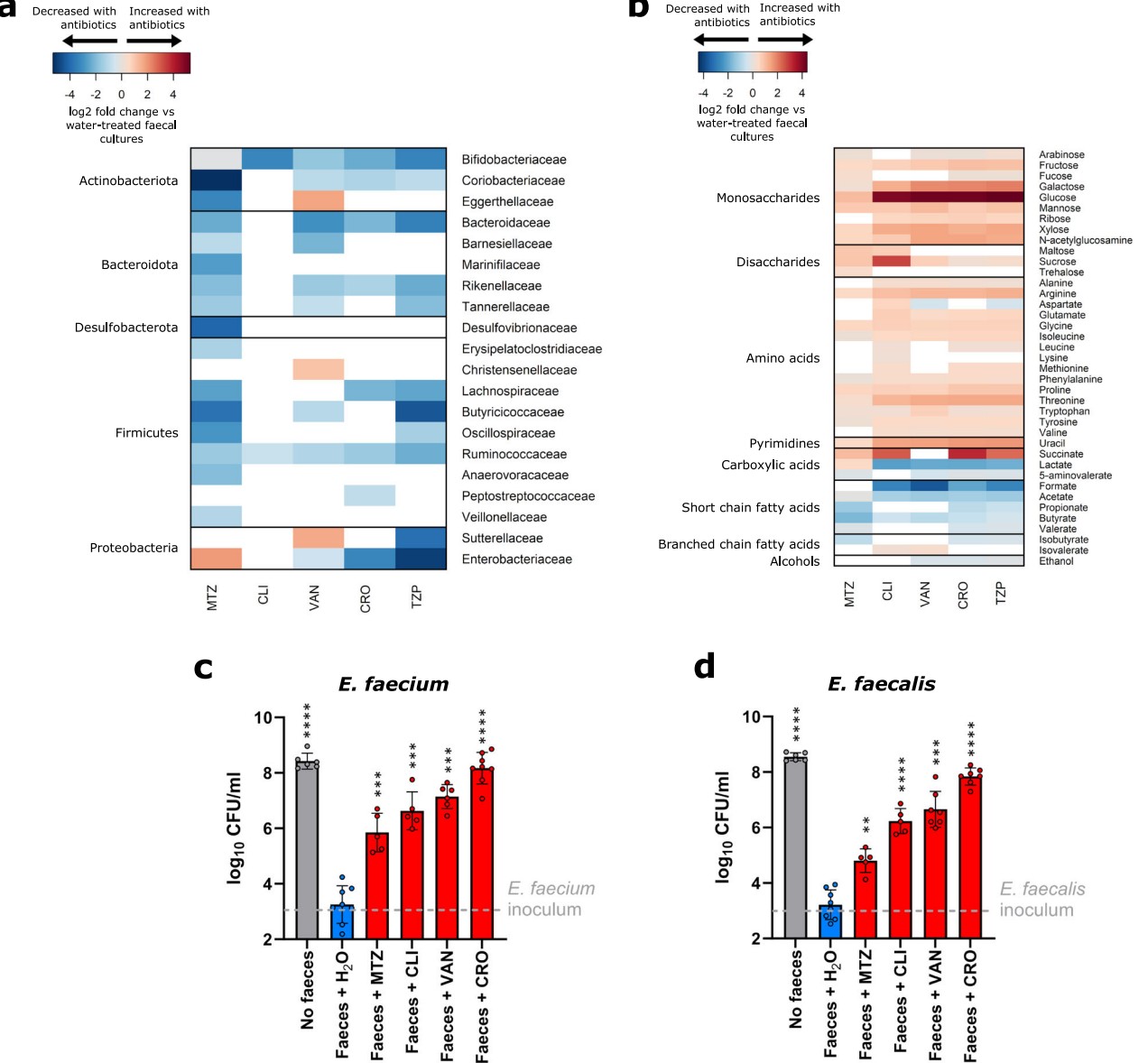

**Fig. 1 | Antibiotic treatment significantly reduced gut commensals, increased nutrients, decreased microbial metabolites, and increased VRE growth.**
**a** Heatmap illustrating the log2 fold change (antibiotic-treated relative to water-treated) in bacterial taxa that were significantly decreased (shown in blue) or increased (shown in red) in faecal cultures following treatment with antibiotics that promote VRE intestinal colonisation. Taxa not significantly changed with antibiotic treatment were not plotted (shown in white). $n = 12$ human faecal donors, Wilcoxon signed rank test (two-sided) of log-transformed abundances with Benjamini-Hochberg false discovery rate (FDR) correction, $p < 0.05$. **b** Heatmap illustrating the log2 fold change (antibiotic-treated relative to water-treated) in nutrients and metabolites that were significantly decreased (shown in blue) or increased (shown in red) in faecal cultures following treatment with antibiotics that promote VRE intestinal colonisation. Nutrients and metabolites not significantly changed with antibiotic treatment were not plotted (shown in white). $n = 12$ human faecal donors, Wilcoxon signed rank test (two-sided) with Benjamini-Hochberg FDR correction,

$p < 0.05$. **c, d** Antibiotic treatment significantly promoted the growth of vancomycin-resistant *E. faecium* (NCTC 12202) and *E. faecalis* (NCTC 12201) in faecal microbiomes. *E. faecium* or *E. faecalis* were spiked into faecal cultures at $10^3$ CFU/ml (horizontal dashed line). Number of human faecal donors used in (**c**): Faeces + H₂O $n = 8$, Faeces + MTZ $n = 5$, Faeces + CLI $n = 5$, Faeces + VAN $n = 6$, Faeces + CRO $n = 8$, No faeces control $n = 6$. Number of human faecal donors used in (**d**): Faeces + H₂O $n = 8$, Faeces + MTZ $n = 5$, Faeces + CLI $n = 5$, Faeces + VAN $n = 7$, Faeces + CRO $n = 7$, No faeces control $n = 6$. Antibiotic-treated faecal culture counts (shown in red) were compared to water-treated faecal culture counts (shown in blue) using a mixed effects model (one-way) of log transformed CFU/ml with Dunnett's multiple comparison. No faeces counts (shown in grey) were compared to water-treated faecal culture counts (shown in blue) using an unpaired t-test (two-sided) of log transformed CFU/ml. Data shown as mean ± SD. ** $= P \leq 0.01$, *** $= P \leq 0.001$, **** $= P \leq 0.0001$. MTZ metronidazole, CLI clindamycin, VAN vancomycin, CRO ceftriaxone, TZP piperacillin/tazobactam.

The 10 metabolite mixture significantly suppressed *E. faecium* and *E. faecalis* growth at the high metabolite concentration at pH 6 (Fig. S8).

Next, the growth of three *E. faecium* and three *E. faecalis* strains were measured in the presence of each individual metabolite. Individual metabolites were tested at concentrations that mimicked the average, low, and high concentrations of these metabolites measured

in healthy human faeces at pH 6[21]. At average concentrations the growth of two *E. faecium* strains and one *E. faecalis* strain were significantly suppressed by acetate, and the growth of one *E. faecium* strain and three *E. faecalis* strains were significantly suppressed by propionate (Fig. 3). At high concentrations the growth of two *E. faecium* strains and two *E. faecalis* strains were significantly suppressed by acetate, the growth of three *E. faecium* strains and three *E. faecalis*

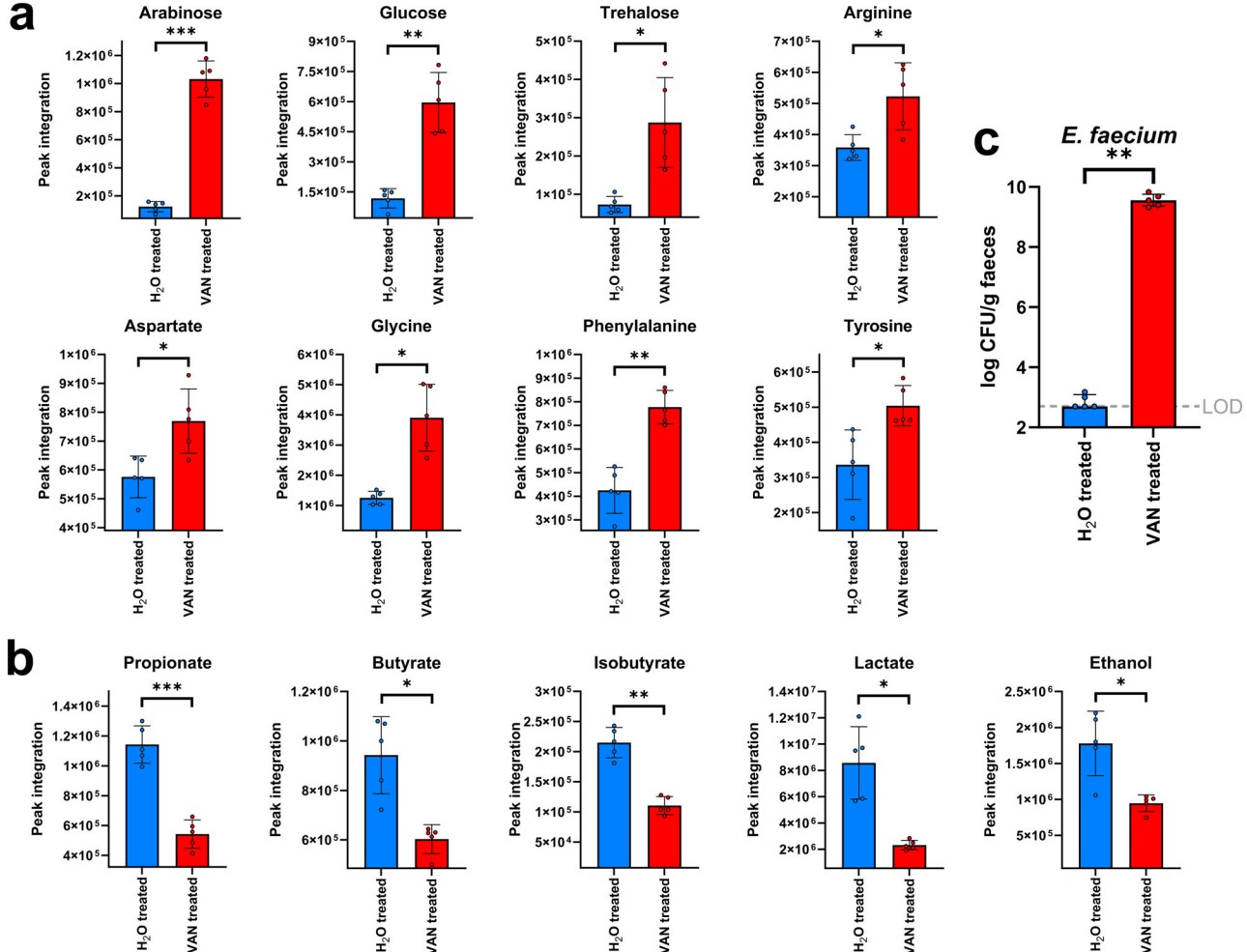

**Fig. 2 | Nutrients were increased and metabolites were decreased in the faeces of vancomycin-treated mice. a** Nutrients that were increased in the faeces of VAN-treated mice (shown in red) compared to $H_2O$-treated mice (shown in blue). **b** Metabolites that were decreased in the faeces of VAN-treated mice (shown in red) compared to $H_2O$-treated mice (shown in blue). **a, b** comparison of nutrients and metabolites in the VAN-treated faeces to the $H_2O$-treated faeces used an unpaired t-test (two-sided) with Benjamini-Hochberg FDR, $n = 5$ mice per group from one independent experiment, with data shown as mean $\pm$ SD. **c** Vancomycin-resistant *E.* *faecium* (NCTC 12202) counts were high in the faeces from vancomycin-treated mice (shown in red) and nearly undetectable in the faeces of water-treated mice (shown in blue). Comparisons of the *E. faecium* plate counts were made using a Mann–Whitney U test (two-sided) of log10-transformed counts. $n = 5$ mice per group from one independent experiment, with data shown as medians $\pm$ IQR. $* = P \le 0.05$, $** = P \le 0.01$, $*** = P \le 0.001$. VAN vancomycin, $H_2O$ water, LOD limit of detection.

strains were significantly suppressed by propionate, the growth of three *E. faecium* strains and two *E. faecalis* strains were significantly suppressed by butyrate, and the growth of three *E. faecium* strains and one *E. faecalis* strain were significantly suppressed by valerate (Fig. 3). Formate, isobutyrate, lactate, 5-aminovalerate, and ethanol did not suppress *E. faecium* or *E. faecalis* growth at these concentrations (Figs. S9–S14). However, the growth of one *E. faecalis* strain was significantly suppressed by isovalerate at the high concentration (Fig. S12).

Next, we compared the effects of each individual metabolite at the same concentrations ranging from 4 to 128 mM, which could be considered for future therapeutic drug designs. For *E. faecium* strains the minimum concentration that caused a significant suppression of growth was 32–64 mM for acetate (1 strain suppressed at concentrations >128 mM), 16–32 mM for propionate, 32 mM for butyrate, and 8–16 mM for valerate (Fig. 3). For *E. faecalis* the minimum concentration that caused a significant suppression of growth was 128 mM for acetate (1 strain suppressed at concentrations >128 mM), 16 mM for propionate, 32–64 mM for butyrate, and 16–32 mM for valerate (Fig. 3). The effects of formate, isobutyrate,

isovalerate, lactate, 5-aminovalerate, and ethanol on *E. faecium* and *E. faecalis* growth at 4–128 mM are shown in Figs. S9–S14. The half maximal inhibitory concentrations ($IC_{50}$) that provide growth suppression against *E. faecium* and *E. faecalis* are shown in Table S3.

We showed that acetate, propionate, butyrate, and valerate were individually capable of suppressing VRE growth at high concentrations found in human faeces. Therefore, next we tested whether a mixture of propionate, butyrate, and valerate (PBV mixture) or a mixture of acetate, propionate, butyrate, and valerate (APBV mixture) were able to more highly suppress VRE growth. At pH 6 we showed that *E. faecium* growth was fully suppressed by the APBV mixture and nearly fully suppressed by the PBV mixture, while *E. faecalis* growth was fully suppressed by both the APBV mixture and PBV mixture (Fig. 4). At pH 6.5 both the APBV and PBV mixtures significantly suppressed *E. faecium* and *E. faecalis* growth (Fig. 4). At pH 7 the APBV mixture significantly suppressed the growth of one *E. faecium* strain and three *E. faecalis* strains, while the PBV mixture significantly suppressed the growth of two *E. faecium* strains and two *E. faecalis* strains (Fig. 4).

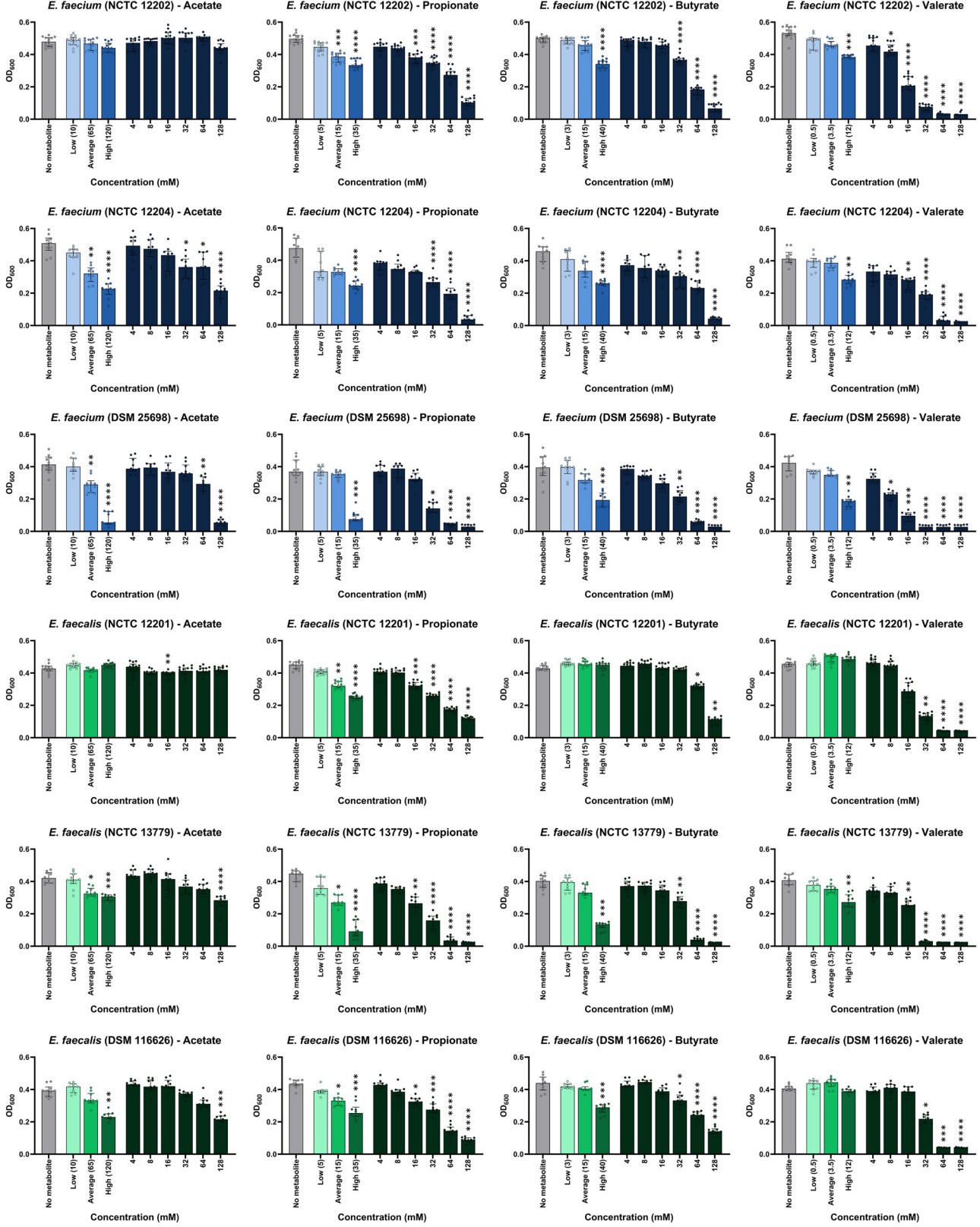

Overall, we observed significant but incomplete suppression of vancomycin-resistant *E. faecium* and *E. faecalis* growth by acetate, propionate, butyrate, and valerate (at high concentrations) when tested individually at pH 6. However, the APBV and PBV mixtures provided complete or near complete suppression of *E. faecium* and *E. faecalis* growth at pH 6.

## Nutrients enriched in antibiotic-treated human faecal microbiomes supported VRE growth

We demonstrated that antibiotics that promote VRE intestinal colonisation increased nutrient availability in faecal microbiomes (Figs. 1, 2). Therefore, next we measured whether the VRE isolates were able to grow in a mixture of nutrients that were enriched with

**Fig. 3 | Individual SCFAs suppressed VRE growth at physiological concentrations.** Vancomycin-resistant *E. faecium* and vancomycin-resistant *E. faecalis* strains were grown in tryptic soy broth (pH 6) supplemented with an individual metabolite at low, average or high concentrations measured in healthy human faeces, concentrations spanning a defined range (4–128 mM), or unsupplemented (no metabolite control). Cultures were incubated under anaerobic conditions overnight. Growth was measured in 14 replicates in two independent experiments for NCTC 12202 and NCTC 12201, and growth was measured in 10 replicates in two independent experiments for NCTC 12204, DSM 25698, NCTC 13779, and DSM 116626. Kruskal-Wallis test with Dunn's multiple comparison test (each metabolite was compared to the no metabolite control). * = $P \le 0.05$, ** = $P \le 0.01$, *** = $P \le 0.001$, **** = $P \le 0.0001$. Data are presented as medians ± IQR. Growth of *E. faecium* or *E. faecalis* without metabolites shown in grey. Growth of *E. faecium* strains with supplemented metabolites shown in shades of blue. Growth of *E. faecalis* strains with supplemented metabolites shown in shades of green. Optical density at 600 nm, OD$_{600}$.

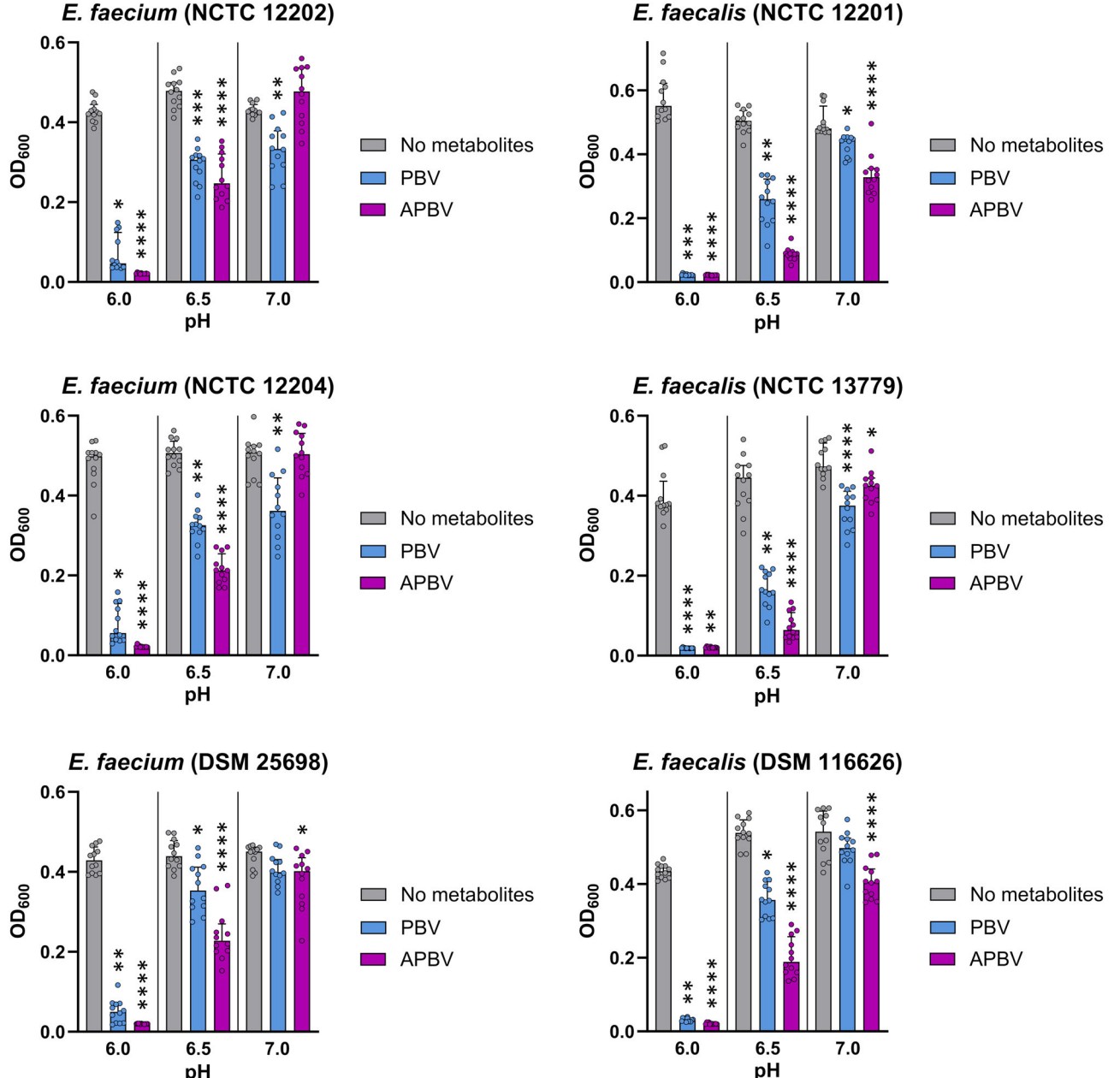

**Fig. 4 | VRE growth was highly suppressed by mixtures of SCFAs at high concentrations.** Vancomycin-resistant *E. faecium* and vancomycin-resistant *E. faecalis* strains were grown in tryptic soy broth supplemented with a mixture of propionate, butyrate, and valerate (PBV) or a mixture of acetate, propionate, butyrate, and valerate (APBV) at concentrations mimicking the high concentrations measured in human faeces, or unsupplemented (No metabolites control). Broth was adjusted to pH 6, 6.5, or 7 to mimic the pH of the healthy large intestine, and cultures were incubated under anaerobic conditions overnight. Data shown as medians ± IQR, with 12 replicates from 3 independent experiments. Kruskal-Wallis with Dunn's multiple comparison test comparing the no metabolites control to each metabolite mixture at the same pH. * $P \le 0.05$, ** = $P \le 0.01$, *** = $P \le 0.001$, **** = $P \le 0.0001$. OD$_{600}$ optical density at 600 nm.

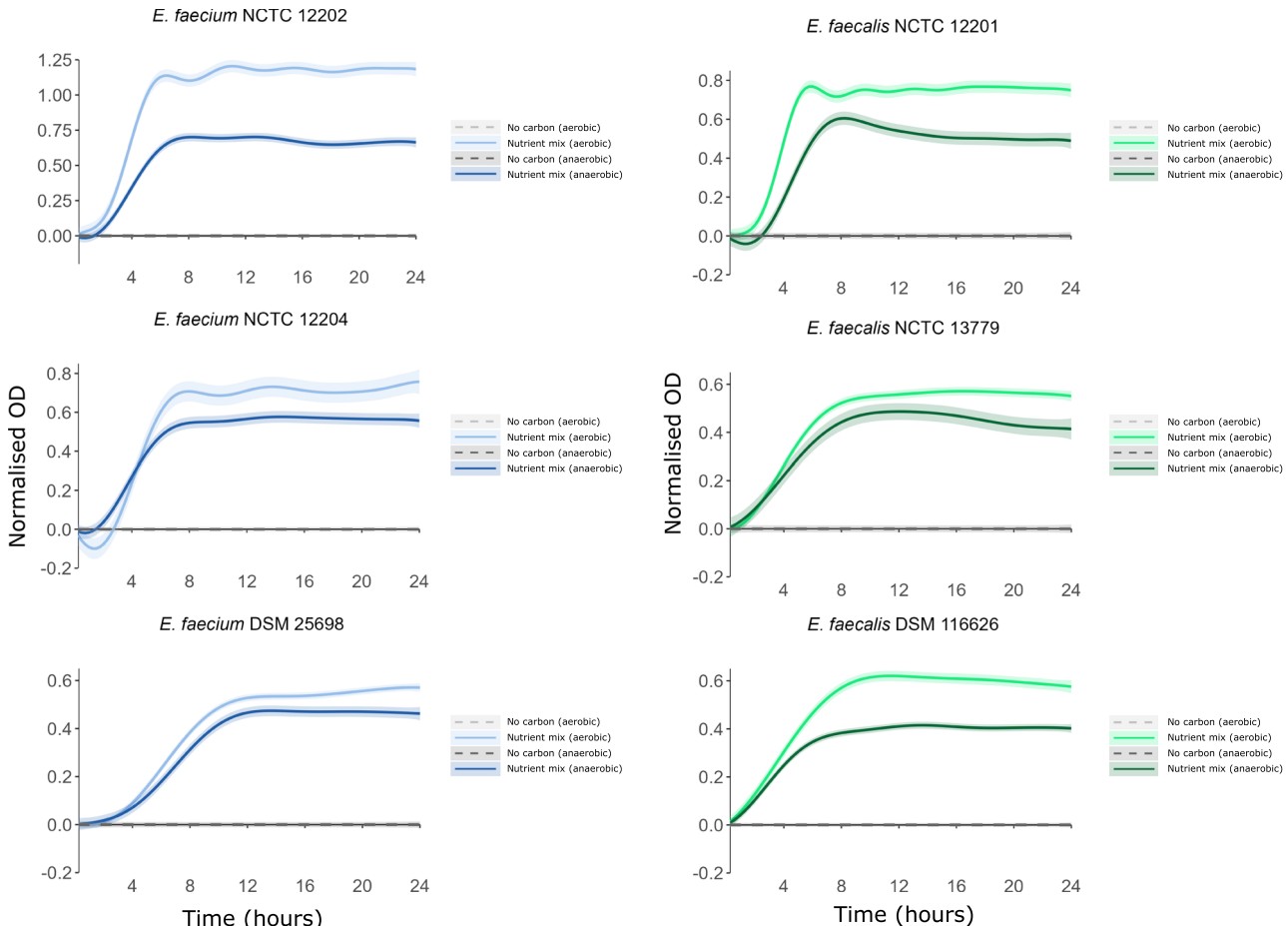

**Fig. 5 | VRE used nutrients enriched in antibiotic-treated faecal microbiome as sole carbon and nitrogen sources.** AMiGA-predicted growth curves for vancomycin-resistant *E. faecium* and vancomycin-resistant *E. faecalis* strains cultured in minimal medium supplemented with a mixture of nutrients under anaerobic or aerobic conditions. The bold lines show the predicted mean of growth while the shaded bands show the predicted 95% credible intervals. *n* = 6 replicates in two independent experiments. OD optical density.

antibiotic treatment. Growth was measured in the presence and absence of oxygen as previous studies demonstrated that antibiotic treatment increased oxygen availability in the intestine, which could impact nutrient utilisation by pathogens[21,37]. Therefore, we wanted to determine whether the presence of oxygen also impacted nutrient utilisation by VRE.

Vancomycin-resistant *E. faecium* and *E. faecalis* strains were cultured in a minimal growth medium supplemented with a mixture of nutrients that were increased in antibiotic-treated faecal microbiomes. VRE growth was supported by this mixture of nutrients under both anaerobic and aerobic conditions (Fig. 5). Growth was higher for *E. faecium* and *E. faecalis* under aerobic conditions compared to anaerobic conditions.

Next, we measured whether a healthy faecal microbiota was able to deplete nutrients (that we demonstrated were increased in antibiotic-treated faecal microbiomes) to restrict VRE growth. We demonstrated that VRE growth was suppressed in spent faecal supernatants and spent VRE supernatants supplemented with water, compared to growth in a minimal medium supplemented with the nutrient mixture (Fig. 6). We also demonstrated that glucose supplementation of spent faecal supernatants and spent VRE supernatants caused an increase in growth that was comparable to growth achieved in the nutrient-supplemented minimal medium. These results support the hypothesis that nutrient depletion contributes to the restriction of VRE growth in a healthy gut microbiome.

## Many nutrients enriched in antibiotic-treated faecal microbiomes acted as sole carbon or nitrogen sources to support VRE growth

We measured VRE growth on individual carbon and nitrogen sources that were enriched in antibiotic-treated faecal microbiomes to determine which specific nutrients were capable of supporting VRE growth. This information could help inform the design of new microbiome therapeutics composed of gut commensals that can outcompete the VRE strains for use of these specific nutrients. We also wanted to determine whether vancomycin-resistant *E. faecium* and *E. faecalis* used different nutrients, as this may indicate whether these strains could share the same intestinal niches.

First, the growth of three vancomycin-resistant *E. faecium* strains and three vancomycin-resistant *E. faecalis* strains were measured in minimal medium supplemented with each individual nutrient as a sole carbon source, under both anaerobic and aerobic conditions.

Under anaerobic conditions *E. faecium* and *E. faecalis* growth was supported by monosaccharides and disaccharides. High or moderate growth was achieved for three *E. faecium* strains on arabinose, fructose, galactose, glucose, mannose, ribose, N-acetylglucosamine, maltose, and trehalose, and high growth was achieved for two *E. faecium* strains on sucrose (Fig. 7, S15). High or moderate growth was achieved for three *E. faecalis* strains on fructose, glucose, mannose, ribose, N-acetylglucosamine, maltose, and trehalose, and high or moderate growth was achieved for two *E. faecalis* strains on galactose and

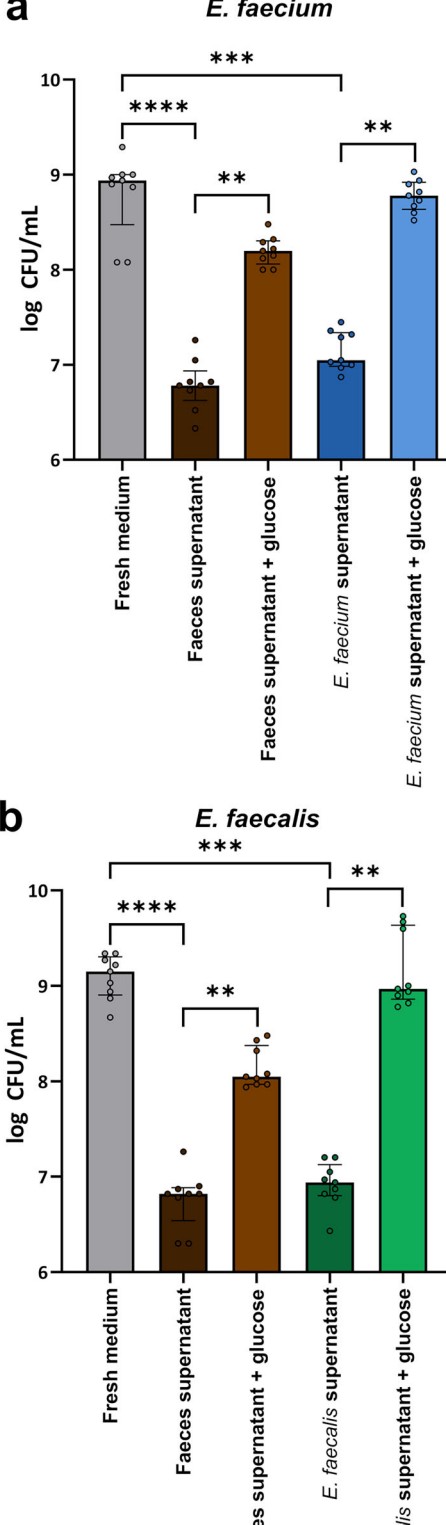

**a** *E. faecium*

**b** *E. faecalis*

**Fig. 6 | Faecal culture supernatants suppressed VRE growth through nutrient depletion.** Minimal medium was supplemented with a mixture of nutrients that were increased with antibiotics and inoculated with either fresh healthy human faeces ($n = 3$ donors for 9 replicates from 3 independent experiments, shown in brown), a single VRE strain (*E. faecium* NCTC 12202 shown in blue or *E. faecalis* NCTC 12201 shown in green; $n = 9$ replicates from 3 independent experiments), or left uninoculated (shown in grey, $n = 9$ replicates from 3 independent experiments). Spent supernatants were filter sterilised and then inoculated with (**a**) *E. faecium* NCTC 12202 or (**b**) *E. faecalis* NCTC 12201. As a control, spent supernatants were also supplemented with 0.5% glucose to confirm nutrient depletion. VRE growth was measured after overnight incubation using plate counts. **a**, **b** comparisons between growth in the spent supernatants and fresh medium were assessed using a Kruskal-Wallis with Dunn's multiple comparison test, and comparisons of unsupplemented vs glucose-supplemented supernatants was assessed through a Wilcoxon signed rank test (two-sided). ** = $P \leq 0.01$, *** = $P \leq 0.001$, **** = $P \leq 0.0001$. Data are presented as medians ± IQR.

was achieved for two *E. faecium* strains on sucrose (Figs. 7, S15). High or moderate growth was achieved for three *E. faecalis* strains on fructose, galactose, glucose, mannose, ribose, N-acetylglucosamine, maltose, and trehalose, and high or moderate growth was achieved for two *E. faecalis* strains on sucrose (Figs. 7, S16). Low growth or negligible growth was achieved on the other carbon sources tested (Figs. S15–S18).

Most amino acids were not used as carbon sources; however, they could act as nitrogen sources to support VRE growth. Therefore, next we tested the ability of nutrients that were increased in antibiotic-treated faecal microbiota to act as nitrogen sources to support the growth of three vancomycin-resistant *E. faecium* strains and three vancomycin-resistant *E. faecalis* strains under anaerobic and aerobic conditions. As *E. faecium* and *E. faecalis* had undefined amino acid auxotrophies, nitrogen utilisation was tested in a leave-one-out assay where growth in a mixture containing all nitrogen sources was compared to growth in a mixture lacking one of the nitrogen sources.

Under anaerobic conditions *E. faecium* and *E. faecalis* growth was supported by amino acids. There was a large or moderate decrease in growth for three *E. faecium* strains when arginine, glutamate, glycine, histidine, isoleucine, leucine, lysine, methionine, serine, threonine, tryptophan, or valine were excluded, a large or moderate decrease in growth for two *E. faecium* strains when phenylalanine was excluded, and a moderate decrease in growth for one *E. faecium* strain when cysteine or tyrosine were excluded (Figs. 8, S19). There was a large or moderate decrease in growth for three *E. faecalis* strains when arginine, glutamate, histidine, isoleucine, leucine, methionine, serine, tryptophan, or valine were excluded, a large or moderate decrease in growth for two *E. faecalis* strains when glycine or phenylalanine were excluded, and a moderate decrease in growth for one *E. faecalis* strain when asparagine or lysine were excluded (Figs. 8, S20). There was a small decrease in growth, negligible decrease in growth, or an increase in growth when the other nitrogen sources were excluded (Figs. 8, S19, S20).

Under aerobic conditions *E. faecium* and *E. faecalis* growth was also supported by amino acids, as well as uracil and N-acetylglucosamine. There was a large decrease in growth for three *E. faecium* strains when arginine, glutamate, glycine, isoleucine, leucine, lysine, methionine, phenylalanine, serine, threonine, tryptophan, or valine were excluded, a large or moderate decrease in growth for two *E. faecium* strains when aspartate, histidine, or tyrosine were excluded, and a moderate decrease in growth for one *E. faecium* strain when cysteine or N-acetylglucosamine were excluded (Figs. 8, S19). There was a large or moderate decrease in growth for three *E. faecalis* strains when arginine, glutamate, histidine, isoleucine, leucine, methionine, serine, tryptophan, or valine were excluded, a large or moderate decrease in growth for two *E. faecalis* strains when lysine, phenylalanine, or uracil were excluded, and a large or moderate decrease in growth for one *E. faecalis* strain when alanine, glycine, threonine, tyrosine, or N-acetylglucosamine were

sucrose (Figs. 7, S16). Low growth or negligible growth was achieved on the other carbon sources tested (Figs. S15–S18).

Under aerobic conditions *E. faecium* and *E. faecalis* growth was also supported by monosaccharides and disaccharides. High or moderate growth was achieved for three *E. faecium* strains on arabinose, fructose, galactose, glucose, mannose, ribose, N-acetylglucosamine, maltose, and trehalose, and high or moderate growth

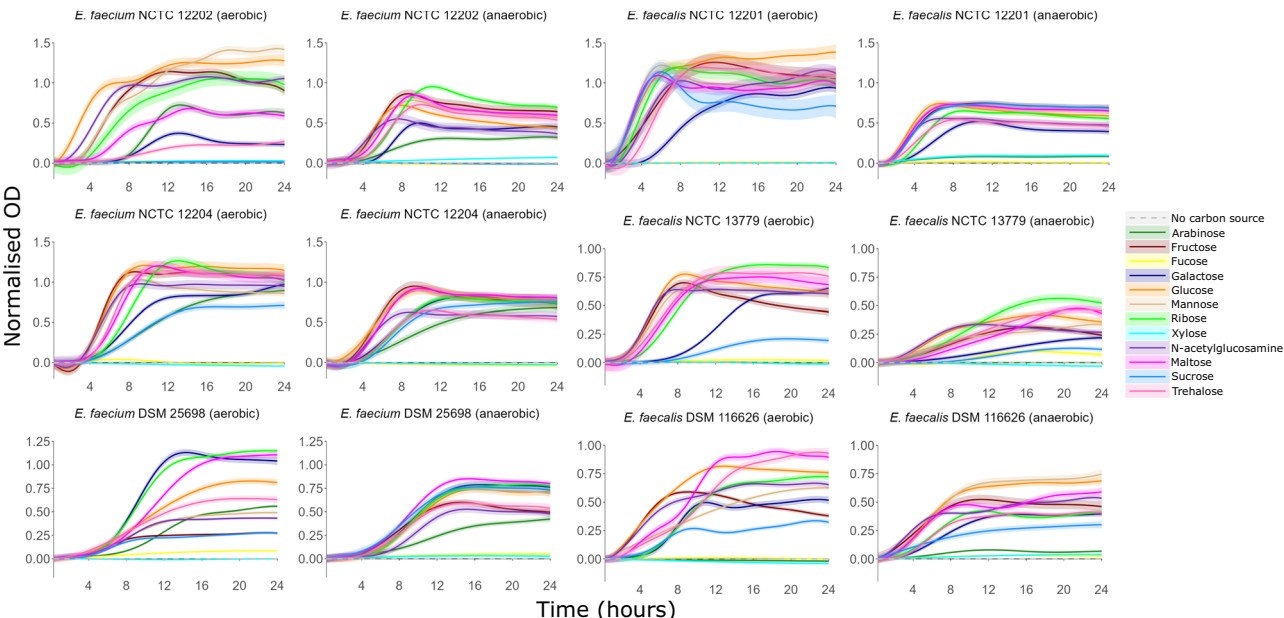

**Fig. 7 | VRE growth was supported by individual monosaccharides and disaccharides as carbon sources.** AMiGA-predicted growth curves for vancomycin-resistant *E. faecium* and vancomycin-resistant *E. faecalis* strains cultured in minimal medium supplemented with single carbon sources under anaerobic or aerobic conditions. The bold lines show the predicted mean of growth while the shaded bands show the predicted 95% credible intervals. *n* = 6 replicates in 2–3 independent experiments. OD optical density.

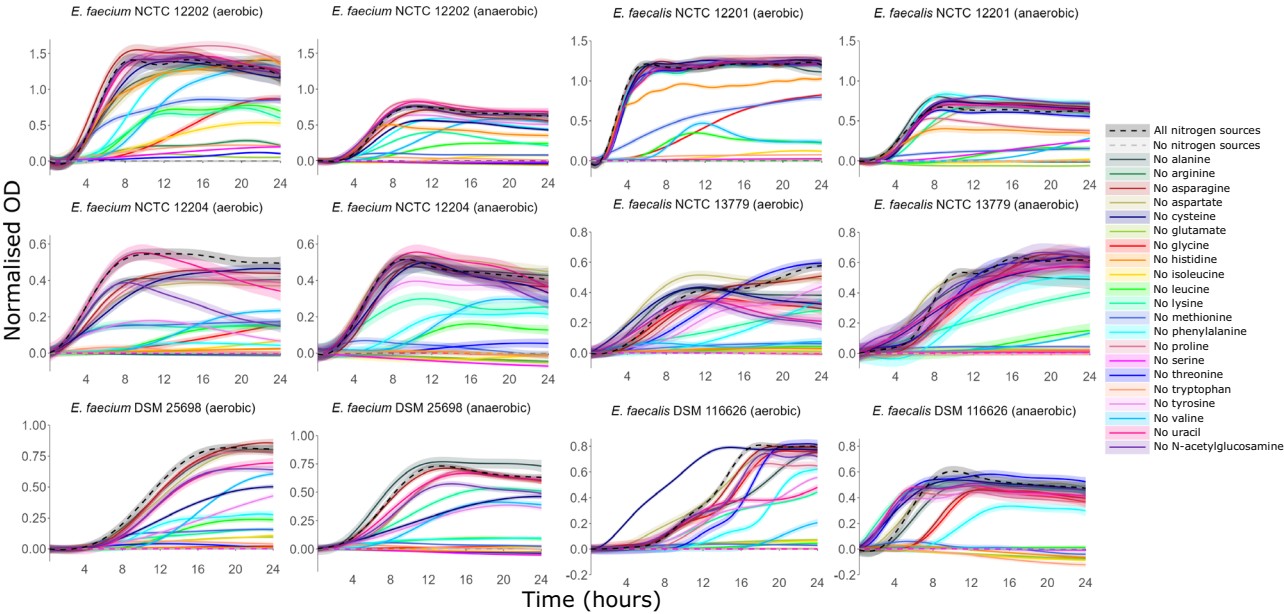

**Fig. 8 | Amino acids served as nitrogen sources to support VRE growth.** AMiGA-predicted growth curves for vancomycin-resistant *E. faecium* and vancomycin-resistant *E. faecalis* strains cultured in minimal medium supplemented with a mixture of all nitrogen sources (all nitrogen sources), a leave-one-out mixture lacking a single nitrogen source, or lacking all nitrogen sources (no nitrogen sources) under anaerobic or aerobic conditions. The bold lines show the predicted mean of growth while the shaded bands show the predicted 95% credible intervals. *n* = 6 replicates in 2–3 independent experiments. OD optical density.

excluded (Figs. 8, S20). There was a small decrease in growth, negligible decrease in growth, or an increase in growth when the other nitrogen sources were excluded (Figs. 8, S19, S20).

Overall, vancomycin-resistant *E. faecium* and vancomycin-resistant *E. faecalis* strains grew on many nutrients that were increased in antibiotic-treated faecal microbiomes. *E. faecium* and *E. faecalis* used monosaccharides and disaccharides as carbon sources and used amino acids as nitrogen sources. However, there were some differences in the utilisation of specific nutrients between *E. faecium*

and *E. faecalis*, and some differences between different strains of the same species.

**VRE preferred to use specific nutrients over others in a mixture of nutrients that were elevated in antibiotic-treated faecal microbiomes**

Previous work demonstrated that pathogens (such as *Escherichia coli*, *Klebsiella pneumoniae*, and *Enterobacter hormaechei*) have a preference for some nutrients over others when presented with a mixture

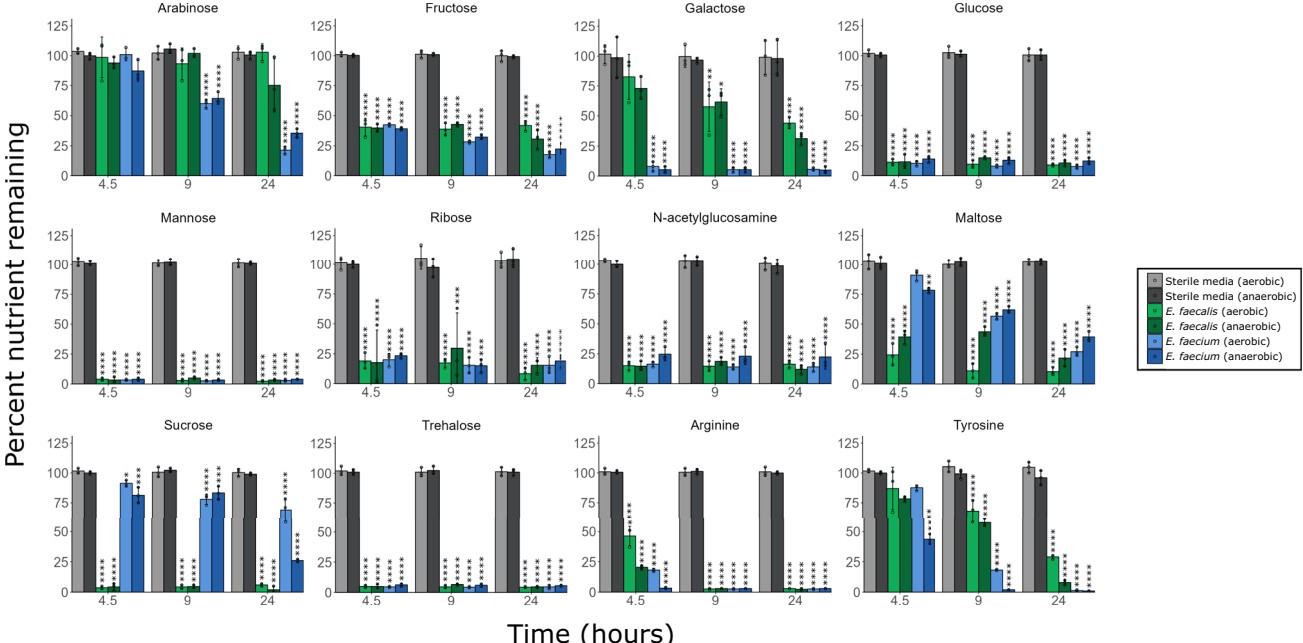

**Fig. 9 | VRE preferred specific nutrients when grown in nutrient mixtures found in antibiotic-treated faecal microbiomes.** Percentage of nutrients remaining in vancomycin-resistant *E. faecium* (NCTC 12202) and *E. faecalis* (NCTC 12201) cultures grown in a minimal medium supplemented with a mixture of nutrients, under aerobic or anaerobic conditions. ¹H-NMR spectroscopy peak integrations were used to measure nutrient concentrations. Comparisons were made using a two-way mixed ANOVA followed by pairwise comparisons with Bonferroni correction. $n = 3$ replicates per group, with data shown as mean ± SD. $* = P \le 0.05$, $** = P \le 0.01$, $*** = P \le 0.001$, $**** = P \le 0.0001$.

of nutrients that they can use to support their growth[21,38]. Moreover, the presence of some nutrients can influence the utilisation of other nutrients[39,40]. This hierarchy of nutrient preferences may have implications for the designs of microbiome therapeutics that are based on nutrient competition. However, nutrient preferences have not been studied in vancomycin-resistant *E. faecium* and *E. faecalis*. Therefore, next we cultured a vancomycin-resistant *E. faecium* strain and a vancomycin-resistant *E. faecalis* strain in a minimal medium supplemented with a mixture of nutrients that were increased in an antibiotic-treated intestine (as sole carbon and nitrogen sources) under both anaerobic and aerobic conditions and measured utilisation of these nutrients over time by ¹H-NMR spectroscopy.

The *E. faecium* and *E. faecalis* strains had some common and some distinct nutrient preferences, as demonstrated by how quickly they utilised each nutrient in the nutrient mixture (Fig. 9). Both *E. faecium* and *E. faecalis* highly utilised glucose, mannose, trehalose, ribose, N-acetylglucosamine, fructose, and arginine by 4.5 h. However, *E. faecium* highly utilised galactose by 4.5 h, while *E. faecalis* highly utilised galactose by 24 h. Conversely, *E. faecalis* highly utilised maltose and sucrose by 4.5 h, while *E. faecium* highly/moderately utilised maltose and sucrose by 24 h. *E. faecium* highly used arabinose by 24 h, while *E. faecalis* did not use arabinose. Finally, *E. faecium* highly used tyrosine by 9 h, while *E. faecalis* highly utilised tyrosine by 24 h. *E. faecalis* also had low utilisation of some amino acids under anaerobic conditions by 24 h (Fig. S21).

In summary, both *E. faecium* and *E. faecalis* strains showed a preference for some nutrients over others when cultured with a mixture of nutrients that were elevated in antibiotic-treated faecal microbiomes. Moreover, nutrient utilisation was generally comparable for these strains when cultured under both anaerobic and aerobic conditions.

## VRE occupied overlapping but distinct intestinal niches with each other and with carbapenem-resistant *Enterobacteriaceae*

Our data demonstrated that *E. faecium* and *E. faecalis* strains consumed a similar, but not identical range of nutrients found in antibiotic-treated faecal microbiomes. Moreover, our data demonstrated that *E. faecium* and *E. faecalis* strains had similar but not identical preferences for these nutrients. Therefore, we wanted to determine whether *E. faecium* and *E. faecalis* occupied distinct nutrient-defined intestinal niches. If these strains occupy distinct intestinal niches, this may mean that different approaches are required to design microbiome-based therapeutics to displace them from the intestine.

To test this, *E. faecium* and *E. faecalis* strains were co-cultured in a minimal medium supplemented with a mixture of nutrients that were elevated in antibiotic-treated faecal microbiota. We demonstrated that both *E. faecium* and *E. faecalis* achieved high growth when co-cultured together (Fig. 10a). However, for *E. faecium* there was a small but significant suppression of growth in the co-culture compared to growth in monoculture. These results indicate that there is partial, but not complete overlap of the nutrient-defined niches, and both strains can simultaneously grow together on nutrients that are enriched in antibiotic-treated faecal microbiomes.

Both VRE and CRE intestinal colonisation are promoted by broad spectrum antibiotics, which we demonstrated enriched for a similar range of nutrients. Therefore, we aimed to determine whether VRE and CRE occupied distinct nutrient-defined intestinal niches which permitted for simultaneous growth of VRE and CRE. Again, if VRE and CRE occupy distinct intestinal niches this may have implications for the design of new microbiome-based therapeutics to decolonise these pathogens.

Co-culture experiments were carried out as described above, except that in this experiment one VRE strain was co-cultured with one CRE strain. Although *E. faecium* and *E. faecalis* growth were slightly suppressed in the co-culture with each CRE strain, we showed that both VRE and CRE strains achieved high growth when co-cultured together (Fig. 10b). These results suggested that there is partial, but not complete overlap of the nutrient-defined niches between VRE and CRE, allowing both types of pathogens to simultaneously grow together to high levels on nutrients that are elevated in antibiotic-treated faecal microbiomes.

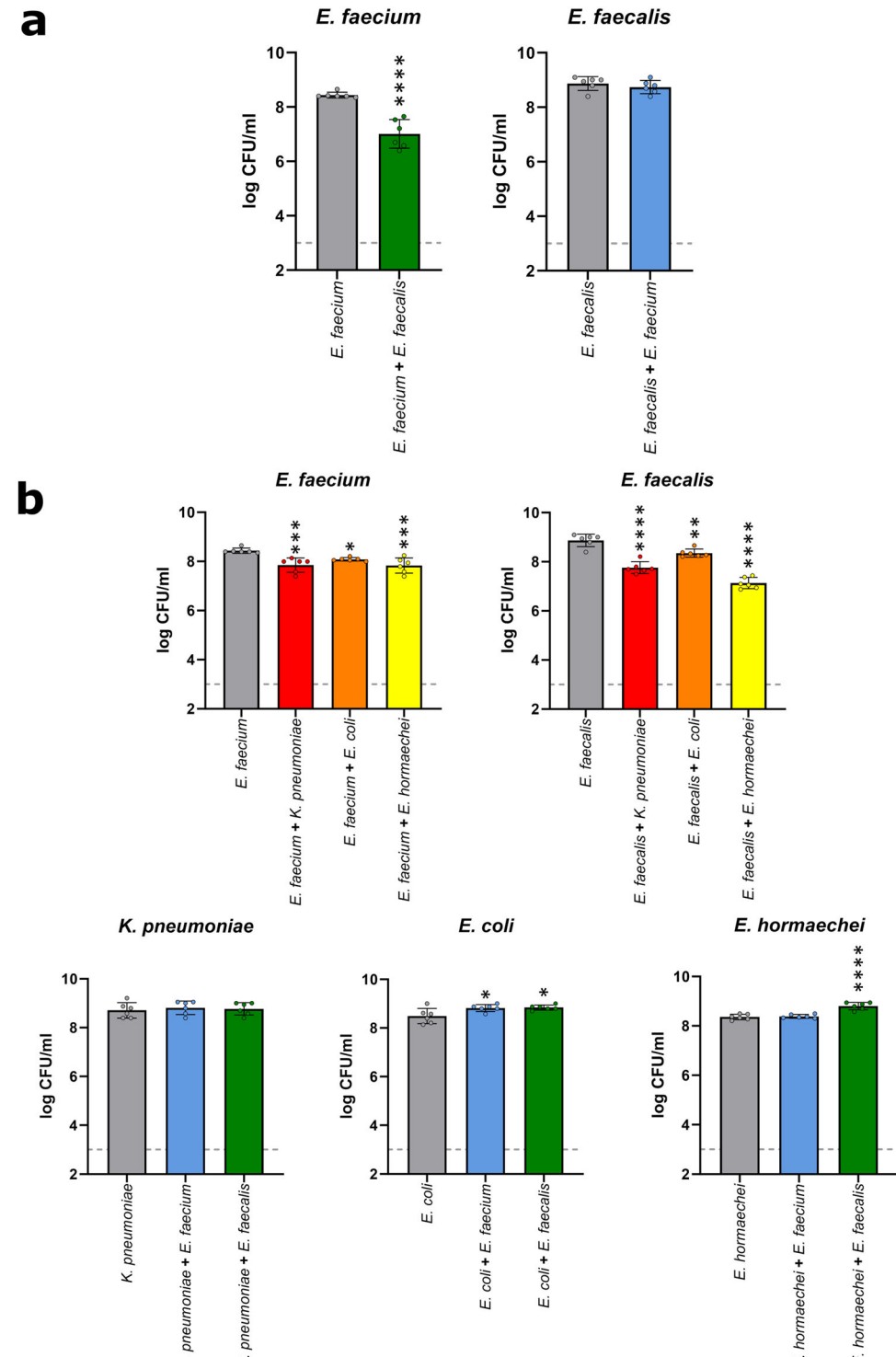

**Fig. 10 | VRE and CRE achieved high growth in co-culture, but VRE growth was slightly suppressed. a** *E. faecium* (NCTC 12202) and *E. faecalis* (NCTC 12201) were co-cultured in a minimal medium supplemented with a mixture of nutrients that were enriched in antibiotic-treated faecal microbiomes. Monoculture growth shown in grey. *E. faecium* growth when co-cultured with *E. faecalis* shown in green. *E. faecalis* growth when co-cultured with *E. faecium* shown in blue. **b** *E. faecium* (NCTC 12202) or *E. faecalis* (NCTC 12201) were co-cultured with *K. pneumoniae*, *E. coli*, or *E. hormaechei* in a minimal medium supplemented with a mixture of nutrients that were enriched in antibiotic-treated faecal microbiomes. Monoculture growth shown in grey. *E. faecium* or *E. faecalis* growth when co-cultured with *K. pneumoniae* shown in red, with *E. coli* shown in orange, and with *E. hormaechei* shown in yellow. *K. pneumoniae*, *E. coli*, or *E. hormaechei* growth when co-cultured

with *E. faecium* shown in blue, and with *E. faecalis* shown in green. For panels a and b, each strain was inoculated into the minimal media at a concentration of $10^3$ CFU/ml (represented as the grey dashed line) and the cultures were incubated for 24 h under anaerobic conditions. Samples were plated on Brilliance VRE plates to quantify *E. faecium* or *E. faecalis* growth and on Brilliance CRE plates to quantify *K. pneumoniae*, *E. coli*, or *E. hormaechei* growth. Growth of each isolate was measured with 6 replicates in two independent experiments. Comparison of *E. faecium* and *E. faecalis* monoculture growth to co-culture growth was made using an unpaired t-test (two-sided). Comparison of VRE strain and CRE strain monoculture growth to co-culture growth was made using a one-way ANOVA with Dunnett's multiple comparison test. * = $P \le 0.05$, ** = $P \le 0.01$, *** = $P \le 0.001$, **** = $P \le 0.0001$. Data are presented as mean values ± SD.

## Discussion

We used both ex vivo and in vivo models to demonstrate that five antibiotics that promoted VRE intestinal colonisation disrupted colonisation resistance, leading to increased availability of a wide range of nutrients and decreased production of a wide range of microbial metabolites. We found significant but incomplete suppression of VRE growth by individual SCFAs (that were decreased in antibiotic-treated faecal microbiomes). However, we found that mixtures of SCFAs provided complete or near complete suppression of VRE growth. We also demonstrated that VRE could use most monosaccharides and disaccharides that were increased in antibiotic-treated gut microbiomes as carbon sources, and most amino acids that were enriched in antibiotic-treated gut microbiomes as nitrogen sources. When cultured with a mixture of nutrients that were found in antibiotic-treated faecal microbiomes, *E. faecium* and *E. faecalis* strains preferentially utilised specific nutrients over others, where the nutrient preferences for each strain were the same for some nutrients and different for others. Finally, we also showed that vancomycin-resistant *E. faecium* and *E. faecalis* occupied overlapping but distinct intestinal niches in an antibiotic-treated gut microbiome that permitted high growth with each other and with carbapenem-resistant *Enterobacteriaceae*.

Antibiotics are a major risk factor for VRE intestinal colonisation, including vancomycin, third generation cephalosporins, and antibiotics with potent activity against anaerobic bacteria[41,42]. In this study we measured the effects of several antibiotics (VAN, CRO, CLI, MTZ, and TZP) on the human faecal microbiome, where these antibiotics are frequently used clinically and have been shown to promote VRE intestinal colonisation in humans and mice. In agreement with previous studies[12–16], we demonstrated that VAN, CRO, CLI, and MTZ promoted the growth of vancomycin-resistant *E. faecium* and *E. faecalis*.

Previous studies demonstrated that prior exposure to TZP was associated with VRE intestinal colonisation[13,14]. However, TZP has anti-enterococcal activity and is secreted in human bile at high concentrations[14]. In this study, we demonstrated that *E. faecium* and *E. faecalis* were killed in the presence of TZP at concentrations found in human faeces. However, TZP may promote VRE intestinal colonisation in vivo if the patient is exposed to VRE shortly after finishing a course of TZP (when TZP concentrations have decreased, but before the gut microbiota has recovered). A previous study demonstrated that mice exposed to VRE following the cessation of TZP treatment had high levels of VRE intestinal colonisation[14]. In this study we demonstrated that TZP-treated human faecal cultures had a similar change in nutrients and metabolites as CRO-treated faecal cultures. We concluded that antibiotics could promote pathogen intestinal colonisation by disrupting the gut microbiome, and colonisation could occur during a course of antibiotics if VRE is resistant to the antibiotic, or shortly after stopping a course of antibiotics (prior to recovery of the gut microbiome) if VRE was susceptible to the antibiotic. This emphasises the need to study VRE growth promotion in an antibiotic-treated faecal microbiota over the course of antibiotic treatment and following cessation of antibiotics.

We demonstrated that antibiotics that promote VRE intestinal colonisation caused a significant decrease in the abundance of a wide range of bacterial families, including *Ruminococcaceae* and *Bifidobacteriaceae* (which were significantly decreased with all antibiotics tested). Moreover, we saw some heterogeneity in the responses of specific bacterial families to antibiotic treatment in faecal cultures seeded with faeces from different donors. This heterogeneity may be due to differences in the presence of different antibiotic-resistance genes in bacterial species in the gut microbiotas from different donors. Finally, because certain bacterial families were found in some donors but not others, changes in these families did not reach statistical significance, even though they appeared to change in faecal cultures from donors that possessed these bacterial families.

It is difficult to understand the impact of antibiotic treatment on the abundance of specific gut commensals without first understanding the functions that these bacteria perform in the gut microbiome. Therefore, we measured changes in nutrients and metabolites directly to define the nutrient and metabolite landscape that VRE encounter during antibiotic treatment. We found that antibiotic treatment enriched for a wide range of nutrients and depleted a wide range of microbial metabolites, and that changes in nutrients and metabolites were less heterogeneous between different donors than changes in bacterial families. This may be due to shared functional redundancy between gut commensals, where different bacterial taxa possess genes that perform similar functions[43].

We showed that the antibiotics which promoted VRE intestinal colonisation killed gut commensals that produced a wide range of microbial metabolites. These metabolites had the potential to restrict pathogen growth, and therefore we tested these metabolites in growth assays both individually and as mixtures. When tested individually we showed that acetate, propionate, butyrate, and valerate significantly suppressed *E. faecium* and *E. faecalis* growth at high concentrations found in healthy human faeces at pH 6 (acetate and propionate also significantly suppressed the growth of some *E. faecium* and *E. faecalis* strains at average concentrations). However, there were some differences in the susceptibility of VRE strains to each metabolite. At high concentrations the growth of all *E. faecium* strains were suppressed by propionate, butyrate, and valerate, while the growth of two *E. faecium* strains were suppressed by acetate. At high concentrations the growth of all *E. faecalis* strains were suppressed by propionate, the growth of two *E. faecalis* strains were suppressed by acetate and butyrate, and the growth of one *E. faecalis* strain was suppressed by valerate. In a previous study we demonstrated that the growth of CRE strains were also suppressed by acetate, propionate, butyrate, and valerate at average and high concentrations measured in healthy human faeces, both when tested individually and when tested as a mixture[21]. Therefore, development of a mixture of SCFAs as a new therapeutic could be beneficial to restrict the growth of both VRE and CRE in the intestine.

Although individually acetate, propionate, butyrate, and valerate significantly suppressed *E. faecium* and *E. faecalis* growth at high concentrations, growth was not fully suppressed. Therefore, we were also interested in whether a mixture of these metabolites showed synergistic or additive effects, where the metabolite mixture provided higher growth suppression than each metabolite provided individually. We showed that the APBV and PBV mixtures provided complete or near complete suppression of *E. faecium* and *E. faecalis* growth at pH 6. The additive effects of these metabolites suggests that they act through a similar mechanism. A previous study by Sorbara and colleagues demonstrated that acetate, propionate, and butyrate inhibited antibiotic-resistant *Enterobacteriaceae* growth through intracellular acidification[33]. Although the inhibitory mechanism of valerate has not been demonstrated yet, it has also been shown to inhibit *C. difficile* and CRE growth[21,24]. Both butyrate and valerate have also been demonstrated to promote intestinal barrier function which may help to prevent dissemination of VRE from the intestine[44]. Further dose testing is required to determine the optimal concentration of each metabolite within this inhibitory metabolite mixture. However, this metabolite mixture shows potential for the use to restrict VRE growth when administered to a patient with VRE intestinal colonisation or when administered to a patient as a prophylactic treatment (e.g. during antibiotic treatment when the patient is susceptible to VRE intestinal colonisation).

We investigated the ability of each metabolite to individually suppress VRE growth at physiologically relevant concentrations, where we tested each metabolite at their low, average, and high concentrations measured in healthy human faeces. However, metabolite concentrations vary along the length of the gastrointestinal tract. SCFA and branched chain fatty acid concentrations are high in the caecum

and ascending colon and progressively decrease in the transverse and descending colon, with a small increase in the sigmoid/rectum[35]. Therefore, microbial metabolites may provide greater suppression of VRE growth in different regions of the intestine.

Another mechanism of colonisation resistance involved in restricting VRE growth is nutrient competition. We demonstrated that killing gut commensals with antibiotics that promote VRE intestinal colonisation resulted in an increase in a broad range of nutrients that contained both carbon and nitrogen sources (including monosaccharides, disaccharides, amino acids, uracil, and succinate). Our group recently demonstrated that antibiotics that promote CRE intestinal colonisation had a similar increase in the concentration of a wide range of nutrients and decrease in the concentration of microbial metabolites in human and mouse faecal microbiomes[21]. This observation is consistent with the literature, which showed that VRE and CRE intestinal colonisation are both promoted by broad-spectrum antibiotics – for example, both CRO and TZP promote the intestinal colonisation of VRE and CRE[13–15,45]. Therefore, broad-spectrum antibiotics that promote VRE and CRE intestinal colonisation resulted in an increase in the availability of similar nutrients, and we conclude that this may be a broader phenomenon that may apply to other intestinal pathogens that have their growth promoted in the intestine following antibiotic treatment.

Microbes with similar or overlapping nutrient utilisation abilities will compete for similar intestinal niches[46]. In this study we demonstrated that *E. faecium* and *E. faecalis* occupy distinct but overlapping intestinal niches. *E. faecium* and *E. faecalis* can both use monosaccharides (fructose, galactose, glucose, mannose, ribose, N-acetylglucosamine) and disaccharides (maltose, sucrose, trehalose) as carbon sources, and can use amino acids (arginine, glutamate, glycine, histidine, isoleucine, leucine, lysine, methionine, phenylalanine, serine, tryptophan, valine) as nitrogen sources under anaerobic and aerobic conditions. However, there were some differences in the utilisation of specific nutrients between *E. faecium* and *E. faecalis*. For example, *E. faecium* showed higher growth on arabinose, cysteine, threonine, and tyrosine under both anaerobic and aerobic conditions, and showed higher growth on aspartate under aerobic conditions. *E. faecalis* showed higher growth on asparagine under anaerobic conditions and higher growth on alanine and uracil under aerobic conditions. Therefore, identification of gut commensal strains that can deplete these nutrients will allow us to develop novel microbiome therapeutics that can re-establish colonisation resistance and restrict VRE intestinal growth. However, we demonstrated that both *E. faecium* and *E. faecalis* can utilise a wide range of nutrients, and this flexibility in nutrient utilisation suggests that microbiome therapeutics targeting VRE must contain several gut commensals that together can deplete all the nutrients that VRE can use to grow.

A previous study by Lebreton and colleagues found that the hospital-associated *E. faecium* lineage lost pathways related to complex carbohydrate utilisation (found in the community-associated *E. faecium* lineage), which were replaced by pathways on mobile elements that were related to the utilisation of amino sugars[47]. This could be advantageous for hospital-associated *E. faecium* strains in an antibiotic-treated intestine, which we demonstrated had increased concentrations of sugars (including increased concentrations of N-acetylglucosamine) that VRE could utilise.

Previous studies demonstrated that the presence of some nutrients may influence the utilisation of others by bacteria, including stimulating or repressing the utilisation of specific nutrients[21,39,40,48,49]. We demonstrated that *E. faecium* and *E. faecalis* strains showed a preference for some nutrients over others when cultured with a mixture of nutrients that were elevated in antibiotic-treated faecal microbiomes. Some nutrients followed expected nutrient utilisation patterns - nutrients that were highly used in the individual nutrient utilisation assays were quickly depleted when *E. faecium* and *E. faecalis* were cultured in a mixture of nutrients (e.g. glucose and arginine). However, other nutrients that were not used when tested individually were used when VRE were cultured in a mixture of nutrients. For example, we showed that *E. faecium* (NCTC 12202) was not able to use sucrose as a sole carbon source but was able to use sucrose when grown in a mixture of nutrients. In contrast, some nutrients that were used in the individual nutrient utilisation assays were not used by VRE when cultured in the mixture of nutrients. For example, we showed that *E. faecium* (NCTC 12202) highly used threonine when tested in the nitrogen leave-one-out assay, but did not use threonine when grown in a mixture of nutrients. Therefore, nutrient utilisation and preferences in mixtures of nutrients are complex and it is not always straightforward to translate the utilisation of nutrients as single carbon or nitrogen sources to their utilisation in mixtures that VRE would encounter in the intestine in vivo. Nutrient utilisation may also be context dependent, meaning that VRE nutrient utilisation and preferences may change under different environmental conditions. For example, future studies should investigate VRE nutrient utilisation and preferences in the context of different host diets, or in different host microbiomes that vary in terms of composition and antibiotic-resistance profiles. Therefore, the best approach to restrict VRE growth may be to deplete all nutrients that are elevated due to antibiotic treatment, so there are no alternative nutrient sources for VRE to utilise.

We demonstrated that antibiotics that promote VRE and CRE intestinal colonisation enriched for a similar broad range of nutrients, and depleted a similar broad range of microbial metabolites[21]. However, it was not clear whether VRE and CRE occupied distinct intestinal niches, which may impact the design of microbiome therapeutics to restrict the growth of these pathogens. A previous study demonstrated that vancomycin-resistant *E. faecium* and carbapenem-resistant *K. pneumoniae* occupied distinct but overlapping intestinal niches, however, the mechanism of how these pathogens occupied their intestinal niches was not defined[22]. We hypothesised that VRE growth was promoted in an antibiotic-treated faecal microbiome because VRE were able to occupy a distinct nutrient-defined intestinal niche from CRE. Compared to our previous study of CRE nutrient utilisation[21], we demonstrated that VRE and CRE had an overlap in the utilisation of many, but not all nutrients. Under anaerobic conditions both *E. faecium* and *K. pneumoniae* could use arabinose, fructose, galactose, glucose, mannose, N-acetylglucosamine, maltose, sucrose, and trehalose as carbon sources and could use arginine as a nitrogen source. However, *K. pneumoniae* could also use xylose as a carbon source and could use aspartate as a nitrogen source. *E. faecium* could also use ribose as a carbon source and could use glutamate, glycine, isoleucine, leucine, lysine, methionine, phenylalanine, threonine, tryptophan, tyrosine, and valine as nitrogen sources. Under aerobic conditions both *E. faecium* and *K. pneumoniae* could use arabinose, fructose, galactose, glucose, mannose, ribose, N-acetylglucosamine, maltose, sucrose, and trehalose as carbon sources and could use arginine, aspartate, glutamate, and uracil as nitrogen sources. However, *K. pneumoniae* could also use xylose, alanine, proline, and glutamate as carbon sources, and could grow more highly on alanine as a nitrogen source. *E. faecium* could also use tryptophan and leucine as carbon sources, could grow more highly on glycine, isoleucine, leucine, lysine, and methionine as nitrogen sources, and could also use phenylalanine, threonine, tryptophan, tyrosine, and valine as nitrogen sources. These differences in nutrient utilisation between *E. faecium* and *K. pneumoniae* support the observations from our co-culture experiments where both strains were able to grow to high levels in the presence of these nutrients.

VRE and CRE also had different preferences for nutrients that they could use. For example, although both *E. faecium* and *K. pneumoniae* can use aspartate, arginine, lysine, and ribose, *E. faecium* prefers to use arginine and ribose while *K. pneumoniae* prefers to use aspartate and

lysine. It is also possible that CRE and VRE may be able to share nutrients to simultaneously grow together. For example, each species may only need one of the many carbon sources that are available in the intestine following antibiotic treatment to support its growth. Therefore, although similar nutrient- and metabolite-defined landscapes are generated in the intestine with broad-spectrum antibiotics that promote VRE and CRE intestinal colonisation, the niches that these pathogens occupy in the intestine are different. This highlights that VRE have different strategies for utilising nutrients to colonise an antibiotic-treated gut microbiome compared to other multidrug-resistant pathogens, which may have implications for the design of new microbiome therapeutics to decolonise pathogens from the intestine.

The results from this study can be used to inform the rational design of microbiome therapeutics that can re-establish colonisation resistance to restrict VRE intestinal growth. A novel microbiome therapeutic could be composed of a cocktail of live gut commensals that can together deplete nutrients that were enriched with antibiotic treatment and restore the production of microbial metabolites that were decreased with antibiotic treatment. Another option could be a mixture of microbial metabolites that can restrict VRE growth, such as the APBV or PBV mixtures tested in this study. Finally, these two new therapeutics could also be combined, such that we could administer the cocktail of live gut commensals and the mixture of metabolites simultaneously, or we could administer the mixture of metabolites while the patient is on antibiotic treatment and then administer the cocktail of live gut commensals after antibiotic treatment is completed.

More research is required before we can design effective microbiome therapeutics to restrict VRE growth. Gut commensals that are killed by antibiotics that promote VRE intestinal colonisation may be important candidates to reintroduce into the intestine to restore colonisation resistance. We need more information on the nutrient utilisation abilities and metabolite production abilities of these gut commensals (both individually and together as mixtures) to determine how they would grow in a nutrient-enriched and metabolite-depleted intestinal environment following antibiotic treatment. We also need more information about the impact of host diet on VRE and gut commensal intestinal colonisation.

There are some limitations in this study that highlight additional areas that require further research. Faecal samples are more commonly used than small intestine samples to investigate VRE intestinal colonisation in both humans and mice, and as such this study investigated VRE intestinal colonisation in the context of the faecal microbiome, which is more reflective of the large intestine microbiome than other regions of the intestine[12–16,50,51]. VRE colonises the antibiotic-treated large intestine to higher concentrations, however VRE can also colonise the antibiotic-treated small intestine[17]. It would also be important to investigate nutrient availability and microbial metabolite production in other regions of the intestine that VRE can colonise (such as the small intestine) to design microbiome therapeutics that can restrict VRE growth at all intestinal regions. Moreover, we have only tested vanA VRE strains in this study as vanA type resistance is the most dominant genotype worldwide, vanA shows higher levels of overall antibiotic-resistance compared to vanB, and vanA strains were the most commercially available strains to test[52–54]. However, future studies should investigate whether there is a difference in nutrient utilisation and metabolite inhibition of other non-vanA type resistant VRE.

In this study we used a mouse model to confirm that our ex vivo results held up in an in vivo intestinal environment. However, like all experimental models, mice have their limitations. For example, humans and mice have some differences in the bacterial species that compose their gut microbiome, and humans and mice consume different diets[55]. However, it is not possible to perform pathogen intestinal colonisation experiments in humans due to ethical limitations, and as such mice are often used in these studies[14–16,21,33,56]. Humans and mice with healthy gut microbiomes have intact colonisation resistance, where both humans and mice are resistant to VRE intestinal colonisation. Moreover, like humans, mice treated with broad spectrum antibiotics become highly susceptible to VRE intestinal colonisation, where VRE intestinal growth reaches high densities. In both our ex vivo human faecal culture experiments and our in vivo mouse experiments we showed that antibiotic treatment caused an enrichment of nutrients (such as sugars and amino acids) and a depletion of metabolites (such as SCFAs, isobutyrate, lactate, and ethanol). Therefore, despite differences in the gut microbiome composition and diets between humans and mice, our in vivo mouse results were supportive of our ex vivo human faecal culture results.

In summary, we demonstrated that VRE take advantage of the nutrient-enriched and metabolite-depleted intestinal niches that are generated during antibiotic treatment to colonise the intestine. We demonstrated that several antibiotics that promote VRE intestinal colonisation changed the abundance of many nutrients and metabolites. We showed that individually acetate, propionate, butyrate, and valerate partially suppressed VRE growth, but a mixture of these metabolites provided complete or near complete suppression of VRE growth. Healthy gut microbiomes depleted nutrients through nutrient competition, however in antibiotic-treated faecal microbiomes killing of gut commensals enriched for nutrients. VRE can use most of these nutrients as carbon or nitrogen sources, where E. faecium and E. faecalis have some similar and some different preferences for these nutrients. Finally, we showed that E. faecium and E. faecalis occupied overlapping but distinct intestinal niches that permit high simultaneous growth with each other and with CRE. Therefore, designing a microbiome therapeutic consisting of gut commensals that deplete these nutrients could restore colonisation resistance and restrict or prevent VRE intestinal colonisation. Removing an intestinal reservoir for VRE would lead to a reduction in the development of invasive VRE infections. Our study also suggested that designing a microbiome therapeutic consisting of a mixture of gut commensals that can deplete nutrients that were enriched with antibiotic treatment and restore the production of microbial metabolites that were depleted with antibiotic treatment could be used to restrict both VRE and CRE growth.

## Methods
### Materials
Table S4 lists the reagents and oligonucleotides used in this study.

### Bacterial isolates
Vancomycin-resistant E. faecium (NCTC 12202 and NCTC 12204) and vancomycin-resistant E. faecalis (NCTC 12201 and NCTC 13779) were purchased from the UK Health Security Agency. Vancomycin-resistant E. faecium (DSM 25698) and vancomycin-resistant E. faecalis (DSM 116626) were purchased from the Leibniz Institute DSMZ. All strains have vanA-type glycopeptide resistance. VRE isolates with the vanA-type resistance gene were chosen as this gene confers a high level of vancomycin resistance and it is one of the most frequently detected vancomycin-resistance genes in enterococci[53,57].

Carbapenem-resistant E. coli ST617 (NDM-5), K. pneumoniae ST1026 (NDM-1) and E. hormaechei ST278 (NDM-1) were isolated and characterised in a previously published study by Yip and colleagues[21]. These carbapenem-resistant Enterobacteriaceae were isolated from rectal swabs from patients in the Imperial College Healthcare NHS Trust (London, UK), with ethical approval from the London - Queen Square Research Ethics Committee, 19/LO/0112.

### Minimum inhibitory concentrations (MICs)
VAN, MTZ, CRO, TZP and CLI MICs were measured for E. faecium (NCTC 12202) and E. faecalis (NCTC 12201) following standard

protocols[58]. Briefly, MIC assays were carried out using Mueller-Hinton broth in 96-well plates, where the broth was supplemented with no antibiotics or with 2-fold dilutions of each antibiotic ranging from 0.125 to 128 mg/L. Each VRE strain was inoculated into the supplemented broth at $5 \times 10^5$ CFU/mL and plates were incubated aerobically at 37 °C for 18 h. Growth was assessed by taking $OD_{600}$ measurements. EUCAST guidelines provided clinical breakpoints for *Enterococcus* spp. for VAN (≤4 mg/L sensitive, >4 mg/L resistant)[59]. According to EUCAST guidance, *Enterococcus* spp. can be reported as resistant to MTZ, CRO, TZP, and CLI without further testing.

## Growth of VRE in faecal concentrations of antibiotics

VRE growth was measured in gut growth medium supplemented with antibiotics at concentrations found in human faeces. The gut growth medium was composed of the following: unmodified starch (5 g/L), casein (3 g/L), inulin (1 g/L), sodium chloride (0.1 g/L), peptone water (2 g/L), yeast extract (2 g/L), sodium bicarbonate (2 g/L), pectin (2 g/L), xylan (2 g/L), arabinogalactan (2 g/L), calcium chloride (0.01 g/L), porcine gastric mucin type II (4 g/L), potassium phosphate monobasic (0.04 g/L), potassium phosphate dibasic (0.04 g/L), magnesium sulphate heptahydrate (0.01 g/L), hemin (0.005 g/L), menadione (0.001 g/L), bile salts (0.5 g/L), and L-cysteine hydrochloride (0.5 g/L)[60]. Gut growth medium was supplemented with water or with one of the following antibiotics at concentrations found in human faeces: 1000 μg/mL VAN, 25 μg/mL MTZ, 152 μg/mL CRO, 139 μg/mL TZP, and 383 μg/mL CLI[27–32,60]. *E. faecium* (NCTC 12202) or *E. faecalis* (NCTC 12201) was inoculated into the growth medium at a concentration of $10^3$ CFU/ml and cultures were incubated at 37 °C anaerobically for 24 h. VRE growth was quantified in VRE cultures by plating samples on Columbia agar plates and incubating the plates at 37 °C for 24 h.

## Human ethical approval

Ethical approval was received by the South Central - Oxford C Research Ethics Committee (16/SC/0021 and 20/SC/0389) to collect fresh faecal samples from healthy human donors. Faecal donors were aged between 18 and 65 years old, did not have a personal or family history of gastrointestinal disease, and did not have any chronic health issues. There is no biological reason to favour one sex over the other, and therefore both male and female donors were recruited, with approximately equal numbers of male and female donors included in each experiment. Faecal donors did not receive antibiotic treatment for at least 6 months prior to donation. Ten donors consumed an omnivore diet, one donor consumed a vegetarian diet, and one donor consumed a mostly vegetarian diet. None of the donors used regular medication or consumed prebiotics, probiotics, or postbiotics. We obtained informed consent from each participant.

## Human faecal culture experiments

Ex vivo faecal culture experiments were performed using a previously published protocol[21]. These experiments were performed in the standardised gut growth medium under anaerobic conditions, where the anaerobic gas mixture contained 10% $CO_2$, 10% $H_2$, 80% $N_2$[60]. Faecal slurries were prepared by homogenising fresh faeces in degassed 0.9% saline. Each faecal slurry was inoculated into the gut growth medium at a final concentration of 2% (w/v). For each donor, fresh faeces were used to seed all experimental conditions, including each of the antibiotic-treated faecal cultures and the water-treated faecal culture. Each treatment was performed in triplicate for each donor.

The first set of faecal culture experiments measured the impact of five broad-spectrum antibiotics (known to promote VRE intestinal colonisation) on the faecal microbiome from 12 healthy human donors. Donors were aged $29.5 \pm 9.5$ years old and consisted of 7 female donors and 5 male donors. The gut growth medium was supplemented with water (vehicle control) or with one of the following antibiotics at concentrations measured in human faeces: 1000 μg/mL VAN, 25 μg/mL MTZ, 152 μg/mL CRO, 139 μg/mL TZP, and 383 μg/mL CLI[27–32]. Faecal cultures were incubated at 37 °C anaerobically for 24 h. Samples were collected for 16S rRNA gene sequencing, 16S rRNA gene qPCR, and ¹H-NMR spectroscopy. Antibiotic-treated faecal culture samples were compared to water-treated faecal culture samples.

The second set of faecal culture experiments were performed to measure VRE growth in antibiotic-treated and water-treated faecal microbiomes. Faecal samples were collected from 5 to 8 healthy human donors (5 females, 3 males) aged $26.8 \pm 4.3$ years old. The gut growth medium was supplemented with water, MTZ (25 μg/mL), CLI (383 μg/mL), VAN (1000 μg/mL), or CRO (152 μg/mL) at concentrations measured in human faeces. TZP was excluded from this experiment as the VRE isolates used in this study were killed by TZP at faecal concentrations. Faecal cultures were incubated at 37 °C anaerobically for 24 h. The next day, *E. faecium* (NCTC 12202) or *E. faecalis* (NCTC 12201) were inoculated into the faecal cultures at a concentration of $10^3$ CFU/mL and cultures were incubated at 37 °C anaerobically for an additional 24 h. Cultures were plated on Brilliance VRE agar (Thermo Fisher Scientific, UK) to measure VRE growth. Brilliance VRE agar is a selective chromogenic agar plate that can differentiate vancomycin-resistant *E. faecium* and *E. faecalis* based on the colour of their colonies (vancomycin-resistant *E. faecium* grow as purple colonies, while vancomycin-resistant *E. faecalis* grow as light blue colonies). VRE growth was also measured in gut growth medium not supplemented with faeces or antibiotics after incubation at 37 °C anaerobically for 24 h, followed by plating on Brilliance VRE agar.

## 16S rRNA gene sequencing and 16S rRNA gene qPCR

The DNeasy PowerLyzer PowerSoil Kit (Qiagen, Germany) was used to extract DNA from faecal culture samples (DNA extracted from 250 μL per sample). The extraction protocol followed the manufacturer's instructions, with the addition of $3 \times 1$ min bead beating cycles at speed 8 (with a 1-min rest between each cycle) using a FastPrep 24 bead beater (MP Biomedicals, USA). The extracted DNA was stored at −80 °C until it was ready to be analysed.

Sample libraries were prepared for 16S rRNA gene sequencing following Illumina's 16S metagenomic sequencing library preparation protocol and as previously described by our group[21,61]. The V1-V2 region was amplified using a forward primer mix (a 4:1:1:1 ratio of 28F-YM, 28F-Borrelia, 28F-Chloroflex, and 28F-Bifdo primers) and the reverse primer 388R[21,61]. The index PCR reactions were cleaned up and normalised using the SequalPrep Normalisation Plate Kit (Life Technologies, UK). Sample libraries were quantified using the NEBNext Library Quant Kit for Illumina (New England Biolabs, UK). Sequencing was performed on an Illumina MiSeq platform (Illumina Inc., UK) with paired-end 300 bp chemistry using the MiSeq Reagent Kit V3 (Illumina). The resulting data was imported into R where it was processed using the standard DADA2 pipeline (version 1.18.0)[62]. Version 138.1 of the SILVA bacterial database was used for sequence variant classification (https://www.arb-silva.de/).

Bacterial biomass was quantified using 16S rRNA gene qPCR, following a previously published protocol[24]. PCR reactions were prepared in 20 μL volumes, containing 1x Platinum Supermix with ROX, 225 nmol/L BactQUANT probe, 1.8 μmol/L BactQUANT forward primer, 1.8 μmol/L BactQUANT reverse primer, 5 μL DNA, and PCR grade water. A 10-fold dilution series of *E. coli* DNA was used to generate the standard curve. All samples, standards, and controls were analysed in triplicate. Amplification and real-time fluorescence detection was performed on the Applied Biosystems StepOnePlus Real-Time PCR System (Applied Biosystems, Waltham, Massachusetts). The PCR cycle conditions for qPCR were as follows: 50 °C for 3 min for UNG treatment, 95 °C for 10 min for *Taq* polymerase activation, and 40 cycles of 95 °C for 15 s for denaturation and 60 °C for 1 min for annealing and extension.

The 16S rRNA gene sequencing data was converted to absolute abundances using the 16S rRNA gene qPCR data using the following formula, as previously described[24]:

$$\begin{aligned} \text{Absolute abundance of taxa} &= \text{relative abundance of taxa} \\ &\times \left( \frac{16S\, rRNA\, gene\, copy\, number\, in\, sample}{highest\, 16S\, rRNA\, gene\, copy\, number\, in\, sample\, set} \right) \end{aligned} \quad (1)$$

## ¹H-NMR spectroscopy

¹H-NMR spectroscopy was performed on human faecal culture samples, VRE culture samples, and mouse faecal samples to measure absolute quantification of metabolites, following previously published protocols[21,24].

Human faecal culture samples and VRE culture samples were prepared as follows. Supernatants were prepared by centrifuging samples at $17,000 \times g$ for 10 min at 4 °C (the pellet was discarded), and supernatants were frozen down until ready to be analysed. Then, supernatants were randomised and defrosted at room temperature. Next, supernatants were centrifuged for 10 min at $17,000 \times g$ and 4 °C (the pellet was discarded). The NMR buffer contained the following per L: 28.85 g $Na_2HPO_4$, 5.25 g $NaH_2PO_4$, 1 mM 3-(trimethylsilyl)propionic-2,2,3,3-$d_4$ acid sodium salt (TSP), 3 mM $NaN_3$, deuterium oxide to 1 L, pH 7.4. For each sample, 400 µl of sample supernatant was mixed with 250 µl NMR buffer.

Mouse faecal samples were prepared as follows. Mouse faecal samples were randomised and defrosted at room temperature. Mouse faecal supernatants were prepared by homogenising 20 mg of mouse faeces in 560 µl of high-performance liquid chromatography-grade water by vortexing for 10 min at 3000 rpm, followed by centrifuging samples at $17,000 \times g$ for 10 min at 4 °C (the pellet was discarded). The NMR buffer contained the following per 100 mL: 20.4 g $KH_2PO_4$, 5.8 mM TSP, 2 mM $NaN_3$, deuterium oxide to 100 mL, pH 7.4. For each sample, 540 µl of supernatant was mixed with 60 µl NMR buffer.

For each sample 600 µl of sample supernatant mixed with NMR buffer was loaded into 5 mm NMR tubes and loaded into the SampleJet system. ¹H-NMR spectra were acquired at 300 K using a Bruker AVANCE III 800 MHz NMR spectrometer (Bruker Bio-Spin, Rheinstetten, Germany) fitted with a 5 mm CPTCI 1H-13C/15 N D Z-gradient cryoprobe. 1D spectra were acquired using a standard NOESYGPPR1D pulse sequence (RD-90°-t1-90°-tm-90°-ACQ). A 4 s recycle delay, 100 ms mixing time, and -10 µs 90° pulse length were used. For the human faecal culture samples and VRE culture samples 32 scans were recorded. For the mouse faecal samples 128 scans were recorded.

NMR spectra pre-processing was performed in Topspin (v3.2.6), with an exponential window function (line broadened by 0.3 Hz), zero-order auto phasing (command apk0), baseline correction (command abs), and referencing TSP (command sref). Further processing was performed by importing NMR spectra into MATLAB r2019b (The Mathworks, USA) using a custom script, as previously described[21]. Following alignment of the spectra the water peak was removed by cutting region 4.6–5.0 ppm and the TSP peak was removed by cutting region −0.2–0.2 ppm. Chenomx NMR Suite v9.02 (Chenomx, Canada) was used for peak identification and representative peaks were selected and integrated. To define the limit of detection, a signal-to-noise ratio cut-off of 3:1 was used[63]. The minimal signal-to-noise ratio for each dataset was as follows: 8.3:1 for the human faecal culture samples, 8.5:1 for the VRE culture samples, 3.2:1 for the mouse faecal samples.

## Mouse model of intestinal colonisation with vancomycin-resistant *E. faecium*

The Imperial College London Animal Welfare and Ethical Review Body (PF93C158E) provided ethical approval for the mouse experiments. The mouse experiment was performed under the authority of the UK Home Office, as described in the Animals (Scientific Procedures) Act 1986. We adhered to standards outlined in the Animal Research: Reporting of In Vivo Experiments (ARRIVE) guidelines.

Female wild-type C57BL/6 mice (8–10 weeks old) were purchased from Envigo (Huntingdon, UK). Mice were housed 5 per cage in individually ventilated cages with bedding (Aspen chip 2 bedding, NEPCO, Warrensburg, New York). Mice were maintained at 20–22 °C and 45–65% humidity, with 12-h light and dark cycles. Mice were given autoclaved food (RM1, Special Diet Services, Essex, UK) and water (provided ad libitum). Prior to the start of the experiment mice were acclimatised for 1 week.

The mouse experiment was designed to measure the impact of VAN treatment on nutrient availability, metabolite production, and vancomycin-resistant *E. faecium* intestinal growth. Mice were orally gavaged with VAN or water once daily for 7 days, and faecal samples were collected for ¹H-NMR spectroscopy 24 h after the last dose of VAN was administered. VAN dosing was based on previously published study, where mice were orally gavaged with water or with VAN at a dose of 50 mg/kg/day[64]. Both groups of mice were orally gavaged with $10^6$ CFU per 200 µl of vancomycin-resistant *E. faecium* (NCTC 12202) 4 days after the last dose of VAN, and 4 days later faecal samples were collected from mice to plate on Brilliance VRE agar to quantify *E. faecium* growth.

## Mixed metabolite inhibition assay

Microbial metabolites that were decreased with antibiotic treatment were tested together as a mixture in an inhibition assay to see if they could suppress VRE growth. The metabolite concentrations found in healthy faeces were obtained from a previously published study, and included the low, average, and high concentrations measured in healthy faeces (see Table S5)[21].

In the first experiment we tested a mixture of formate, acetate, propionate, butyrate, valerate, isobutyrate, isovalerate, lactate, 5-aminovalerate, and ethanol, where metabolites mimicked their concentrations measured in healthy human faeces. For the low mixture all metabolites were tested at the low concentrations measured in faeces, for the average mixture all metabolites were tested at the average concentrations measured in faeces, and for the high mixture all metabolites were tested at the high concentrations measured in faeces. Tryptic soy broth was supplemented with the metabolite mixture and the pH of the supplemented broth was adjusted to 6, 6.5, or 7 to mimic the pH of the healthy intestine[36,65]. *E. faecium* (NCTC 12202) or *E. faecalis* (NCTC 12201) were inoculated into the broth at a concentration of $10^3$ CFU/ml, cultures were incubated at 37 °C anaerobically overnight, and $OD_{600}$ measurements were taken the next day.

In the second experiment we tested a mixture of acetate, propionate, butyrate, and valerate (APBV mixture), or a mixture of propionate, butyrate, and valerate (PBV mixture), where metabolite concentrations mimicked high concentrations measured in healthy human faeces. Tryptic soy broth was supplemented with the metabolite mixture or unsupplemented (no metabolite control) and the pH of the broth was adjusted to 6, 6.5, or 7. Vancomycin-resistant *E. faecium* strains or *E. faecalis* strains were inoculated into the broth at a concentration of $10^3$ CFU/ml, cultures were incubated at 37 °C anaerobically overnight, and $OD_{600}$ measurements were taken the next day.

## Individual metabolite inhibition assay

Microbial metabolites were individually tested to determine whether they could individually suppress VRE growth. Each metabolite was tested at concentrations mimicking the low, average, and high concentrations measured in healthy human faeces[21]. We tested the following individual metabolites: formate, acetate, propionate, butyrate, valerate, isobutyrate, isovalerate, lactate, 5-aminovalerate, and ethanol. We also tested each individual metabolite as 2-fold dilutions between 4-512 mM to calculate the half-maximal inhibitory concentrations ($IC_{50}$). Tryptic soy broth was supplemented with each

metabolite and the pH of the supplemented broth was adjusted to 6. Vancomycin-resistant *E. faecium* strains or *E. faecalis* strains were inoculated into the broth at a concentration of $10^3$ CFU/ml, cultures were incubated at 37 °C anaerobically overnight, and $OD_{600}$ measurements were taken the next day.

### Growth of VRE in faecal culture supernatants and VRE supernatant

Basal minimal medium (see Supplementary Data 1) was supplemented with a mixture of nutrients that were enriched in antibiotic-treated faecal microbiomes at a final concentration of 0.015% (w/v): L-alanine, L-arginine, L-aspartate, L-glutamate, L-glycine, L-isoleucine, L-leucine, L-lysine, L-methionine, L-phenylalanine, L-proline, L-threonine, L-tryptophan, L-tyrosine, L-valine, L-arabinose, D-fructose, L-fucose, D-galactose, D-glucose, D-mannose, D-ribose, D-xylose, N-acetylglucosamine, D-maltose, sucrose, D-trehalose, uracil and succinate. The supplemented basal minimal medium was inoculated with fresh healthy human faeces (2 female donors and 1 male donor, average age $29.3 \pm 4.0$ years old), a single VRE strain (*E. faecium* NCTC 12202 or *E. faecalis* NCTC 12201), or left uninoculated. To prepare the faecal inoculum, fresh faeces were prepared as a 10% (w/v) faecal slurry in 0.9% (w/v) saline and homogenised in a stomacher at 250 rpm for 1 min. The faecal slurry was diluted 1 in 10 into degassed supplemented basal minimal medium to give a final concentration of 1%. *E. faecium* (NCTC 12202) and *E. faecalis* (NCTC 12201) were inoculated into degassed supplemented basal minimal medium at a concentration of $10^7$ CFU/ml. Cultures were incubated at 37 °C anaerobically for 24 h. The next day, the spent supernatants were adjusted to pH 7, filter sterilised, and inoculated with a single VRE strain at $10^6$ CFU/ml. As a control, spent supernatants were also supplemented with 0.5% (w/v) glucose to confirm nutrient depletion. Cultures were incubated at 37 °C anaerobically for 24 h, then cultures were plated on fastidious anaerobe agar with 5% (v/v) defibrinated horse blood to quantify VRE growth.

### Carbon and nitrogen utilisation assays

Nutrients increased in antibiotic-treated faecal cultures were tested for their potential to serve as individual carbon and/or nitrogen sources for *E. faecium* and *E. faecalis*. We designed an *Enterococcus* minimal medium to test the individual nutrients, which did not allow any background growth without a supplemented carbon and nitrogen source. The minimal medium consisted of the following: potassium phosphate monobasic (4 g/L), potassium phosphate dibasic (14 g/L), sodium citrate (1 g/L), magnesium sulphate (0.2 g/L), ammonium sulphate (2 g/L), adenosine (0.04 g/L), thymine (0.04 g/L), Kao and Michayluk vitamin solution (Merck, Cat. No. K3129; 10 mL/L). If the study was performed under anaerobic conditions, fumarate (5.2 g/L) was added to increase growth of VRE strains. The minimal medium was sterilised with a 0.22 μm filter and used immediately for aerobic incubations or degassed overnight if used for anaerobic incubations.

For the carbon assays the minimal medium was supplemented with the RPMI-1640 amino acid solution (Merck, Cat. No. R7131; 50 mL/L) as nitrogen sources, and each individual nutrient was supplemented in at a concentration of 0.5% (w/v) to test their utilisation as a carbon source.

For the nitrogen assays the minimal medium was supplemented with 0.5% (w/v) glucose as a carbon source. As *E. faecium* and *E. faecalis* had undefined amino acid auxotrophies, nitrogen utilisation was assessed by comparing growth of the nitrogen mixture lacking one of the nitrogen sources to growth on the complete nitrogen mixture (leave-one-out experimental design). The complete all nitrogen mixture contained the following: L-alanine (2 g/L), L-arginine (2.5 g/L), L-asparagine (0.71 g/L), L-aspartate (0.25 g/L), L-cysteine (0.625 g/L), L-glutamate (0.25 g/L), L-glycine (0.125 g/L), L-histidine (0.1875 g/L), L-isoleucine (0.625 g/L), L-leucine (0.625 g/L), L-lysine HCl (0.5 g/L), L-methionine (0.1875 g/L), L-phenylalanine (0.1875 g/L), L-proline (0.25 g/L), L-serine (0.375 g/L), L-threonine

(0.25 g/L), L-tryptophan (0.0625 g/L), L-tyrosine (0.145 g/L), L-valine (0.25 g/L), N-acetylglucosamine (2 g/L), and uracil (0.5 g/L). Each leave-one-out mixture excluded one of the nitrogen sources found in the all nitrogen mixture.

For both carbon and nitrogen assays, VRE strains were grown on Reasoner's 2 A (R2A) agar overnight and inoculated into supplemented minimal medium at a starting $OD_{600}$ of -0.05. Cultures were incubated either anaerobically or aerobically in a plate reader which measured $OD_{600}$ every 15 min for 24 h.

### Mixed nutrient preference assays

Nutrient preference was assessed for *E. faecium* (NCTC 12202) and *E. faecalis* (NCTC 12201) by inoculating each strain in a minimal medium supplemented with a mixture of nutrients that were increased in antibiotic-treated faecal cultures. Decreases in the concentrations of each nutrient were measured over time. The *Enterococcus* minimal medium was supplemented with a mix of the following nutrients at 0.015% (w/v) for each nutrient: L-alanine, L-arginine, L-aspartate, L-glutamate, L-glycine, L-isoleucine, L-leucine, L-lysine, L-methionine, L-phenylalanine, L-proline, L-threonine, L-tryptophan, L-tyrosine, L-valine, L-arabinose, D-fructose, L-fucose, D-galactose, D-glucose, D-mannose, D-ribose, D-xylose, N-acetylglucosamine, D-maltose, sucrose, D-trehalose, uracil and succinate.

VRE strains were grown on Reasoner's 2 A (R2A) agar overnight and inoculated into the supplemented minimal medium at a starting $OD_{600}$ of -0.05. Cultures were incubated either anaerobically or aerobically in a plate reader which measured $OD_{600}$ every 15 min for 24 h. Samples were collected for $^1$H-NMR spectroscopy at 0, 4.5, 9, and 24 h. Nutrient utilisation was expressed as a percentage of the peak integration value for each nutrient at 0 h.

### Analysis of Microbial Growth Assays (AMiGA)

The growth curve data generated for *E. faecium* and *E. faecalis* strains were analysed using the AMiGA software (available at https://github.com/firasmidani/amiga) in Python (v3.6.5 or v3.12.9)[66]. The carbon source utilisation growth curves and mixed nutrient growth curves were normalised to the no carbon control using the subtraction method, where AMiGA subtracted the growth parameters of the no carbon control from the carbon test samples. The nitrogen utilisation growth curves were also normalised to the no nitrogen control using the subtraction method, where AMiGA subtracted the growth parameters of the no nitrogen control from the nitrogen test samples. Differential growth was compared between the substrate and the no carbon control for the carbon assays, or to the all nitrogen control for the nitrogen assays by performing Gaussian Process regression in AMiGA. The functional difference and its 95% credible interval were also calculated between the substrate and no carbon control for the carbon assay, or to the all nitrogen control for the nitrogen assays.

### VRE and CRE co-culture assays

VRE strains (*E. faecium* NCTC 12202 or *E. faecalis* NCTC 12201) and CRE strains (*E. coli*, *K. pneumoniae*, or *E. hormaechei*) were co-cultured in a basal minimal medium supplemented with a mixture of nutrients that are enriched in antibiotic-treated faecal microbiomes. The composition of the basal minimal medium and the steps used to prepare it are shown in Supplementary Data 1. The minimal medium was supplemented with a mixture of nutrients that were enriched in antibiotic-treated faecal microbiomes at a final concentration of 0.015% (w/v): L-alanine, L-arginine, L-aspartate, L-glutamate, L-glycine, L-isoleucine, L-leucine, L-lysine, L-methionine, L-phenylalanine, L-proline, L-threonine, L-tryptophan, L-tyrosine, L-valine, L-arabinose, D-fructose, L-fucose, D-galactose, D-glucose, D-mannose, D-ribose, D-xylose, N-acetylglucosamine, D-maltose, sucrose, D-trehalose, uracil and succinate.

The following co-cultures were tested and compared to their growth as monocultures: *E. faecium + E. faecalis*, *E. faecium + K. pneumoniae*, *E. faecium + E. coli*, *E. faecium + E. hormaechei*, *E. faecalis + K. pneumoniae*, *E. faecalis + E. coli*, *E. faecalis + E. hormaechei*. Each strain was inoculated into the minimal medium at a concentration of $10^3$ CFU/ml and the cultures were incubated for 24 h under anaerobic conditions. Pathogen growth was quantified by plating samples on Brilliance VRE plates to quantify *E. faecium* or *E. faecalis* growth and on Brilliance CRE plates to quantify *K. pneumoniae*, *E. coli*, or *E. hormaechei* growth. Growth of each isolate was measured with 6 replicates from two independent experiments.

## Statistical analysis

GraphPad Prism 10.1.0 (La Jolla, California) or R (v4.2.3) were used to perform the statistical tests. P values or q values less than 0.05 were considered to be significant. For the ex vivo human faecal culture experiment, the 16S rRNA gene sequencing data (log transformed absolute abundances) and the ${}^1$H-NMR spectroscopy data (peak integrations) were analysed using the Wilcoxon signed rank test (two-sided) with Benjamini Hochberg false discovery rate (FDR) correction using the DA.wil function within in the DAtest R package version 2.7.18. For the log transformed *E. faecium* and *E. faecalis* plate counts, antibiotic-treated faecal culture counts were compared to water-treated faecal culture counts using a mixed effects model (one-way) with Dunnett's multiple comparison, and no faeces counts were compared to water-treated faecal culture counts using an unpaired t-test (two-sided). For the experiment measuring *E. faecium* and *E. faecalis* growth in concentrations of antibiotics measured in human faeces, antibiotic- or water-treated samples were compared to the inoculum using a one-way ANOVA with Dunnett's multiple comparison test. For the mouse experiment, the ${}^1$H-NMR spectroscopy data (peak integrations) were analysed using an unpaired t-test (two-sided) with Benjamini Hochberg FDR using the t_test function within the rstatix R package version 0.7.2. The log transformed *E. faecium* plate counts were compared using Mann–Whitney U test (two-sided). For the mixed metabolite inhibition assays, a Kruskal-Wallis with Dunn's multiple comparison test compared the no metabolite control to each concentration of the metabolite mixture at the same pH. For the individual metabolite inhibition assay, a Kruskal-Wallis test with Dunn's multiple comparison test compared the no metabolite control to each concentration of the individual metabolite. $IC_{50}$ values were calculated for each individual metabolite in Graphpad Prism. The mixed nutrient growth curves, carbon assay growth curves, and nitrogen assay growth curves were analysed using the AMiGA software in Python (v3.6.5) as described above. For the supernatant growth tests, comparisons between growth in the spent supernatants and fresh medium used a Kruskal-Wallis with Dunn's multiple comparison test, while comparison of unsupplemented vs glucose-supplemented supernatants used a Wilcoxon signed rank test (two-sided). For the nutrient preference experiment comparisons were made between the *E. faecium* or *E. faecalis* cultures and the sterile media controls over time using a two-way mixed ANOVA followed by pairwise comparisons with Bonferroni correction. For the *E. faecium* and *E. faecalis* co-culture experiment, log transformed monoculture plate counts were compared to log transformed co-culture plate counts using an unpaired t-test (two-sided). For the VRE strain and CRE strain co-culture experiment, log transformed monoculture plate counts were compared to log transformed co-culture plate counts using a one-way ANOVA with Dunnett's multiple comparison test.

## Reporting summary

Further information on research design is available in the Nature Portfolio Reporting Summary linked to this article.

## Data availability

We have provided source data with this paper. The 16S rRNA gene sequencing, 16S rRNA gene qPCR, and ${}^1$H-NMR spectroscopy datasets were deposited into the Figshare repository at https://doi.org/10.6084/m9.figshare.28001687.v1. The 16S rRNA gene sequencing raw reads have also been deposited into the ENA under BioProject PRJEB83283 with sample accession numbers and links listed in the Fig. 1 Source Data file. The SILVA bacterial database version 138.1 can be found at https://www.arb-silva.de/. The compound database for the Chenomx NMR Suite v9.02 can be found at https://www.chenomx.com/. Source data are provided with this paper.

## Code availability

MATLAB r2019b was used to import and process ${}^1$H-NMR spectra using a custom script which is available in the Figshare repository at https://doi.org/10.6084/m9.figshare.28001687.v1, using code found in the Zenodo repository at https://doi.org/10.5281/zenodo.3077413.

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

## Acknowledgements

This work was supported by funding awarded to JAKM: a Wellcome Trust Career Development Award (227795/Z/23/Z) and start-up funds from the Department of Life Sciences at Imperial College London. TBC received funding from a UKRI Impact Accelerator Award (MR/X502959/1) and the Rosetrees Trust and Stoneygate Trust (100275). BHM was the recipient of an NIHR Academic Clinical Lectureship (CL-2019-21-002) and is now the recipient of a Medical Research Council (MRC) Clinician Scientist Fellowship (MR/Z504002/1). Financial and infrastructure support was provided to the Division of Digestive Diseases from the National Institute for Health Research (NIHR) Imperial Biomedical Research Centre (BRC) based at Imperial College Healthcare NHS Trust and Imperial College London.

## Author contributions

O.G.K.—formal analysis: equal; investigation: lead; methodology: equal; visualisation: equal; writing – review & editing: supporting. A.Y.G.Y.— formal analysis: supporting; investigation: supporting; writing – review & editing: supporting. V.H.—formal analysis: supporting; investigation: supporting; writing – review & editing: supporting. J.M.B.—investigation: supporting. J.R.M.—resources: supporting; writing – review & editing: supporting. B.H.M.—resources: supporting; writing – review & editing: supporting. T.B.C.—investigation: supporting; methodology: supporting; writing – review & editing: supporting. J.A.K.M.—conceptualisation: lead; data curation: lead; formal analysis: equal; funding acquisition: lead; methodology: equal; project administration: lead; resources: lead; supervision: lead; visualisation: equal; writing – original draft: lead; writing – review & editing: lead.

## Competing interests

J.A.K.M., A.Y.G.Y. and O.G.K. filed a patent application (patent application number 2217266.2) related to this work. B.H.M. received consultancy fees from Finch Therapeutics Group and Ferring Pharmaceuticals outside the submitted work. J.R.M. received consultancy fees from EnteroBiotix Ltd., Cultech Ltd. and Crescent Enterprise Ltd. outside the submitted work. All other authors declare no competing interests.
