## [Transparent Peer Review file · Nature Communications]

Vancomycin-resistant enterococci utilise antibiotic-enriched nutrients for intestinal colonisation

Corresponding Author: Dr Julie McDonald

Version 0:

Reviewer comments:

Reviewer #1

(Remarks to the Author)

King and colleagues have investigated expansion of vancomycin-resistant *Enterococcus faecium* and *E. faecalis* in antibiotic-treated fecal cultures and antibiotic-treated mice. They demonstrate that antibiotics reduce the density of commensal bacterial species and microbial metabolite concentrations while increasing nutrient concentrations. Antibiotic treatment also leads to increased expansion of VRE. The authors extend these studies by growing VRE in trypticase soy broth supplemented with various concentrations of metabolites that were reduced in feces by antibiotic treatment and demonstrate that at pH 6 they inhibit VRE growth. Interestingly, upon testing with individual metabolites, the authors discovered differences in sensitivity to butyrate, 5-aminovalerate and ethanol between *E. faecium* and *E. faecalis*. The authors investigate the role of different carbon and nitrogen sources on growth of VRE and demonstrate the broad range of substrates they can use. Here again they demonstrate differences between *E. faecium* and *E. faecalis* that are potentially interesting. The kinetics of nutrient depletion by VRE are investigated and demonstrate some interesting differences between *E. faecium* and *E. faecalis*, particularly in depletion of different sugars. The final figure demonstrates some competition between *E. faecium* and *E. faecalis* but minimal competition between VRE and different Gram-negative organisms.

VRE infection is an important clinical problem and clinical and experimental studies have demonstrated the dramatic expansion of VRE in the intestinal lumen of humans and antibiotic-treated mice. The authors have generated some interesting but preliminary and incompletely investigated findings. The authors exaggerate the paucity of information on mechanisms of colonization resistance against VRE (lines 58-9) and should mention the role of the microbiota in stimulating host secretion of antimicrobial C-type lectins to inhibit VRE in the small intestine and production of VRE-inhibitory bacteriocins by commensal bacterial strains. With respect to inhibition of VRE by microbial metabolites, a recent manuscript demonstrated that metabolites produced by *Bifidobacterium* species and media acidification can inhibit VRE growth and may account for reduced infections in patients with liver disease (PMID: 37845315). Nevertheless, the role of nutrient depletion has not been adequately investigated and this manuscript makes important progress in that direction. The results, however, provide a very limited perspective and do not address the fact that VRE colonizes the entire intestine, and thus nutrients that are available in the small intestine likely play a significant role in expansion and domination.

Major Issues:

There are several surprising results shown in Figure 1A. First, clindamycin has broad activity against obligate anaerobes but seems to only reduce the density of *Bifidobacteriaceae*. Is there an explanation for this result? Second, almost all human fecal samples have endogenous *Enterococci* that would not be sensitive to, for example, metronidazole or ceftriaxone. It is surprising therefore, that endogenous *Enterococci* did not expand in fecal cultures. It raises the possibility that VRE strains may differ in some way from endogenous, commensal *Enterococcal* species/strains. An additional issue with Figure 1, panels A and B, is that quantitation is only relative but not absolute. Thus, for bacterial strains or metabolites that are present at very low densities/concentrations, slight increases can look dramatic but still fall below a range that is biologically important.

Throughout the manuscript, the authors describe differences in sensitivity of *E. faecium* and *E. faecalis* in terms of sensitivity to butyrate or dependence on different amino acids. These differences are very interesting and may provide insights into shifting prevalences of these two species in clinical settings over time and regionally. That said, the findings presented in this manuscript only focus on one strain of each species. Given the remarkable strain-to-strain variation that can occur within bacterial species, the authors should confirm these results with panels of *E. faecium* and *E. faecalis* strains.

In lines 263-268, the authors point out that metabolite concentrations measured in feces do not fully inhibit VRE. Many microbial metabolites, butyrate in particular, are metabolized by intestinal epithelial cells and thus concentrations vary along

the length of the intestine. The highest concentrations of butyrate are detected in the cecum and proximal large intestine of mice and likely also humans.

In Figure 7, did *E. faecium* not grow on sucrose, or was it not tested?

In figures 7 and 8, the authors show interesting results for VRE growth in minimal media in the presence of different carbon and nitrogen sources. These results are interesting and raise many questions but the authors do not provide any new insights. At a minimum, the authors should correlate genomic differences between *E. faecium* and *E. faecalis* with differences in carbohydrate and amino acid metabolism. It would be interesting to investigate VRE transcriptomes, for example, in these different media. These types of analyses would enable the investigators to potentially identify genes and operons involved in carbohydrate and amino acid metabolism.

The results shown in Figure 9 are interesting but inadequately interpreted by the authors. The three timepoints they investigate focus on log phase (4.5 hours), early stationary (9 hours) and late stationary phase (24 hours). Thus, between 9 and 24 hours, expansion has ceased. The authors should discuss these different scenarios and perhaps consider that the shifts they detect may not result from depletion of preferred substrates but from dramatic changes in metabolism that occur when microbes progress from exponential growth to the stationary state.

Minor issues:

Line 187 should state that oral vancomycin was used.

Figures 7 and 8 are difficult to interpret. Do the authors have CFU data to complement the OD values?

Reviewer #2

(Remarks to the Author)

This manuscript makes an important contribution to research in the field of antibiotic resistance, a global health challenge that is of utmost importance.

The title of the manuscript is: "Vancomycin-resistant *Enterococcus* colonise the antibiotic-treated intestine by occupying distinct nutrient- and metabolite-defined intestinal niches".

First, I would suggest using the name 'Vancomycin-resistant enterococci' as it is used as a plural in the title, but also throughout the entire manuscript. Therefore, the abbreviation VRE stands for 'Vancomycin-resistant enterococci', and needs to be adapted in the manuscript (line 21 & 43).

The research shows that VRE occupy distinct but overlapping intestinal niches. However, in the title, it is stated that VRE only occupy distinct niches. Not sure if the title needs to be adjusted, but there is a discrepancy between the title and the message of the article.

The mouse experiment only confirms the overall findings of the ex vivo experiment, but how do we translate the findings to humans? Mice consume a different diet; respond differently to antibiotics than humans, and have a much faster metabolism leading to different nutrient and metabolic phenotypes. Can some interpretation be added to the discussion?

Stool samples of healthy individuals were examined. Do the authors have more information on the dietary pattern, use of medication, intake of pre-, pro-, and postbiotics, inflammatory parameters, serum SCFA, ...? Was a colonoscopy performed on these subjects to rule out any significant inflammation? As these factors can all influence faecal nutrient and microbial metabolite concentrations, this could influence the interpretation of the findings.

Faeces of twelve healthy human donors was used in the experiments. Why is there a difference in the number of donors in Fig 1 (n = 5-8) and Fig 6 (n = 3), and which samples from which donors were selected for the experiments described in Fig 1 and Fig 6?

The gastrointestinal tract of hospitalized patients presents a markedly different environment compared to the healthy human gut, offering a unique metabolic landscape for intestinal microorganisms (Stellfox et al., 2022, <https://doi.org:10.1128/mbio.00670-22>). In this study, antibiotic-treated healthy human faeces were used to explore colonization resistance, nutrient availability, and the production of microbial metabolites. However, treatment needs to be established for hospitalized patients and not for healthy humans treated with antibiotics. Why didn't the researchers examine patients' stool samples?

Genomic studies have shown that hospital-associated *E. faecium* strains have acquired specific genes enabling them to access and utilize nutrient sources unavailable to commensal strains, such as amino-sugars found on cell surfaces and in mucin. This metabolic shift distinguishes them from community-associated strains, which typically utilize complex carbohydrates (Lebreton et al., 2013, doi <https://doi.org:10.1128/mBio.00534-13>; Cattoir 2022, <https://doi.org:10.1016/j.mib.2021.10.013>).

Why did the researchers choose to use 1H-NMR spectroscopy and not untargeted LC/GC-MS which can cover a wider range of metabolites? Would MS analyses provide added value?

How can the authors guarantee that this study has sufficient statistical power? Were power calculations performed? If so, please add this information to the manuscript.

Fig 6: the preferred approach when using non-parametric tests, including the Wilcoxon signed-rank test or Kruskal-Wallis test, is to report medians along with interquartile ranges (IQR), as these better reflect the central tendency and variability in non-normally distributed data. If mean \pm SD is reported alongside a Wilcoxon signed-rank test, the authors should clarify

why they chose this presentation (e.g., for comparability to other studies or because the data are close to normally distributed

The authors mention several times that novel microbiome therapeutics can be used to re-establish colonization resistance and restrict VRE intestinal growth, and that gut commensals can be used for this purpose. What specific therapies are being considered and can this be discussed more in depth? In what way can the current results be implemented in the clinical field?

Some minor remarks:

- Figure 3: please change low/average/high 'metabolites' into low/average/high 'concentration'
- Line 92 .. niches, as they showed that growth (of what?) was not impaired...
- Line 104 .. colonisation decreased the concentration of a wide range of metabolites in the gut microbiome (change into microbial metabolites?)
- Line 143-146: listing how many bacterial families' concentration was increased/decreased per antibiotic seems quite irrelevant info. Can the authors limit the information to what families' concentration changes and the most important findings?
- Line 149-160: iso-valerate shows a different result as compared to the other SCFA and BCAA, but this is not described.
- Line 217 (add) human microbiomes showed moderate inhibition of VRE growth
- Line 223-224: We were therefore interested in whether microbial metabolites - that were (delete 'were') decreased in concentration by antibiotics - could also inhibit VRE growth (add) at physiologically relevant concentrations.
- Line 238-239 (also 241-243) *E. faecium* (add) growth was moderately inhibited by propionate, butyrate, and valerate at the high concentration, and propionate also inhibited *E. faecium* (add) growth at the average concentration (Fig. 4a).
- Line 270 Nutrients enriched in antibiotic-treated (add) human faecal microbiomes supported VRE growth
- Line 272-273 We demonstrated that antibiotics that promote VRE intestinal colonisation increased the (delete) concentration of nutrients (add) nutrient availability in faecal microbiomes (Figs. 1-2).
- Line 289-294: move to Methods section?
- Line 301-302: These results support the hypothesis that nutrient depletion (delete) was responsible for (add) can contribute to suppressing VRE growth in a healthy gut microbiome.
- Line 652 typo: through instead of though

Reviewer #3

(Remarks to the Author)

King et al., Vancomycin-resistant *Enterococcus* colonize the antibiotic-treated intestine by occupying distinct nutrient- and metabolite-defined intestinal niches

In the present work, the authors investigate antibiotic-induced niches in the gut micromilieu that facilitate the colonization and/or expansion of *Enterococcus* spp. with a focus on VREs. The authors demonstrate that multiple antibiotics modulate the availability of nutrients for *Enterococcus*, but also decrease microbial metabolites that can inhibit *Enterococcus*. Notably, they also found that *Enterococcus* can occupy overlapping niches with other gut-residing pathobionts. This work extends prior reports of Dr. McDonald, and adds important aspects of microbiota-intrinsic regulation of pathobiont colonization and expansion.

I have a few comments that require attention by the authors:

1. Ex vivo fecal culture experiments, method: Human microbiotas were treated for 24h with various antibiotics and afterwards harvested for 16S rRNA seq and targeted metabolomics. How was the analysis carried out – inoculum vs. 24h slurries? If so, what happens to the inoculum not treated with antibiotics over 24h, and how representative is the microbiota (composition) after 24h in the incubator/fermenter compared to the native donor stool? How many anaerobes do you lose, and does this reflect the metabolite output of the microbiome as well? Did you add CO₂ to the culture atmosphere? What is the variance of the microbiome changes per donor in your assay assuming that there should be considerable variance between microbiome donors and – consecutively – strong variability in the antibiotic changes from minor changes to strong damages of the microbiome based on individual antibiotic resistance profiles? Please provide compositional plots of the individual probands along individual plots of alpha and beta diversity with control microbiotas incubated without antibiotics.
2. Ex vivo fecal culture experiments, *Enterococcus*: As *Enterococcus* is a highly prevalent, though low abundant commensal in healthy individuals, what about the expansion of enterococci (various spp.) when treated with antibiotics as some clones may exist in the healthy gut microbiome that harbor antibiotic resistance genes blooming after antibiotic expansion. Did you investigate whether the enterococci found in Fig. 1C-D are intrinsic to the subject or coming from the spiked ones. Line 167: What are the concentrations of antibiotics found in human feces? Growth of *E. faecalis* (Fig 1D) seems to differ profoundly between the antibiotic-pretreated microbiotas, can you do a statistic? What is the reason for that as the metabolic milieu shown in Fig 1B does not seem to be so different?
3. Mouse experiment: *E. faecalis* and *E. faecium* don't show a strong growth in Vanc-containing media compared to other antibiotics (Fig. S1). Are there any non-Vanc-resistant *Enterococcus* clones present in the *Enterococcus* colonies used, or what could be the reason for that and why did the authors use the "weakest antibiotic" for their mouse experiments? Compared to the human gut micromilieu (Fig. 1B vs. Fig 2), there seems to be inconsistencies between mouse and human Vanc effects. Was this correlated to any microbiome composition and differences between the mouse microbiota and human microbiota used? Please provide some data-based explanation and perhaps revise the conclusion drawn in line 209 as it reads too simplistic.
4. pH and metabolite experiments: Why do the authors use a low pH in their model apart from the fact that there is a range of pH values along the gut axis? Is it related to antibiotics and the depletion of commensals or the expansion of acid producing

pathobionts? How does the pH change after antibiotics in human slurries in vitro? What was the pH in the mouse model after Vanc administration? The OD values shown in Fig 3 and 4 for no metabolites group are very low for Enterococci, although the bacteria were cultured over night. Is the TS media or why do you see so little growth in your controls? Can you show the growth curves instead of ODs measured at an overnight time point?

Minor point:

- Fig. 7 and 8: Do the shaded bands show the predicted 95% confidence intervals?

Reviewer #4

(Remarks to the Author)

In this manuscript the authors investigate mechanisms associated with antibiotics-driven colonization of vancomycin resistant Enterococcus (VRE) in the intestine, which is a highly relevant clinical challenge. A main strength is the inclusion of both ex-vivo faecal culture experiments and investigations in mice. However, several aspects need to be clarified to determine the validity of the study and results. The conclusions drawn not be fully supported by the results partly due to the lack of suitable controls.

Results:

- First of all, the authors need to clarify if the metabolites are from relative or absolute quantification. This is not clear from the methods description, but will be crucial for the interpretations in Fig 1b and Fig 2. In relative quantification, the decrease of a group of metabolites will automatically result in increase of the remaining metabolites, although the absolute levels might be stable. The authors interpret the metabolic changes after antibiotics as increased or decreased, and these interpretations make up the basis for the subsequent experiments. Please clarify how the quantification of NMR metabolites was performed, and how absolute metabolite concentrations were achieved if this is the case.
- Figure 3 (A), Figure 4: The authors aim to test if metabolites can inhibit VRE growth. Use of the term "inhibit" indicates causality, and the authors need to clarify or hypothesize which biological mechanisms could explain that a metabolite directly inhibits survival of a bacteria. In fact, figure 3 clearly shows that low and average concentrations of a mix of metabolites leads to significantly increased growth of both *E. faecium* and *E. faecalis*. Even a high mix of metabolites lead to increased growth at pH 7, and it is only at pH 6 that the high concentrations lead to decreased growth. pH 6 is low at the pH spectrum of healthy feces. A condition with these high levels of several metabolites simultaneously may not be realistic or promote survival of any bacteria. This raises the question if a high concentration of a mixture of metabolites at low pH would lead to decreased growth of any bacteria, which should be tested using a control. Wouldn't any other bacteria, commensal or pathobionts, suffer from low growth under those conditions?
- For figure 4, use of a control species would be useful to investigate if the decreased survival is specific to the VRE or applies to bacteria in general. Furthermore, How did you decide on ranges for low, medium and high mixture concentrations? For example, when isovalerate inhibits bacterial growth with 128 mM, is this type of concentration found only at the tail-end of a population of healthy (or sick) individuals?
- Figure 5 – What is the main conclusion from this result? Obviously adding micronutrient to minimal media will boost growth of any other bacteria? Again, the use of a suitable control would be advisable.
- Figure 6 Did the antibiotic-naïve faecal microbiome possess the same "strain" of *E. faecium* you inoculated for the other conditions (spent supernatant)? Or are they *E. faecium* free?
What if you were to inoculate with any other bacteria a spent supernatant (generated as in "faeces supernatant"): would they not stop growing and show "suppression" due to lack of nutrients?
- Please clarify what the results in Fig 5 and 8 are normalized against; medium without the nutrients?
- The authors investigated preferred nutrients for VRE, with the claim that "This information could help inform the design of new microbiome therapeutics composed of gut commensals that can outcompete the VRE strains for use of these specific nutrients". However, as shown in Fig 7-9 the VRE can utilize several very common carbon and nitrogen sources, which are also utilized by many other strains, and it would be crucial to investigate how therapeutic depletion of these nutrient sources would affect the remaining gut microbiota.
- Figure 10 a – What makes you conclude that there is a low inhibition? How could you co-culture of *E. faecium* and *E. faecalis* together and be able to separately assess their CFU? Please explain the methodology used to differentiate this.
- In this study, strains with vanA-type glycopeptide resistance gene is utilized. Do the authors expect similar results in VRE strains with other resistance genes?

Discussion:

- Overall, the discussion is long and repetitive and could benefit from more concise language avoiding repetitions. It also lacks connections to the larger context of other antibiotic-resistant pathobionts.
- Please make sure to differentiate between colonization and expansion of pathobionts that already exist in the microbiome
- Faecal metabolites are produced by a combination of interactions within the microbiome, but also between the microbiome and the human host. The connection to the human host should be elaborated.
- L .631 –Wouldn't a bloom be associated with VRE > 10% of gut microbiome composition, as in *C. difficile* infections? The experiments shows log CFU/mL of 8 to 9 for VRE. Which is way lower than the 12 that can be found in a healthy gut microbiome (colon). If we unduly compare, it would be a very tiny portion of a microbiome composition, perhaps <0.1%.

Introduction:

The manuscript would benefit from additional introduction on the following topics:

- Since the aim is to understand mechanisms for resistance, it Who are the Enterococcus genus, and what vancomycin antibiotics do on a molecular level?
- Please describe why E. faecalis is relevant. E. faecalis is not mentioned in the introduction and only in passing in most of the paper.
- Are E. faecalis and E. faecium always present in the human gut, or is a de-novo colonization necessary to create VRE infection?

Methods:

- Add cat number and company that supplied the materials. Some are missing.
- Line 687 – Please, briefly mention how you assessed the MIC.
- Line 694 – What is “distal gut growth medium”? Formulation? Or company ?
- Line 698 – How was “VRE growth” was assessed? OD or CFU?
- Line 715 – What is a “faecal slurry”? A mixture of liquified feces and PBS?
- Line 773 – Please add which is elongation, annealing, etc.
- Line 845 to 847 – How do you distinguish and assess E. faecalis?
- Line 966 – Please explain what Credible intervals are?
- Line 985 to 989– How did you quantify pathogen growth?

Reviewer #5

(Remarks to the Author)

Version 1:

Reviewer comments:

Reviewer #1

(Remarks to the Author)

The authors have thoughtfully addressed the issues raised in my primary review.

Reviewer #2

(Remarks to the Author)

Thank you for taking my comments into account and for thoughtfully incorporating them into the revised manuscript. I am satisfied with the changes made and your answers to my questions. I agree with the adjustments that were made and I have no further requests.

Reviewer #3

(Remarks to the Author)

All the points I raised were answered sufficiently.

Reviewer #4

(Remarks to the Author)

The authors have to a large degree given thorough answers to answer the raised concerns, which have improved and clarified the manuscript. However, a few aspects remain:

- Regarding comment 1: Yes, NMR spectroscopy provides absolute quantification when using a reference signal (usually TSP or DSS or ERETIC signal). Please clarify in the manuscript which reference has been used.

- Regarding comment 4: I remain skeptical that the concentration used and shown is representative of a healthy colon. In fact, a previous study (Higher total faecal short-chain fatty acid concentrations correlate with increasing proportions of butyrate and decreasing proportions of branched-chain fatty acids across multiple human studies; PMID: 39295782) show that the concentrations used in this draft and the table in the rebuttal, are borderline of the «healthy» spectrum. As stated by the author, it is «not unreasonable to find these metabolite concentrations». That is true, but being “not unreasonable” does not mean such concentrations are representative of the average gut environment; they seem to be a rather extreme case. I recommend the authors to discuss clearly this limitation. Lastly, I think the additional figure R6 should indeed be added to paper. It is an important control and reference necessary to properly interpret the data of Fig 4.

- Regarding comment 10: Please explain what “the subtraction method” is? It remains unclear how results in Fig 5 and 8 are normalized.

- Regarding comment 12: It is well explained in the rebuttal. Please, state more explicitly in the results and methods, shortly, how you were able to select *E. faecium* combining vancomycin-resistant and Brilliance VRE plates.
- Regarding comment 24: Please describe exactly what was quantified by plating on Columbia agar? This remains unclear.

Reviewer #5

(Remarks to the Author)

RESPONSE TO REVIEWER COMMENTS

We would like to thank the reviewers for giving their time and effort to review our manuscript, and for providing their valuable comments and expertise. Below we have outlined the improvements we made to our manuscript based on the reviewer comments. We have numbered the comments from each reviewer to make it easier to refer to specific reviewer comments. We have referred to Figures and Tables as R1, R2, etc if we included them in the response to the reviewer comments but not in the main manuscript file or supplementary information file.

REVIEWER #1 (REMARKS TO THE AUTHOR):

Reviewer 1 Comment 1:

King and colleagues have investigated expansion of vancomycin-resistant *Enterococcus faecium* and *E. faecalis* in antibiotic-treated fecal cultures and antibiotic-treated mice. They demonstrate that antibiotics reduce the density of commensal bacterial species and microbial metabolite concentrations while increasing nutrient concentrations. Antibiotic treatment also leads to increased expansion of VRE. The authors extend these studies by growing VRE in trypticase soy broth supplemented with various concentrations of metabolites that were reduced in feces by antibiotic treatment and demonstrate that at pH 6 they inhibit VRE growth. Interestingly, upon testing with individual metabolites, the authors discovered differences in sensitivity to butyrate, 5-aminovalerate and ethanol between *E. faecium* and *E. faecalis*. The authors investigate the role of different carbon and nitrogen sources on growth of VRE and demonstrate the broad range of substrates they can use. Here again they demonstrate differences between *E. faecium* and *E. faecalis* that are potentially interesting. The kinetics of nutrient depletion by VRE are investigated and demonstrate some interesting differences between *E. faecium* and *E. faecalis*, particularly in depletion of different sugars. The final figure demonstrates some competition between *E. faecium* and *E. faecalis* but minimal competition between VRE and different Gram-negative organisms.

Author response:

Thank you for this summary of our manuscript.

Reviewer 1 Comment 2:

VRE infection is an important clinical problem and clinical and experimental studies have demonstrated the dramatic expansion of VRE in the intestinal lumen of humans and antibiotic-treated mice. The authors have generated some interesting but preliminary and incompletely investigated findings. The authors exaggerate the paucity of information on mechanisms of colonization resistance against VRE (lines 58-9) and should mention the role of the microbiota in stimulating host secretion of antimicrobial C-type lectins to inhibit VRE in the small intestine and production of VRE-inhibitory bacteriocins by commensal bacterial strains. With respect to inhibition of VRE by microbial metabolites, a recent manuscript demonstrated that metabolites produced by *Bifidobacterium* species and media acidification can inhibit VRE growth and may account for reduced infections in patients with liver disease (PMID: 37845315). Nevertheless, the role of nutrient depletion has not been adequately investigated and this manuscript makes important progress in that direction.

Author response:

We thank the reviewer for highlighting the novelty of our findings for the role of nutrient depletion by the gut microbiota to restrict VRE growth. We also thank the reviewer for highlighting these papers for us, and we have added the following text to our introduction (see lines 69-79):

“Currently, we have an incomplete understanding of the mechanisms of colonisation resistance that protect the host against VRE intestinal colonisation. However, a few studies have provided some insights. Previous work demonstrated that antibiotic treatment downregulated the expression of RegIIIy, a C-type lectin that kills VRE in the small intestine¹⁷. Another study demonstrated that a gut commensal strain of *Blautia producta* can produce a lantibiotic that can reduce VRE growth¹⁸. Finally, another study

demonstrated that lactulose treatment in patients with severe liver disease increased *Bifidobacterium* growth, increased the production of microbial metabolites, acidified the gut lumen, and reduced the incidence of systemic infections and mortality¹⁹. However, we lack information on how broad-spectrum antibiotics that promote VRE intestinal colonisation impact nutrient competition and inhibitory metabolite production by the gut microbiome, and how this in turn impacts VRE growth.”

Reviewer 1 Comment 3:

The results, however, provide a very limited perspective and do not address the fact that VRE colonizes the entire intestine, and thus nutrients that are available in the small intestine likely play a significant role in expansion and domination.

Author response:

Faecal samples are more commonly used than small intestine samples to investigate VRE intestinal colonisation in both humans and mice¹⁻⁷. However, a previous study demonstrated that mice treated with a combination of the broad-spectrum antibiotics metronidazole, neomycin and vancomycin had higher VRE counts in their colons ($\sim 10^7$ - 10^8 CFU/segment or $\sim 10^6$ - 10^7 CFU/cm intestine) as compared to their small intestines ($\sim 10^5$ CFU/segment or $\sim 10^3$ CFU/cm intestine)^{8,9}. Another study showed that mice treated with clindamycin had higher indigenous enterococci counts in their colon content ($\sim 10^{10}$ - 10^{11} CFU/g) compared to their small intestine content ($\sim 10^8$ CFU/g)¹⁰. These findings, combined with the relative ease at collecting fresh faeces from healthy human donors as compared to fresh small intestine contents is why we chose to investigate the faecal microbiome instead of the small intestine microbiome in this study.

However, we agree that it would be interesting to investigate how nutrient availability and metabolite production may vary at different intestinal sites during antibiotic treatment, and as such we have added this text to our discussion (see lines 734-742):

“Faecal samples are more commonly used than small intestine samples to investigate VRE intestinal colonisation in both humans and mice, and as such this study investigated VRE intestinal colonisation in the context of the faecal microbiome, which is more reflective of the large intestine microbiome than other regions of the intestine^{12-16,50,51}. VRE colonises the antibiotic-treated large intestine to higher concentrations, however VRE can also colonise the antibiotic-treated small intestine¹⁷. It would also be important to investigate nutrient availability and microbial metabolite production in other regions of the intestine that VRE can colonise (such as the small intestine) to design microbiome therapeutics that can restrict VRE growth at all intestinal regions.”

Major Issues:

Reviewer 1 Comment 4:

There are several surprising results shown in Figure 1A. First, clindamycin has broad activity against obligate anaerobes but seems to only reduce the density of Bifidobacteriaceae. Is there an explanation for this result?

Author response:

As shown in **Fig. 1a** at the family level, CLI decreased the abundance of *Bifidobacteriaceae* and *Ruminococcaceae* (not just *Bifidobacteriaceae*). As shown in our response to Reviewer 3 Comment 4 and Reviewer 3 Comment 5, there is some heterogeneity in the response of different donors to CLI, which may be why some of these taxa were not statistically significant. This may be due to differences in the types and abundance of antibiotic resistance genes present in the gut microbiome of these donors, however we did not measure this in our study.

To demonstrate that CLI does have broad activity against obligate anaerobes we have also plotted the changes with each antibiotic at the phylum, class, and order level (**Fig. S4**). In these plots we can see CLI significantly reduced Actinobacteriota, Bacteroidota, and Firmicutes at the phylum level, Actinobacteria, Bacteroidia, Clostridia, and Negativicutes at the class level, and Bifidobacteriales, Lachnospirales, and

Oscillospirales at the order level. We have added this figure to the Supplementary Information file and added the following text to the results (see lines 167-168):

“Antibiotic-treated faecal cultures also had significant decreases in taxa at the phylum, class, and order levels (**Fig. S4**).”

Figure S4: Antibiotic treatment significantly reduced the abundance of gut commensals at the phylum, class, and order levels. Heatmaps illustrating the log₂ fold change (antibiotic-treated relative to water-treated) in bacterial taxa that were significantly decreased (shown in blue) or increased (shown in red) in faecal cultures following treatment with antibiotics that promote VRE intestinal colonisation. Taxa that were not significantly changed with antibiotic treatment were not plotted (shown in white). Changes are plotted at **(a)** phylum level, **(b)** class level, and **(c)** order level. n = 12 healthy human faecal donors, Wilcoxon signed rank test (two-sided) of log-transformed abundances with Benjamini Hochberg false discovery rate (FDR) correction, p < 0.05. MTZ = metronidazole, CLI = clindamycin, VAN = vancomycin, CRO = ceftriaxone, TZP = piperacillin/tazobactam.

Reviewer 1 Comment 5:

Second, almost all human fecal samples have endogenous Enterococci that would not be sensitive to, for example, metronidazole or ceftriaxone. It is surprising therefore, that endogenous Enterococci did not expand in fecal cultures. It raises the possibility that VRE strains may differ in some way from endogenous, commensal Enterococcal species/strains.

Author response:

This is a good question, and we were also interested in this as well. Endogenous *Enterococcus* was only detected in the faeces of 1 of the 12 healthy faecal donors. **Table R1** below shows the relative abundances of endogenous *Enterococcus* found in the faecal culture samples. As shown in this table, *Enterococcus* was only detected in 2 of the 12 donors in the water-treated control group, and therefore we were not able to measure statistically significant changes in *Enterococcus* with antibiotic treatment.

Table R1: Relative abundance (%) of *Enterococcus* found in 16S rRNA gene sequencing data from the ex vivo human faecal cultures. Values highlighted in red show detectable levels of *Enterococcus* (greater than 0%).

Donor	H ₂ O-treated	MTZ-treated	CLI-treated	VAN-treated	CRO-treated	TZP-treated
1	0.00	3.11	0.00	0.00	0.00	0.00
2	0.00	0.00	0.00	0.00	0.00	0.00
3	0.00	0.00	0.00	0.00	0.00	0.00
4	0.00	0.00	0.00	0.00	0.00	0.00
5	0.00	0.00	0.00	0.00	0.00	0.00
6	0.00	0.00	0.00	0.00	0.00	0.00
7	0.00	0.00	0.00	0.00	0.00	0.00
8	0.00	0.00	0.00	0.00	0.00	0.00
9	0.00	0.00	0.00	0.00	0.00	0.00
10	0.12	0.34	0.01	0.00	0.10	1.21
11	0.11	1.18	0.00	0.00	0.08	0.05
12	0.00	0.00	0.00	0.00	0.00	0.00

Reviewer 1 Comment 6:

An additional issue with Figure 1, panels A and B, is that quantitation is only relative but not absolute. Thus, for bacterial strains or metabolites that are present at very low densities/concentrations, slight increases can look dramatic but still fall below a range that is biologically important.

Author response:

The data presented in **Figs. 1a and 1b** are absolute abundance data but are plotted as log₂FCs relative to the water-treated control as this was the most feasible way to show these large datasets in one figure. Showing this data as log₂FC allows readers to quickly and easily see what was decreased or increased with antibiotic treatment. However, we have also shown the individual 16S rRNA gene sequencing profiles as absolute abundances in **Fig. S2** in response to Reviewer 3 Comment 5.

In terms of whether these changes are biologically important, it is important to remember that low abundance bacteria can play a crucial role in the functioning of a healthy gut microbiome. For example, in the colon almost 100% of the bile acids are derived from microbial metabolism, however only a few gut commensals can perform 7 α -dehydroxylation, where their abundance composes less than 0.025% of the total gut microbiota and less than 0.0001% of the total colonic microbiota¹¹.

The purpose of showing the data in **Figs. 1c and 1d** was to demonstrate that VRE could grow in the nutrient-enriched and metabolite-depleted environment of the antibiotic-treated faecal cultures, but could not grow in the nutrient-depleted and metabolite-enriched environment of the water-treated faecal cultures. Indeed, these

experiments showed that VRE growth was restricted in water-treated faecal cultures and significantly promoted in antibiotic-treated faecal cultures.

However, to demonstrate that the nutrients had increased to sufficient concentrations to support VRE growth, we measured the median concentration of each nutrient that was found in the VAN-treated faecal cultures, and then supplemented these nutrients into a minimal medium at these concentrations as sole carbon and nitrogen sources (“Mock VAN Nutrient Mix”). As a positive control we supplemented the nutrients into the minimal medium at an equal concentration of each nutrient (“Nutrient Mix” control, as we did for the data presented in Fig. 5) or we did not supplement in any nutrients (“No Nutrients” control). The same *E. faecium* and *E. faecalis* strains tested in Figs. 1c and 1d were inoculated into the minimal medium and the cultures were incubated anaerobically overnight. As shown in Fig. R1 below, the Mock VAN Nutrient Mix and the Nutrient Mix both promoted the growth of each VRE strain to levels that we expect these strains to grow to under anaerobic conditions.

Figure R1: A mixture of nutrients mimicking nutrient concentrations found in VAN-treated faecal cultures promoted VRE growth. Vancomycin-resistant *E. faecium* (NCTC 12202) and vancomycin-resistant *E. faecalis* (NCTC 12201) were cultured in minimal medium supplemented with a mixture of nutrients at equal concentrations (“Nutrient Mix”) or at concentrations mimicking the concentrations found in VAN-treated faecal cultures (“Mock VAN Nutrient Mix”), under anaerobic conditions. n = 15 replicates in 3 independent experiments. ANOVA with Dunnett’s multiple comparison test comparing the nutrient supplemented groups to the no nutrients control. **** = P ≤ 0.0001. Data are presented as mean values ± SD. OD₆₀₀ = optical density at 600 nm.

Reviewer 1 Comment 7:

Throughout the manuscript, the authors describe differences in sensitivity of *E. faecium* and *E. faecalis* in terms of sensitivity to butyrate or dependence on different amino acids. These differences are very interesting and may provide insights into shifting prevalences of these two species in clinical settings over time and regionally. That said, the findings presented in this manuscript only focus on one strain of each species. Given the remarkable strain-to-strain variation that can occur within bacterial species, the authors should confirm these results with panels of *E. faecium* and *E. faecalis* strains.

Author response:

We agree that adding analysis of additional strains to this study would improve our manuscript. As such, we have repeated the individual metabolite inhibition assays (Figs. 3, S9-S14), mixed nutrient utilisation assays (Fig. 5), individual carbon assays (Figs. 7, S15-S18), and individual nitrogen utilisation assays (Figs. 8, S19,

S20) using additional vancomycin-resistant *E. faecium* and *E. faecalis* strains. Our manuscript now includes three vancomycin-resistant *E. faecium* strains and three vancomycin-resistant *E. faecalis* strains for each of these assays. We have updated the Results and Discussion sections accordingly. Overall, these new strains supported the results presented in the previous version of our manuscript: monosaccharides and disaccharides supported VRE growth as carbon sources, amino acids supported VRE growth as nitrogen sources, and short chain fatty acids (SCFAs) inhibited VRE growth. Growth of each species was generally supported by the same nutrients and inhibited by the same metabolites, but there were some differences. For example, two *E. faecium* strains (NCTC 12204 and DSM 25698) were inhibited by acetate at average and high concentrations found in healthy human faeces, while one *E. faecium* strain (NCTC 12202) was resistant to acetate at these same concentrations.

Reviewer 1 Comment 8:

In lines 263-268, the authors point out that metabolite concentrations measured in feces do not fully inhibit VRE. Many microbial metabolites, butyrate in particular, are metabolized by intestinal epithelial cells and thus concentrations vary along the length of the intestine. The highest concentrations of butyrate are detected in the cecum and proximal large intestine of mice and likely also humans.

Author response:

This is a good point. We tested a large range of metabolite concentrations in this study, however when discussing our results we only related them to the faecal concentrations. A previous study by Cummings *et al.* (1987) measured the concentrations of metabolites (including SCFAs and branched chain fatty acids (BCFAs)) in different regions of the small and large intestine from 4-6 sudden death victims¹². SCFA and BCFA concentrations were low in the jejunum and ileum but increased by over 10-fold in the caecum. Concentrations then progressively decreased in the ascending, transverse, and descending colon, with a small increase in the sigmoid/rectum. We have revised our discussion to add in the following text (see lines 592-599):

“We investigated the ability of each metabolite to individually suppress VRE growth at physiologically relevant concentrations, where we tested each metabolite at their low, average, and high concentrations measured in healthy human faeces. However, metabolite concentrations vary along the length of the gastrointestinal tract. SCFA and branched chain fatty acid concentrations are high in the caecum and ascending colon and progressively decrease in the transverse and descending colon, with a small increase in the sigmoid/rectum³⁵. Therefore, microbial metabolites may provide greater suppression of VRE growth in different regions of the intestine.”

We have also updated the summary from lines 263-268 in our results section upon testing of additional VRE strains (in response to Reviewer 1 Comment 7) and with additional data testing a mixture of acetate, propionate, butyrate, and valerate and a mixture of propionate, butyrate, and valerate (in response to Reviewer 4 Comment 3). It now reads as follows (see lines 306-310):

“Overall, we observed significant but incomplete suppression of vancomycin-resistant *E. faecium* and *E. faecalis* growth by acetate, propionate, butyrate, and valerate (at high concentrations) when tested individually at pH 6. However, the APBV and PBV mixtures provided complete or near complete suppression of *E. faecium* and *E. faecalis* growth at pH 6.”

Reviewer 1 Comment 9:

In Figure 7, did *E. faecium* not grow on sucrose, or was it not tested?

Author response:

E. faecium (NCTC 12202) did not grow on sucrose in the individual carbon assays, and so we showed this data in a supplementary figure in the previous version of our manuscript. Our intention was to only show the nutrients that were utilised in **Figs. 7 and 8** to keep the plots less cluttered. However, to avoid confusion we have updated **Fig. 7** to show all the monosaccharides and disaccharides, and we have updated **Fig. 8** to show all the amino acid leave one out mixtures.

Reviewer 1 Comment 10:

In figures 7 and 8, the authors show interesting results for VRE growth in minimal media in the presence of different carbon and nitrogen sources. These results are interesting and raise many questions but the authors do not provide any new insights. At a minimum, the authors should correlate genomic differences between *E. faecium* and *E. faecalis* with differences in carbohydrate and amino acid metabolism. It would be interesting to investigate VRE transcriptomes, for example, in these different media. These types of analyses would enable the investigators to potentially identify genes and operons involved in carbohydrate and amino acid metabolism.

Author response:

The aims of this manuscript were as follows:

- (1) to measure the nutrient and metabolite landscapes encountered by VRE in a gut microbiome treated with multiple different antibiotics that are known to promote VRE intestinal colonisation.
- (2) to define the niche that VRE occupied in an antibiotic-treated gut microbiome, how distinct these niches were between *E. faecium* and *E. faecalis*, and how different these niches were compared to niches occupied by other multidrug-resistant pathogens.
- (3) to inform the rational design of a new microbiome therapeutic that can occupy the same niche as VRE to displace them from the intestine.

The purpose of the experiments used to generate the data in **Figs. 7 and 8** was to determine which nutrients (that were increased in an antibiotic-treated gut microbiome) could be used as carbon or nitrogen sources to support vancomycin-resistant *E. faecium* or *E. faecalis* growth. Measurement of bacterial growth in minimal media supplemented with defined carbon and nitrogen sources is the standard established method used for this purpose. This data is crucial to the design of new microbiome therapeutics that take advantage of restoring nutrient depletion to restrict VRE growth. We also needed to collect this information to understand how the nutrient-defined niches occupied by VRE varied from the niches occupied by other multidrug-resistant pathogens such as CRE. Therefore, the data outlined in **Figs. 7 and 8** align with aims 2 and 3.

Identification of genes or operons involved in carbohydrate and amino acid metabolism does not align with the aims of this manuscript. We have already demonstrated that these nutrients can support the growth of VRE strains, and identifying the specific genes involved in this process does not change the main findings of this paper. Moreover, mapping the genes and operons involved in the metabolism of this large number of nutrients (including validation of correlations) would be a very large undertaking, and would make up its own separate manuscript.

Reviewer 1 Comment 11:

The results shown in Figure 9 are interesting but inadequately interpreted by the authors. The three timepoints they investigate focus on log phase (4.5 hours), early stationary (9 hours) and late stationary phase (24 hours). Thus, between 9 and 24 hours, expansion has ceased. The authors should discuss these different scenarios and perhaps consider that the shifts they detect may not result from depletion of preferred substrates but from dramatic changes in metabolism that occur when microbes progress from exponential growth to the stationary state.

Author response:

Preferred nutrients are nutrients that are utilised before other available nutrients, which we can see by the 4.5 hour and 9 hour time points. We could have stopped collecting samples at 9 hours, but then we would have missed data about other nutrients that could be used once the preferred nutrients were depleted. Although expansion had ceased during stationary phase, during this phase the growth rate is equal to the death rate. As such, cells can still grow and use nutrients during this phase, which we measured at the 24 hour time point.

There are few studies investigating nutrient preference, however a previous study investigating nutrient preference in *E. coli* EDL933 and *E. coli* MG1655 included final timepoints at ~37.5 hours and 25 hours respectively¹³, and our previous study investigating nutrient preference in carbapenem-resistant *Enterobacteriaceae* also included a final timepoint at 24 hours¹⁴.

Minor issues:

Reviewer 1 Comment 12:

Line 187 should state that oral vancomycin was used.

Author response:

We have made this edit (see lines 214-216):

“Mice were treated with oral vancomycin as it has been previously demonstrated to promote VRE intestinal colonisation in both humans and mice^{12,13,15,16}.”

Reviewer 1 Comment 13:

Figures 7 and 8 are difficult to interpret. Do the authors have CFU data to complement the OD values?

Author response:

No, we do not have CFU data for these experiments as OD₆₀₀ measurements are routinely used for bacterial growth curves. AMiGA infers the underlying growth curves to generate models with 95% credible intervals, where growth of the no carbon source control or the no nitrogen source control was subtracted. To interpret these plots more simply, these plots are essentially averaging the growth curves from multiple replicates and providing 95% confidence intervals and subtracting the no carbon or no nitrogen control.

To illustrate this, we show below an example of a set of growth curves for *E. faecium* grown on glucose as the sole carbon source as compared to the no carbon source control (under aerobic conditions). **Fig. R2** shows the raw growth curve data and the corresponding plot generated by AMiGA.

REVIEWER #2 (REMARKS TO THE AUTHOR):

Reviewer 2 Comment 1:

This manuscript makes an important contribution to research in the field of antibiotic resistance, a global health challenge that is of utmost importance.

Author response:

Thank you.

Reviewer 2 Comment 2:

The title of the manuscript is: “Vancomycin-resistant Enterococcus colonise the antibiotic-treated intestine by occupying distinct nutrient- and metabolite-defined intestinal niches”.

First, I would suggest using the name ‘Vancomycin-resistant enterococci’ as it is used as a plural in the title, but also throughout the entire manuscript. Therefore, the abbreviation VRE stands for ‘Vancomycin-resistant enterococci’, and needs to be adapted in the manuscript (line 21 & 43).

The research shows that VRE occupy distinct but overlapping intestinal niches. However, in the title, it is stated that VRE only occupy distinct niches. Not sure if the title needs to be adjusted, but there is a discrepancy between the title and the message of the article.

Author response:

Based on the feedback from the reviewer we have revised the manuscript title as follows:

“Vancomycin-resistant enterococci colonise the antibiotic-treated intestine by occupying overlapping but distinct nutrient- and metabolite-defined intestinal niches”

We have also revised the abbreviation for VRE to read as “vancomycin-resistant enterococci” in the abstract (lines 20-21) and the introduction (line 43).

Reviewer 2 Comment 3:

The mouse experiment only confirms the overall findings of the *ex vivo* experiment, but how do we translate the findings to humans? Mice consume a different diet; respond differently to antibiotics than humans, and have a much faster metabolism leading to different nutrient and metabolic phenotypes. Can some interpretation be added to the discussion?

Author response:

The purpose of the mouse experiment was to confirm that our *ex vivo* results held up in an *in vivo* model. Like all experimental models, mice have their limitations, such as differences between the human and mouse gut microbiome, and differences between human and mouse diets¹⁵. However, it is not possible to do this type of experiment in humans due to ethical limitations, and as such mice are often used in pathogen gut colonisation studies^{3,6,7,14,16,17}. Like humans, mice with a healthy gut microbiome have intact colonisation resistance, where mice are resistant to VRE intestinal colonisation. Moreover, like humans, mice treated with antibiotics also get disrupted colonisation resistance, where they have high levels of VRE intestinal colonisation. Therefore, we wanted to show that antibiotic treatment in mice also promoted VRE growth in an intestinal environment that was enriched in nutrients (such as sugars and amino acids) and depleted in metabolites (such as SCFAs and BCFAs).

Due to differences between humans and mice we did not expect to see the exact same changes in nutrients and metabolites. Rather, we expected to see the same overall trend of an enrichment of nutrients that support VRE growth, and a depletion of metabolites that can inhibit VRE growth. As shown in **Fig. 2**, we showed that an antibiotic-treated mouse faecal microbiome that supported high levels of vancomycin-resistant *E. faecium* growth had an increase in the concentrations of in sugars (arabinose, glucose, trehalose) and amino acids (arginine, aspartate, glycine, phenylalanine, tyrosine) and a decrease in the concentration of metabolites such as SCFAs (propionate, butyrate), isobutyrate, lactate, and ethanol. Despite differences in the gut microbiome and diet between humans and mice, our *in vivo* mouse results are supportive of our *ex vivo* human faecal culture results, which is a strength of our study.

We have added a discussion of the limitations of mouse models and an explanation of how the mouse data supports our human faecal culture results in the discussion section (see lines 749-763):

“In this study we used a mouse model to confirm that our *ex vivo* results held up in an *in vivo* intestinal environment. However, like all experimental models, mice have their limitations. For example, humans and mice have some differences in the bacterial species that compose their gut microbiome, and humans and mice consume different diets⁵⁵. However, it is not possible to perform pathogen intestinal colonisation experiments in humans due to ethical limitations, and as such mice are often used in these studies^{14-16,21,33,56}. Humans and mice with healthy gut microbiomes have intact colonisation resistance, where both humans and mice are resistant to VRE intestinal colonisation. Moreover, like humans, mice treated with broad spectrum antibiotics become highly susceptible to VRE intestinal colonisation, where VRE intestinal growth reaches high densities. In both our *ex vivo* human faecal culture experiments and our *in vivo* mouse experiments we showed that antibiotic treatment caused an enrichment of nutrients (such as sugars and amino acids) and a depletion of metabolites (such as SCFAs, isobutyrate, lactate, and ethanol). Therefore, despite differences in the gut microbiome composition and diets between humans and mice, our *in vivo* mouse results were supportive of our *ex vivo* human faecal culture results.”

Reviewer 2 Comment 4:

Stool samples of healthy individuals were examined. Do the authors have more information on the dietary pattern, use of medication, intake of pre-, pro-, and postbiotics, inflammatory parameters, serum SCFA, ...? Was a colonoscopy performed on these subjects to rule out any significant inflammation? As these factors can all influence faecal nutrient and microbial metabolite concentrations, this could influence the interpretation of the findings.

Author response:

We did not measure inflammatory parameters or serum SCFA concentrations and we did not perform a colonoscopy on these donors as these are not standard measurements to perform on healthy subjects in microbiome studies. We were not looking to recruit “super” healthy donors for this study, just “typical” healthy donors. “Typical” healthy donors are resistant to VRE intestinal colonisation and as such “super” healthy donors are not required. All recruited donors did not report any chronic health issues and did not have a personal or family history of gastrointestinal disease.

We have added the additional information about the health status of the donors to the methods section of our manuscript (lines 840-841):

“Faecal donors were aged between 18-65 years old, did not have a personal or family history of gastrointestinal disease, and did not have any chronic health issues.”

We have added the additional information about the dietary pattern, use of medication, intake of pre, pro- and post-biotics to the methods section of our manuscript (see lines 845-847):

“Ten donors consumed an omnivore diet, one donor consumed a vegetarian diet, and one donor consumed a mostly vegetarian diet. None of the donors used regular medication or consumed prebiotics, probiotics, or postbiotics.”

Reviewer 2 Comment 5:

Faeces of twelve healthy human donors was used in the experiments. Why is there a difference in the number of donors in Fig 1 (n = 5-8) and Fig 6 (n = 3), and which samples from which donors were selected for the experiments described in Fig 1 and Fig 6?

Author response:

Each of the faecal donors used in this manuscript are one of the 12 faecal donors that were used to generate the data in **Figs. 1a and 1b**.

We first performed the experiments outlined in **Figs. 1c and 1d** to confirm that VRE could grow in these antibiotic-treated faecal cultures and to also perform a power calculation to confirm how many donors were required for the experiments outlined in **Figs. 1a and 1b**. When writing the paper, we were concerned that showing the data outlined in **Figs. 1c and 1d** first might confuse readers (readers might have thought the data presented in **Figs. 1a and 1b** contained VRE, when they did not). Therefore, we showed the data presented in **Figs. 1c and 1d** after the data presented in **Figs. 1a and 1b**.

For the data presented in **Figs. 1c and 1d**, we were exploring which antibiotics we wanted to test based on our reading from the literature, so we started with VAN and CRO, then added MTZ and CLI as we tested additional donors. We didn't see any reason to exclude the extra replicates obtained from our earlier experiments, so the experiments performed for **Figs. 1c and 1d** used 5-8 donors. Power calculations (shown below in Reviewer 2 Comment 9) showed that 2-4 donors were sufficient for this experiment.

Using the data presented in **Figs. 1c and 1d** we performed power calculations to determine how many donors we should use to generate the data in **Figs. 1a and 1b** (see our response to Reviewer 2 Comment 9 below). These power calculations showed that we only needed 2-4 donors. We recruited the same donors used in **Figs. 1c and 1d**, plus additional donors for this experiment, as we had free access to ¹H-NMR spectroscopy

instrument time and we thought it would be beneficial to include as many donors as possible to capture potential differences in the responses of donors to the different antibiotic treatments.

The data presented in **Fig. 6** was again performed on different days than the experiments performed in **Fig. 1**, but these 3 donors were 3 of the donors that were used in the experiments shown in **Figs. 1a and 1b**, and in **Figs. 1c and 1d**. As our power calculations using the data outlined in **Figs. 1c and 1d** showed that we needed 2-4 healthy donors, we recruited 3 donors here as a starting point to see if we saw a significant difference (and we did see a significant difference).

Reviewer 2 Comment 6:

The gastrointestinal tract of hospitalized patients presents a markedly different environment compared to the healthy human gut, offering a unique metabolic landscape for intestinal microorganisms (Stellfox et al., 2022, <https://doi.org:10.1128/mbio.00670-22>). In this study, antibiotic-treated healthy human faeces were used to explore colonization resistance, nutrient availability, and the production of microbial metabolites. However, treatment needs to be established for hospitalized patients and not for healthy humans treated with antibiotics. Why didn't the researchers examine patients' stool samples?

Author response:

We believe the sentence the reviewer is referring to in the Stellfox et al. (2022) commentary paper reads as follows:

“The GI tract of the hospitalized patient differs significantly from the healthy human gut and provides a vastly different metabolic environment for intestinal flora.”

The authors did not provide a reference for this statement, however based on what was written earlier in the commentary we believe the authors were referring to the fact that hospitalised patients receive antibiotic treatment:

“A prime example is vancomycin-resistant *Enterococcus faecium* (VREfm), a hospital-associated pathogen that readily colonizes the antibiotic-perturbed GI tract and also causes deadly systemic infections. VREfm colonization and infection are facilitated by the fact that enterococci are often the “last bacteria standing” after other members of the GI flora are removed via broad-spectrum antibiotics and other treatments that deplete commensal flora, such as alterations in enteral nutrition, gastric acid suppression, skin and oropharyngeal decontamination, as well as mucosal and skin barrier disruption (3).”

The paper the authors referenced above does not refer to VRE but rather refers to dysbiosis in ICU patients. The main risk factor for VRE intestinal colonisation is treatment with broad spectrum antibiotics³⁻⁷, which we investigated in this study. Enteral nutrition and gastric acid suppression alone have not been shown to promote VRE intestinal colonisation^{18,19}. Skin and oropharyngeal decontamination and mucosal and skin barrier disruption are not relevant to this study.

Investigating the antibiotic treatment of healthy donors allows us to eliminate many confounding variables that are associated with hospitalised patients, such as comorbidities, medication use, time since previous antibiotic treatment, administration of multiple antibiotics simultaneously, etc. Using healthy donors also allowed us to compare multiple different antibiotics to a water-treated control for the same gut microbiome. It would not be feasible to collect healthy (non-antibiotic-treated) samples from hospitalised patients, and so the healthy microbiome samples and antibiotic-treated samples would come from different donors, which would likely be confounded by differences in the diets of different donors.

Reviewer 2 Comment 7:

Genomic studies have shown that hospital-associated *E. faecium* strains have acquired specific genes enabling them to access and utilize nutrient sources unavailable to commensal strains, such as amino-sugars found on cell surfaces and in mucin. This metabolic shift distinguishes them from

community-associated strains, which typically utilize complex carbohydrates (Lebreton et al., 2013, doi <https://doi.org:10.1128/mBio.00534-13>; Cattoir 2022, <https://doi.org:10.1016/j.mib.2021.10.013>).

Author response:

We thank the reviewer for bringing this paper to our attention, and we have added a brief mention of these papers to our discussion (see lines 636-642):

“A previous study by Lebreton and colleagues found that the hospital-associated *E. faecium* lineage lost pathways related to complex carbohydrate utilisation (found in the community-associated *E. faecium* lineage), which were replaced by pathways on mobile elements that were related to the utilisation of amino sugars⁴⁷. This could be advantageous for hospital-associated *E. faecium* strains in an antibiotic-treated intestine, which we demonstrated had increased concentrations of sugars (including increased concentrations of N-acetylglucosamine) that VRE could utilise.”

Reviewer 2 Comment 8:

Why did the researchers choose to use ¹H-NMR spectroscopy and not untargeted LC/GC-MS which can cover a wider range of metabolites? Would MS analyses provide added value?

Author response:

¹H-NMR spectroscopy is commonly used in microbiome studies and has many strengths that justifies its use. For example, ¹H-NMR spectroscopy is an untargeted and unbiased metabolomics approach that allows us to capture a broad range of nutrients and metabolites in one measurement²⁰. It is highly reproducible and provides absolute quantification of all metabolites in the spectrum^{20,21}. It is also the gold standard technique used to identify unknown metabolites^{20,21}.

In addition, one of the aims of this study was to determine how distinct the intestinal niches occupied by VRE were compared to the niches occupied by carbapenem-resistant *Enterobacteriaceae* (CRE). We previously defined the intestinal niches that CRE occupied using ¹H-NMR spectroscopy, and as such using the same technique in this paper makes this comparison more straightforward¹⁴.

Reviewer 2 Comment 9:

How can the authors guarantee that this study has sufficient statistical power? Were power calculations performed? If so, please add this information to the manuscript.

Author response:

Yes, we performed power calculations to determine how many donors to test in this study. We have shown the power calculations below.

Experiments performed for data shown in Figs. 1c and 1d:

To give us an estimate of how many donors we would need for our faecal cultures spiked with vancomycin-resistant *E. faecium* or *E. faecalis* we used data from faecal culture experiments we performed in a previous publication that also tested TZP, one of our antibiotics of interest¹⁴. In this experiment we tested TZP-treated vs H₂O-treated faecal cultures spiked with carbapenem-resistant *E. coli* (two strains tested) or carbapenem-resistant *K. pneumoniae* (three strains tested). Using the log transformed plate counts for each carbapenem-resistant *E. coli* or *K. pneumoniae* strain from TZP-treated vs water-treated faecal cultures, we calculated effect sizes ranging from 2.82-13.55. Conducting a power analysis (using G*Power software, v3.1.9.7) for a two-tailed t-test with this effect size, an alpha of 0.05, and a power of 0.80 revealed a sample size of 2-4 donors were required per experimental group. This is consistent with power calculations performed using vancomycin-treated mice (as described below in the power calculation for **Fig. 2**) which demonstrated that a sample size of 2 was required. Therefore, the 5-8 donors used for this experiment was above the minimum number of donors that we calculated would be required.

Experiments performed for data shown in Figs. 1a and 1b:

We performed a power calculation using the data from **Figs. 1c and 1d** for MTZ, CLI, VAN, and CRO. Using the log transformed plate counts for vancomycin-resistant *E. faecium* from MTZ-treated, CLI-treated, VAN-treated, or CRO-treated faecal cultures vs H₂O-treated faecal cultures, we calculated effect sizes ranging from 4.42-7.37. Conducting a power analysis (using G*Power software, v3.1.9.7) for a two-tailed t-test with this effect size, an alpha of 0.05, and a power of 0.80 revealed a sample size of 3 donors were required per experimental group.

These calculations were repeated using the log transformed plate counts for vancomycin-resistant *E. faecalis* from MTZ-treated, CLI-treated, VAN-treated, or CRO-treated faecal cultures vs water-treated faecal cultures, we calculated effect sizes ranging from 2.84-15.90. Conducting a power analysis (using G*Power software, v3.1.9.7) for a two-tailed t-test with this effect size, an alpha of 0.05, and a power of 0.80 revealed a sample size of 2-4 donors were required per experimental group.

Figs. 1c and 1d did not contain data for TZP as the *E. faecium* and *E. faecalis* strains tested were sensitive to TZP at concentrations found in faeces. However, to give us an estimate for numbers of donors needed for the experiments shown in **Figs. 1a and 1b** we used data from similar experiments we performed in our previous publication¹⁴ (as described above in the power calculation for **Figs. 1c and 1d**), which revealed a sample size of 2-4 donors were required per experimental group.

We used 12 faecal donors for the experiments outlined in **Figs. 1a and 1b**, which is well above the minimum number of donors that we calculated would be required.

Experiments performed for data shown in Fig. 2:

We performed a power calculation using the data from Isaac et al. (2022) where mice were untreated or treated with vancomycin and colonised with vancomycin-resistant *E. faecium*³. Using the log transformed plate counts for *E. faecium* from vancomycin-treated or antibiotic-naïve mouse faeces, we calculated an effect size of 28.90. The maximum effect size the software allows is 20, and as such we used an effect size of 20 for this calculation. Conducting a power analysis (using G*Power software, v3.1.9.7) for a two-tailed Wilcoxon-Mann-Whitney test with an effect size of 20, an alpha of 0.05, and a power of 0.80 revealed a sample size of 2 mice are required per experimental group. Therefore, we used 5 mice per group as this is above the sample number calculated and is also a typical number of mice per group used in similar studies³.

To the best of our knowledge it is not common practice to include power calculations in the manuscript, so we have not included them in this revised version of the manuscript. However, we can include them if requested.

Reviewer 2 Comment 10:

Fig 6: the preferred approach when using non-parametric tests, including the Wilcoxon signed-rank test or Kruskal-Wallis test, is to report medians along with interquartile ranges (IQR), as these better reflect the central tendency and variability in non-normally distributed data. If mean ± SD is reported alongside a Wilcoxon signed-rank test, the authors should clarify why they chose this presentation (e.g., for comparability to other studies or because the data are close to normally distributed

Author response:

Thank you for bringing this to our attention. We have replotted all non-parametric data plots to show medians along with interquartile ranges.

Reviewer 2 Comment 11:

The authors mention several times that novel microbiome therapeutics can be used to re-establish colonization resistance and restrict VRE intestinal growth, and that gut commensals can be used for this purpose. What specific therapies are being considered and can this be discussed more in depth? In what way can the current results be implemented in the clinical field?

Author response:

A novel microbiome therapeutic could be composed of a cocktail of live gut commensals that can together deplete nutrients that were enriched with antibiotic treatment and also restore the production of microbial metabolites that were decreased with antibiotic treatment. Another option could be a microbial metabolite or mixture of microbial metabolites that can restrict VRE growth, such as a mixture of propionate, butyrate, and valerate tested in this study. Finally, these two new therapeutics could also be combined, such that we administer a cocktail of live gut commensals and the mixture of metabolites simultaneously, or we could administer the mixture of metabolites while the patient is on antibiotic treatment and then administer the cocktail of live gut commensals after antibiotic treatment is completed. We have added a summary of these new treatment options to our discussion (see lines 712-722):

“The results from this study can be used to inform the rational design of microbiome therapeutics that can re-establish colonisation resistance to restrict VRE intestinal growth. A novel microbiome therapeutic could be composed of a cocktail of live gut commensals that can together deplete nutrients that were enriched with antibiotic treatment and restore the production of microbial metabolites that were decreased with antibiotic treatment. Another option could be a mixture of microbial metabolites that can restrict VRE growth, such as the APBV or PBV mixtures tested in this study. Finally, these two new therapeutics could also be combined, such that we could administer the cocktail of live gut commensals and the mixture of metabolites simultaneously, or we could administer the mixture of metabolites while the patient is on antibiotic treatment and then administer the cocktail of live gut commensals after antibiotic treatment is completed.”

Some minor remarks:

Reviewer 2 Comment 12:

•Figure 3: please change low/average/high ‘metabolites’ into low/average/high ‘concentration’

Author response:

We have revised the figure as requested see (**Fig. S8**).

Reviewer 2 Comment 13:

•Line 92 .. niches, as they showed that growth (of what?) was not impaired...

Author response:

We have edited the text as requested (see lines 106-110):

“A previous study by Caballero and colleagues proposed that vancomycin-resistant *E. faecium* and carbapenem-resistant *Klebsiella pneumoniae* occupied distinct but overlapping intestinal niches, as they showed that growth of each strain was not impaired when they colonised the mouse intestine simultaneously²².”

Reviewer 2 Comment 14:

•Line 104 .. colonisation decreased the concentration of a wide range of metabolites in the gut microbiome (change into microbial metabolites?)

Author response:

We have edited the text as requested (see lines 119-122):

“We previously demonstrated that broad-spectrum antibiotics that promote CRE intestinal colonisation decreased the concentration of a wide range of microbial metabolites in the gut microbiome, and several of these metabolites highly inhibited CRE growth²¹.”

Reviewer 2 Comment 15:

•Line 143-146: listing how many bacterial families' concentration was increased/decreased per antibiotic seems quite irrelevant info. Can the authors limit the information to what families' concentration changes and the most important findings?

Author response:

We have edited this section to be more informative, removing the number of families increased/decreased per antibiotic and instead simply refer to the figure as this clearly illustrates all the changes seen for each antibiotic (see lines 161-165):

“Faecal cultures treated with antibiotics that promote VRE intestinal colonisation had significant decreases in many bacterial families, as shown in **Fig. 1a**. *Ruminococcaceae* and *Bifidobacteriaceae* were decreased in all the antibiotic-treated groups, while decreases in other bacterial families varied depending on the antibiotic tested.”

Reviewer 2 Comment 16:

•Line 149-160: iso-valerate shows a different result as compared to the other SCFA and BCAA, but this is not described.

Author response:

We have added the following sentence to this part of the results section (see lines 182-183):

“In contrast, isovalerate increased in CLI-treated and VAN-treated faecal cultures.”

Reviewer 2 Comment 17:

•Line 217 (add) human microbiomes showed moderate inhibition of VRE growth

Author response:

We have edited the text after incorporating additional data into our manuscript (see lines 243-244):

“Short chain fatty acids that were decreased in antibiotic-treated human faecal microbiomes significantly suppressed VRE growth”

Reviewer 2 Comment 18:

•Line 223-224: We were therefore interested in whether microbial metabolites - that were (delete 'were') decreased in concentration by antibiotics - could also inhibit VRE growth (add) at physiologically relevant concentrations.

Author response:

We have edited the text as requested (see lines 249-251):

“Therefore, we were interested in whether microbial metabolites (that decreased in concentration with antibiotics) could also suppress VRE growth at physiologically relevant concentrations.”

Reviewer 2 Comment 19:

•Line 238-239 (also 241-243) *E. faecium* (add) growth was moderately inhibited by propionate, butyrate, and valerate at the high concentration, and propionate also inhibited *E. faecium* (add) growth at the average concentration (Fig. 4a).

Author response:

After adding in the new data with the additional VRE strains in response to Reviewer 1 Comment 7, this section has been rewritten and now reads as follows (see lines 267-275):

“At average concentrations the growth of two *E. faecium* strains and one *E. faecalis* strain were significantly suppressed by acetate, and the growth of one *E. faecium* strain and three *E. faecalis* strains were significantly suppressed by propionate (Fig. 3). At high concentrations the growth of two *E. faecium* strains and two *E. faecalis* strains were significantly suppressed by acetate, the growth of three *E. faecium* strains and three *E. faecalis* strains were significantly suppressed by propionate, the growth of three *E. faecium* strains and two *E. faecalis* strains were significantly suppressed by butyrate, and the growth of three *E. faecium* strains and one *E. faecalis* strain were significantly suppressed by valerate (Fig. 3).”

Reviewer 2 Comment 20:

•Line 270 Nutrients enriched in antibiotic-treated (add) human faecal microbiomes supported VRE growth

Author response:

We have edited the text as requested (see lines 312-313):

“Nutrients enriched in antibiotic-treated human faecal microbiomes supported VRE growth”

Reviewer 2 Comment 21:

•Line 272-273 We demonstrated that antibiotics that promote VRE intestinal colonisation increased the (delete) concentration of nutrients (add) nutrient availability in faecal microbiomes (Figs. 1-2).

Author response:

We have edited the text as requested (see lines 315-316):

“We demonstrated that antibiotics that promote VRE intestinal colonisation increased nutrient availability in faecal microbiomes (Figs. 1-2).”

Reviewer 2 Comment 22:

•Line 289-294: move to Methods section?

Author response:

We have already included these details in the methods section, and so we have deleted the methods detail in this part of the results.

Reviewer 2 Comment 23:

•Line 301-302: These results support the hypothesis that nutrient depletion (delete) was responsible for (add) can contribute to suppressing VRE growth in a healthy gut microbiome.

Author response:

We have edited the text as requested (see lines 337-338):

“These results support the hypothesis that nutrient depletion contributes to the restriction of VRE growth in a healthy gut microbiome.”

Reviewer 2 Comment 24:

•Line 652 typo: through instead of though

Author response:

We have edited the text as requested (see lines 771-772):

“Healthy gut microbiomes depleted nutrients through nutrient competition, however in antibiotic-treated faecal microbiomes killing of gut commensals enriched for nutrients.”

REVIEWER #3 (REMARKS TO THE AUTHOR):

Reviewer 3 Comment 1:

King et al., Vancomycin-resistant Enterococcus colonize the antibiotic-treated intestine by occupying distinct nutrient- and metabolite-defined intestinal niches

In the present work, the authors investigate antibiotic-induced niches in the gut micromilieu that facilitate the colonization and/or expansion of *Enterococcus* spp. with a focus on VREs. The authors demonstrate that multiple antibiotics modulate the availability of nutrients for *Enterococcus*, but also decrease microbial metabolites that can inhibit *Enterococcus*. Notably, they also found that *Enterococcus* can occupy overlapping niches with other gut-residing pathobionts. This work extends prior reports of Dr. McDonald, and adds important aspects of microbiota-intrinsic regulation of pathobiont colonization and expansion.

Author response:

Thank you.

I have a few comments that require attention by the authors:

Reviewer 3 Comment 2:

1. Ex vivo fecal culture experiments, method: Human microbiotas were treated for 24h with various antibiotics and afterwards harvested for 16S rRNA seq and targeted metabolomics. How was the analysis carried out – inoculum vs. 24h slurries? If so, what happens to the inoculum not treated with antibiotics over 24h, and how representative is the microbiota (composition) after 24h in the incubator/fermenter compared to the native donor stool? How many anaerobes do you lose, and does this reflect the metabolite output of the microbiome as well?

Author response:

The analysis presented in **Figs. 1a and 1b** were carried out by comparing the antibiotic-treated faecal culture samples to the water-treated (vehicle control) faecal culture samples. We have clarified this in the figure, figure legend, the methods section, and the results section.

Plots comparing the donor faecal samples to the water-treated faecal culture samples are shown below in **Fig. R3**.

Overall, the 16S rRNA gene sequencing data showed that the water-treated faecal cultures were comparable to the donor faeces, and contained bacterial families commonly associated with healthy gut microbiota.

We expect to see some changes between the donor faecal inoculum and the water-treated faecal cultures as each faecal donor consumed their own unique diet, while the faecal cultures were all cultured in the same standardised and validated growth medium designed to mimic nutrients found in the colon^{22,23}. Culturing all the faecal microbiota in the same standardised gut growth medium (so with the same “diet”) is an advantage of this technique, as we can be sure that the same nutrients were available to the microbes within all the faecal microbiotas tested from different donors, whether treated with water or one of the different antibiotics. This means that the metabolite output from different donors was comparable between different donors in the water-treated faecal cultures (so had a more comparable “baseline” for comparisons to the different antibiotic treatments).

The metabolite output of the water-treated faecal cultures was representative of healthy faeces, although we do see a shift in some of the metabolites produced, likely due to the change in diet associated with culturing faeces with the standardised growth medium. It may be that these faecal cultures mimic the proximal colon more than the distal colon, where gut commensals prefer to ferment carbohydrates over protein²⁴. We see higher amounts of acetate, propionate, butyrate, lactate, and ethanol (metabolites that are associated with carbohydrate fermentation) and lower amounts of valerate, isobutyrate, and isovalerate (metabolites that are associated with protein fermentation) in water-treated faecal culture samples vs faecal samples²⁴⁻²⁶. Alternatively, our standardised growth medium may represent a healthier diet that is higher in fibre and lower in protein than the diets consumed by some of our faecal donors.

Reviewer 3 Comment 3:

Did you add CO₂ to the culture atmosphere?

Author response:

We used a standard anaerobic mix which contained 10% CO₂, 10% H₂, 80% N₂. We have added these details to our methods section (see lines 853-854):

“These experiments were performed in the standardised gut growth medium under anaerobic conditions, where the anaerobic gas mixture contained 10% CO₂, 10% H₂, 80% N₂⁶⁰.”

Reviewer 3 Comment 4:

What is the variance of the microbiome changes per donor in your assay assuming that there should be considerable variance between microbiome donors and – consecutively – strong variability in the antibiotic changes from minor changes to strong damages of the microbiome based on individual antibiotic resistance profiles?

Author response:

Yes, we do see some heterogeneity in the responses of the microbiome to antibiotic treatment, which is not unexpected as each donor has their own unique gut microbiome profile. As this is a large amount of data to plot, we have summarised these changes as log₂FC per donor in **Fig. S3** below, where bacterial families shown in bold font on the x-axis were significantly changed with each antibiotic treatment.

As mentioned by Reviewer 3, it may be that the presence of antibiotic-resistance genes in some bacteria in the gut microbiomes of specific donors but not others led to some heterogeneity in response to an antibiotic. Moreover, we saw that some bacterial families were present in the water-treated faecal culture samples of some donors but not others, and so these families did not reach statistical significance. For example, *Acidaminococcaceae* was detected in water-treated faecal cultures of 5 of the 12 donors and decreased in these 5 donors with MTZ, CLI, and TZP, and increased in these 5 donors with VAN.

We have added this figure to the supplementary information file and added the following text to the results and discussion:

Text added to the results section (see lines 165-167):

“We found that there was some heterogeneity in the responses of specific bacterial families to antibiotic treatment in faecal cultures inoculated with faeces from different donors (**Figs. S2 and S3**).”

Text added to the discussion section (see lines 536-542):

“Moreover, we saw some heterogeneity in the responses of specific bacterial families to antibiotic treatment in faecal cultures seeded with faeces from different donors. This heterogeneity may be due to differences in the presence of different antibiotic-resistance genes in bacterial species in the gut microbiotas from different donors. Finally, because certain bacterial families were found in some donors but not others, changes in these families did not reach statistical significance, even though they appeared to change in faecal cultures from donors that possessed these bacterial families.”

Figure S3: Log₂FC of bacterial families in antibiotic-treated faecal cultures as compared to water-treated faecal cultures by individual donor. Bacterial families shown in bold font on the x-axis had a significant change in antibiotic-treated faecal cultures as compared to water-treated faecal cultures. n = 12 healthy human faecal donors, Wilcoxon signed rank test (two-sided) of log-transformed abundances with Benjamini Hochberg false discovery rate (FDR) correction, p < 0.05. MTZ = metronidazole, CLI = clindamycin, VAN = vancomycin, CRO = ceftriaxone, TZP = piperacillin/tazobactam.

We saw less donor-to-donor heterogeneity in nutrients and metabolites in response to each antibiotic, as shown in **Fig. S5** below, likely due to the functional redundancy shared by some members of the gut microbiome.

We have added this figure to the supplementary information file and added the following text to the results and discussion:

Text added to the results section (see lines 183-185):

“We found that there was less heterogeneity in the responses of nutrients and metabolites to antibiotic treatment in faecal cultures inoculated with faeces from different donors (**Fig. S5**).”

Text added to the discussion section (see lines 547-552):

“We found that antibiotic treatment enriched for a wide range of nutrients and depleted a wide range of microbial metabolites, and that changes in nutrients and metabolites were less heterogeneous between different donors than changes in bacterial families. This may be due to shared functional redundancy between gut commensals, where different bacterial taxa possess genes that perform similar functions⁴³.”

Reviewer 3 Comment 5:

Please provide compositional plots of the individual probands along individual plots of alpha and beta diversity with control microbiotas incubated without antibiotics.

Author response:

We chose not to display the 16S rRNA gene sequencing data as compositional plots in the previous version of this manuscript as this was a large number of samples to plot, and it left the work of comparing changes in individual bacterial families up to the reader, which can be laborious and also time consuming, especially with differences in the gut microbiomes that we see with different faecal donors. However, we realise some readers might want to look at this data, and so we have added this into the supplementary information section (see **Fig. S2** below).

Figure S2: Stacked bar plots showing the absolute abundances of bacterial families found in water-treated and antibiotic-treated faecal cultures, presented by individual donors. MTZ = metronidazole, CLI = clindamycin, VAN = vancomycin, CRO = ceftriaxone, TZP = piperacillin/tazobactam.

The main alpha diversity measures used in microbiome studies (Shannon index, Simpson index, inverse Simpson index) calculate their indices by taking proportions of the species in the sample. This means these indices are appropriate for relative abundance sequencing data, but not absolute abundance sequencing data. These alpha diversity measures can be misleading for absolute abundance data sets that have large changes in biomass, as we see with broad spectrum antibiotics in this study. For example, imagine that in **Fig. R4** below we have two samples that are each composed of 10 species with identical relative abundances, but have significant differences in their total biomass. If we calculate alpha diversity indices we get identical

values for both samples when we calculate Shannon index, Simpson index, inverse Simpson index, and richness, even though there are clear differences between these two samples. For this reason, we decided not to include alpha diversity analysis in this manuscript.

Figure R4: Example of two samples with an identical number of species and identical relative abundances, but with different total biomass. This data is an illustrative example and does not represent data from our study.

PCA plots have been previously used as beta diversity measures to analyse absolute abundance 16S rRNA gene sequencing data²⁷. Therefore, we generated PCA plots for our data as shown in **Fig. S1** below, and added this figure to the supplementary information. In these plots we can see the majority of the separation occurs along the first principal component, where separation corresponds to antibiotic treatment (separation of the antibiotic-treated faecal culture samples from the water-treated faecal culture samples). We have added this result to the results section (see lines 159-161):

“Principal component analysis (PCA) plots showed separation between antibiotic-treated and water-treated faecal culture samples along the first principal component, which corresponded to antibiotic treatment (**Fig. S1**).”

Figure S1: Principal component analysis (PCA) plots of water-treated vs antibiotic-treated faecal culture samples. Plots used log-transformed absolute abundance data at the family level. MTZ = metronidazole, CLI = clindamycin, VAN = vancomycin, CRO = ceftriaxone, TZP = piperacillin/tazobactam. D1 - D12 = Donor 1 – Donor 12.

Reviewer 3 Comment 6:

2. Ex vivo fecal culture experiments, Enterococcus: As Enterococcus is a highly prevalent, though low abundant commensal in healthy individuals, what about the expansion of enterococci (various spp.) when treated with antibiotics as some clones may exist in the healthy gut microbiome that harbor antibiotic resistance genes blooming after antibiotic expansion.

Author response:

Please see our response to Reviewer 1 Comment 5, which addresses the same question.

Reviewer 3 Comment 7:

Did you investigate whether the enterococci found in Fig. 1C-D are intrinsic to the subject or coming from the spiked ones.

Author response:

An exclusion criterion for recruitment of healthy faecal donors in this study was detection of VRE in faeces. Therefore, the VRE detected in this study correspond to the VRE spiked into the assay. The faecal cultures were plated on Brilliance VRE agar to quantify VRE growth and the donor faeces was screened on Brilliance VRE agar to ensure the donor faeces did not contain VRE.

Reviewer 3 Comment 8:

Line 167: What are the concentrations of antibiotics found in human feces?

Author response:

These concentrations were listed in the methods section (see lines 828-831):

“Gut growth medium was supplemented with water or with one of the following antibiotics at concentrations found in human faeces: 1000 µg/mL VAN, 25 µg/mL MTZ, 152 µg/mL CRO, 139 µg/mL TZP, and 383 µg/mL CLI^{27-32,60}.”

Reviewer 3 Comment 9:

Growth of *E. faecalis* (Fig 1D) seems to differ profoundly between the antibiotic-pretreated microbiotas, can you do a statistic? What is the reason for that as the metabolic milieu shown in Fig 1B does not seem to be so different?

Author response:

There were too many comparisons to include all the p-values on the plots for **Figs. 1c and 1d**, so we have instead summarised them in **Table S2** below and included this table in the supplementary information file:

Table S2: Comparison of *E. faecium* (NCTC 12202) or *E. faecalis* (NCTC 12201) growth in faecal cultures treated with different antibiotics (data from Figs. 1c and 1d). Mixed effects model of log transformed CFU/ml with Tukey's multiple comparisons test. * = $P \leq 0.05$, ** = $P \leq 0.01$, *** = $P \leq 0.001$, **** = $P \leq 0.0001$.

Comparison	E. faecium		E. faecalis	
	Adjusted P value	Summary	Adjusted P value	Summary
Faeces + MTZ vs. Faeces + CLI	0.0858	ns	0.0008	***
Faeces + MTZ vs. Faeces + VAN	0.0129	*	<0.0001	****
Faeces + MTZ vs. Faeces + CRO	<0.0001	****	<0.0001	****
Faeces + CLI vs. Faeces + VAN	0.6969	ns	0.4509	ns
Faeces + CLI vs. Faeces + CRO	0.0008	***	<0.0001	****
Faeces + VAN vs. Faeces + CRO	0.0038	**	0.0009	***

We have added the following text to the results section (see lines 201-202):

“We also found that there was a significant difference in VRE growth in some of the antibiotic-treated faecal cultures compared to others (**Table S2**).”

There are differences in the nutrients that are enriched in the faecal cultures that are treated with different antibiotics that may explain the differences in VRE growth. For example, faecal cultures treated with CRO had higher VRE growth than faecal cultures treated with MTZ (see **Figs. 1c and 1d** and **Table S2**). CRO-treated faecal cultures had a higher enrichment of monosaccharides, amino acids, uracil, and succinate compared to MTZ-treated faecal cultures (see **Fig. R5** below) and the VRE strains could use many of these nutrients to support their growth (see **Figs. 7 and 8**).

Figure R5: CRO-treated faecal cultures had higher enrichment of nutrients compared to MTZ-treated faecal cultures. Heatmap illustrating the log₂ fold change in nutrients that were significantly increased (shown in red) or decreased (shown in blue) in CRO-treated faecal cultures relative to MTZ-treated faecal cultures. Nutrients that were not significantly different were not plotted (shown in white). n = 12 healthy human faecal donors, Wilcoxon signed rank test (two-sided) with Benjamini Hochberg FDR correction, $p < 0.05$. MTZ = metronidazole, CRO = ceftriaxone.

Reviewer 3 Comment 10:

3. Mouse experiment: *E faecalis* and *E faecium* don't show a strong growth in Vanc-containing media compared to other antibiotics (Fig. S1). Are there any non-Vanc-resistant *Enterococcus* clones present in the *Enterococcus* colonies used, or what could be the reason for that and why did the authors use the "weakest antibiotic" for their mouse experiments?

Author response:

VAN was not the weakest antibiotic tested on our study. As shown in **Figs. 1c and 1d**, VAN promoted the second highest VRE growth of the antibiotics tested in the human faecal cultures.

Mice were treated with VAN as it has been frequently demonstrated to promote VRE intestinal colonisation in both humans and mice^{3-5,7}. In this manuscript we showed that VAN treatment resulted in high VRE growth in human faecal cultures (**Figs. 1c and 1d**), and high vancomycin-resistant *E. faecium* growth in mice (**Fig. 2c**). In both figures VRE counts were quantified on Brilliance VRE plates, and so the quantified growth corresponds to vancomycin-resistant colonies only.

The data shown in **Fig. S6** (previously named Fig. S1) shows VRE growth in pure cultures. In the faecal culture experiments outlined in **Figs. 1c and 1d** the faecal cultures were first treated with antibiotics and then the VRE were spiked into the antibiotic-treated or water-treated faecal cultures 24 hours later. This was to simulate VRE exposure to a susceptible (antibiotic-treated) patient. Therefore, it may be that by the time the VRE were spiked into the faecal cultures some of the VAN was used up by killing gut commensals and therefore allowed VRE to grow to higher levels than in pure VRE cultures.

Differences in nutrient utilisation and the presence of microbial metabolites have also been demonstrated to affect antibiotic susceptibility. Nutrient availability is different in faecal cultures with VRE as compared to pure cultures of VRE. Previous work demonstrated that the metabolic state of bacteria influences their susceptibility to antibiotics, and growth on different carbon sources can affect antibiotic efficacy^{28,29}. There are also different metabolites present in faecal cultures with VRE as compared to pure cultures of VRE. Previous studies have demonstrated that some metabolites can lower antibiotic efficacy^{30,31}.

Regardless, we demonstrated that VAN treatment promoted high growth of VRE in human and mouse gut microbiomes, in agreement with previously published literature.

Reviewer 3 Comment 11:

Compared to the human gut micromilieu (Fig. 1B vs. Fig 2), there seems to be inconsistencies between mouse and human Vanc effects. Was this correlated to any microbiome composition and differences between the mouse microbiota and human microbiota used? Please provide some data-based explanation and perhaps revise the conclusion drawn in line 209 as it reads too simplistic.

Author response:

We did not perform sequencing to analyse the bacterial composition of the mouse faeces due to the limited amount of faecal sample that each mouse can provide daily, which was used for ¹H-NMR spectroscopy and vancomycin-resistant *E. faecium* plate counts. However, it is well established that humans and mice have different gut microbiome compositions and consume different diets¹⁵. Regardless, we showed that VAN-treated human faecal cultures and mouse faeces that promoted high levels of VRE growth both showed an enrichment of nutrients (including sugars and amino acids) and a depletion of microbial metabolites (including SCFAs, isobutyrate, lactate, and ethanol).

We have revised the sentence previously on line 209 (see lines 236-239):

"In summary, the results from our mouse experiment were supportive of the results we observed in our *ex vivo* human faecal culture experiments, where we showed that VAN-treated faeces were enriched in nutrients (including sugars and amino acids) and depleted in microbial metabolites (including SCFAs, isobutyrate, lactate, and ethanol)."

As mentioned in our response to Reviewer 2 Comment 3, we have added a discussion of the limitations of mouse models and an explanation of how the mouse data supports our human faecal culture results in the discussion section.

Reviewer 3 Comment 12:

4. pH and metabolite experiments: Why do the authors use a low pH in their model apart from the fact that there is a range of pH values along the gut axis? Is it related to antibiotics and the depletion of commensals or the expansion of acid producing pathobionts? How does the pH change after antibiotics in human slurries in vitro? What was the pH in the mouse model after Vanc administration?

Author response:

We apologise that we did not make it clear why we tested different pH values in these experiments. Previous studies demonstrated that antibiotic treatment increases the pH in the intestine^{16,32}. Moreover, previous studies demonstrated that short chain fatty acids are more inhibitory towards *Enterobacteriaceae* at more acidic physiological pH values^{14,16}. Therefore, we wanted to determine if the microbial metabolites were inhibitory at the more acidic pH values that are found in healthy gut microbiomes^{12,33}.

We have clarified our reasoning for testing the different pH values in the metabolite experiments in the manuscript (see lines 251-255):

“Previous studies also demonstrated that antibiotic treatment increased the pH in the intestine^{33,34}. Moreover, previous studies demonstrated that short chain fatty acids were more inhibitory towards *Enterobacteriaceae* growth at more acidic physiological pH values^{21,33}. Therefore, we measured the effects of microbial metabolites at physiological pH values found in the healthy large intestine (pH 6-7)^{35,36}.”

As the impact of antibiotic treatment on gut content pH had already been established in the literature, we did not collect pH measurements in the faecal culture experiments or the mouse experiment. However, as an example a previous study by Shimizu *et al.* (2021) investigated the effects of antibiotics (1 week of vancomycin, cefoperazone, or ampicillin) on the pH of intestinal contents from mice, and compared this to the pH of intestinal contents from untreated healthy control mice and to germ free mice³². After antibiotic treatment, there was a significant increase in the pH of the caecal and colon contents. The following pH values were measured in the colon contents from each of the treatment groups: pH 5.87 (range 5.63-6.02) in healthy control mice, pH 7.97 (range 7.76–8.12) in vancomycin-treated mice, pH 7.94 (7.82–8.20) in cefoperazone-treated mice, pH 7.89 (7.71–8.07) in ampicillin-treated mice, and pH 7.95 (7.72–8.14) in germ-free mice.

Reviewer 3 Comment 13:

The OD values shown in Fig 3 and 4 for no metabolites group are very low for Enterococci, although the bacteria were cultured over night. Is the TS media or why do you see so little growth in your controls? Can you show the growth curves instead of ODs measured at an overnight time point?

Author response:

We did not collect growth curves in this experiment, we only took end point OD measurements. The OD may be lower than the reviewer expected as these cultures were incubated under anaerobic conditions, not aerobic conditions. We chose to perform these experiments under anaerobic conditions as we were mimicking conditions found in the healthy (non-antibiotic-treated) intestine where VRE would encounter microbial metabolites under anaerobic conditions. We have updated the figure legends to clarify that the metabolite inhibition assays were performed under anaerobic conditions.

Minor point:

Reviewer 3 Comment 14:

- Fig. 7 and 8: Do the shaded bands show the predicted 95% confidence intervals?

Author response:

Each condition shows the predicted 95% credible intervals (aka Bayesian 95% confidence intervals) as shaded bands. It may be difficult to see some shaded bands in conditions where the replicates were all very similar, as the 95% credible intervals would be very narrow.

REVIEWER #4 (REMARKS TO THE AUTHOR):

Reviewer 4 Comment 1:

In this manuscript the authors investigate mechanisms associated with antibiotics-driven colonization of vancomycin resistant *Enterococcus* (VRE) in the intestine, which is a highly relevant clinical challenge. A main strength is the inclusion of both ex-vivo faecal culture experiments and investigations in mice. However, several aspects need to be clarified to determine the validity of the study and results. The conclusions drawn not be fully supported by the results partly due to the lack of suitable controls.

Author response:

We address the reviewer's specific concerns in detail in our responses below.

Results:

Reviewer 4 Comment 2:

• First of all, the authors need to clarify if the metabolites are from relative or absolute quantification. This is not clear from the methods description, but will be crucial for the interpretations in Fig 1b and Fig 2. In relative quantification, the decrease of a group of metabolites will automatically result in increase of the remaining metabolites, although the absolute levels might be stable. The authors interpret the metabolic changes after antibiotics as increased or decreased, and these interpretations make up the basis for the subsequent experiments. Please clarify how the quantification of NMR metabolites was performed, and how absolute metabolite concentrations were achieved if this is the case.

Author response:

¹H-NMR spectroscopy provides absolute quantification²¹. We have clarified this in the methods section (see lines 925-927):

“¹H-NMR spectroscopy was performed on human faecal culture samples, VRE culture samples, and mouse faecal samples to measure absolute quantification of metabolites, following previously published protocols^{21,24}.”

Metabolites were quantified as peak integrations, as we outlined in the methods section (see lines 961-962):

“Chenomx NMR Suite v9.02 (Chenomx, Canada) was used for peak identification and representative peaks were selected and integrated.”

Reviewer 4 Comment 3:

• **Figure 3 (A), Figure 4:** The authors aim to test if metabolites can inhibit VRE growth. Use of the term “inhibit” indicates causality, and the authors need to clarify or hypothesize which biological mechanisms could explain that a metabolite directly inhibits survival of a bacteria. In fact, figure 3 clearly shows that low and average concentrations of a mix of metabolites leads to significantly increased growth of both *E.faecium* and *E.faecalis*. Even a high mix of metabolites lead to increased growth at pH 7, and it is only at pH 6 that the high concentrations lead to decreased growth. pH 6 is low at the pH spectrum of healthy feces.

Author response:

The data shown in **Fig. S8** (previously Fig. 3) demonstrates the effects of a mixture of 10 metabolites (that we saw were decreased in antibiotic-treated faecal cultures) on VRE growth. This metabolite mixture contained some metabolites that suppressed VRE growth (e.g. acetate, propionate, butyrate, and valerate, as seen in **Fig. 3**) and some metabolites that promoted VRE growth (e.g. 5-aminovalerate and ethanol, as seen in **Fig. S12**).

We observed significant but incomplete suppression of vancomycin-resistant *E. faecium* and *E. faecalis* growth by acetate, propionate, butyrate, and valerate when tested individually. However, in this revised version of the manuscript we provide new data showing that a mixture of acetate, propionate, butyrate, and valerate (“APBV mixture”) and a mixture of propionate, butyrate, and valerate (“PBV mixture”) provided complete or near complete suppression of *E. faecium* and *E. faecalis* growth at pH 6 (**Fig. 4**). We have added this data into the results section and discussion section of our manuscript:

Text added to results section (see lines 293-303):

“We showed that acetate, propionate, butyrate, and valerate were individually capable of suppressing VRE growth at high concentrations found in human faeces. Therefore, next we tested whether a mixture of propionate, butyrate, and valerate (“PBV” mixture) or a mixture of acetate, propionate, butyrate, and valerate (“APBV” mixture) were able to more highly suppress VRE growth. At pH 6 we showed that *E. faecium* growth was fully suppressed by the APBV mixture and nearly fully suppressed by the PBV mixture, while *E. faecalis* growth was fully suppressed by both the APBV mixture and PBV mixture (**Fig. 4**). At pH 6.5 both the APBV and PBV mixtures significantly suppressed *E. faecium* and *E. faecalis* growth (**Fig. 4**). At pH 7 the APBV mixture significantly suppressed the growth of one *E. faecium* strain and three *E. faecalis* strains, while the PBV mixture significantly suppressed the growth of two *E. faecium* strains and two *E. faecalis* strains (**Fig. 4**).”

Text added to results section (see lines 306-310):

“Overall, we observed significant but incomplete suppression of vancomycin-resistant *E. faecium* and *E. faecalis* growth by acetate, propionate, butyrate, and valerate (at high concentrations) when tested individually at pH 6. However, the APBV and PBV mixtures provided complete or near complete suppression of *E. faecium* and *E. faecalis* growth at pH 6.”

Figure 4: VRE growth was highly suppressed by mixtures of SCFAs at high concentrations. Vancomycin-resistant *E. faecium* and vancomycin-resistant *E. faecalis* strains were grown in tryptic soy broth supplemented with a mixture of propionate, butyrate, and valerate (“PBV”) or a mixture of acetate, propionate, butyrate, and valerate (“APBV”) at concentrations mimicking the high concentrations measured in human faeces, or unsupplemented (“No metabolites” control). Broth was adjusted to pH 6, 6.5, or 7 to mimic the pH of the healthy large intestine, and cultures were incubated under anaerobic conditions overnight. Data shown as medians \pm IQR, with 12 replicates from 3 independent experiments. Kruskal-Wallis with Dunn’s multiple comparison test comparing the no metabolites control to each metabolite mixture at the same pH. * $P \leq 0.05$, ** = $P \leq 0.01$, *** = $P \leq 0.001$, **** = $P \leq 0.0001$. OD₆₀₀ = optical density at 600 nm.

We have used the term “suppress” instead of “inhibit” in our results section to avoid confusion, but we speculate on the mechanism of growth suppression in our discussion (see lines 573-590):

“Although individually acetate, propionate, butyrate, and valerate significantly suppressed *E. faecium* and *E. faecalis* growth at high concentrations, growth was not fully suppressed. Therefore, we were also interested in whether a mixture of these metabolites showed synergistic or additive effects, where the metabolite mixture provided higher growth suppression than each metabolite provided individually. We showed that the APBV and PBV mixtures provided complete or near complete suppression of *E. faecium* and *E. faecalis* growth at pH 6. The additive effects of these metabolites suggests that they act through a similar mechanism. A previous study by Sorbara and colleagues demonstrated that acetate, propionate, and butyrate inhibited antibiotic-resistant *Enterobacteriaceae* growth through intracellular acidification³³. Although the inhibitory mechanism of valerate has not been demonstrated yet, it has also been shown to

inhibit *C. difficile* and CRE growth^{21,24}. Both butyrate and valerate have also been demonstrated to promote intestinal barrier function which may help to prevent dissemination of VRE from the intestine⁴⁴. Further dose testing is required to determine the optimal concentration of each metabolite within this inhibitory metabolite mixture. However, this metabolite mixture shows potential for the use to restrict VRE growth when administered to a patient with VRE intestinal colonisation or when administered to a patient as a prophylactic treatment (e.g. during antibiotic treatment when the patient is susceptible to VRE intestinal colonisation).”

A pH of 6 is representative of the pH of the healthy human colon. For example, Cummings *et al.* (1987) quantified the pH in different regions of the intestine in sudden death victims¹². Their averaged pH measurements in different regions of the large intestine were as follows:

- Ascending large intestine: pH 5.7
- Transverse large intestine: pH 6.2
- Descending large intestine: pH 6.6
- Sigmoid/rectum: pH 6.3

In another example, healthy mice have a pH of 5.87 (range 5.63-6.02) in their colon content³².

Reviewer 4 Comment 4:

A condition with these high levels of several metabolites simultaneously may not be realistic or promote survival of any bacteria. This raises the question if a high concentration of a mixture of metabolites at low pH would lead to decreased growth of any bacteria, which should be tested using a control. Wouldn't any other bacteria, commensal or pathobionts, suffer from low growth under those conditions?

Author response:

We looked at the metabolite concentration measurements from the faeces of the healthy donors that were used to define the low, average, and high concentrations of metabolites tested in **Figs. 3, 4, and S8-S14**. Interestingly, the highest concentrations of acetate, propionate, and butyrate were all found in the same donor, while the highest valerate, isobutyrate, and isovalerate concentrations were found in the donor with the second highest concentration of propionate and butyrate and the third highest concentration of acetate (**Table R2**). This data demonstrates that it is not unreasonable to find these high metabolite concentrations for propionate, butyrate, and valerate within the same healthy gut microbiome.

Table R2: Comparison of the high metabolite concentrations tested in this study to the metabolite concentrations measured in two of the healthy faecal donors used in this study.

Metabolite	High concentrations tested (mM)	Concentrations measured from Donor 4 (mM)	Concentrations measured from Donor 9 (mM)
Acetate	120	122.7	94.5
Propionate	35	34.7	28.7
Butyrate	40	38.8	30.5
Valerate	12	2.7	12.0
Isobutyrate	5	0.3	5.3
Isovalerate	5	0.5	4.6

A pH of 6 is a normal pH to find in a healthy large intestine, as explained in Reviewer 4 Comment 3.

It is possible that supplementing the gut microbiome with a high or even average concentration of metabolites may inhibit the growth of some gut commensals. This is not unexpected, as the rules governing colonisation resistance influence commensals as well as pathogens, and we would expect that individuals with different metabolite concentrations would have different abundances of gut commensals. In other words, gut commensals may not be achieving to their maximum growth in the healthy gut microbiome that could be achieved if the commensals colonised a sterile intestine due to the presence of microbial metabolites.

There is no “control” commensal to test for the healthy gut microbiome, and selecting one commensal to test would not be representative of others and may risk introducing bias. However, below in **Fig. R6** we show a

few examples of the effects that a mixture of propionate, butyrate, and valerate (“PBV”) can have against a panel of gut commensals. *Bacteroides luhongzhouii* growth was promoted by the PBV mixture at average and high concentrations. *Flavonifractor plautii* growth was promoted at average concentrations but inhibited at high concentrations. *Acidaminococcus fermentans* growth was inhibited at average and high PBV concentrations, but growth at average and high concentrations was comparable. *Bifidobacterium longum* growth was unaffected at average PBV concentrations but inhibited at high concentrations. Regardless, in all these examples there is still growth of the gut commensals in the high concentration PBV mixture at pH 6, unlike the VRE at in this mixture at pH 6 (see Fig. 4).

Figure R6: A mixture of propionate, butyrate, and valerate (mimicking concentrations found in healthy human faeces) have different effects on the growth of several gut commensals. Each gut commensal strain was grown in modified GAM broth supplemented with a mixture of propionate, butyrate, and valerate (“PBV”) at average or high concentrations measured in faeces, or unsupplemented (“No metabolites”) at pH 6. Data shown as median ± IQR, with 12 replicates from 3 independent experiments. Kruskal-Wallis test, * $P \leq 0.05$, ** = $P \leq 0.01$, *** = $P \leq 0.001$, **** = $P \leq 0.0001$. OD₆₀₀ = optical density at 600 nm.

It may also be possible to further optimise the dose of the PBV mixture to determine if there is a concentration that still provides high inhibition of VRE while providing less inhibition of gut commensals, but this is beyond the scope of this manuscript.

We have not added this commensal metabolite inhibition data to the manuscript, as there are no specific “control” bacteria for the gut microbiome, and we had generated this data for a separate publication we are working on. However, we have added the following to our discussion (see lines 585-587):

“Further dose testing is required to determine the optimal concentration of each metabolite within this inhibitory metabolite mixture.”

Reviewer 4 Comment 5:

• For figure 4, use of a control species would be useful to investigate if the decreased survival is specific to the VRE or applies to bacteria in general.

Author response:

See our response to Reviewer 4 Comment 4 above. This is beyond the scope of this manuscript, and this is something we are currently investigating for another manuscript that we are writing.

Reviewer 4 Comment 6:

Furthermore, How did do you decided on ranges for low, medium and high mixture concentrations? For example, when isovalerate inhibits bacterial growth with 128 mM, is this type of concentration found only at the tail-end of a population of healthy (or sick) individuals?

Author response:

We used concentrations of each metabolite measured in healthy human faeces, as mentioned in the methods section (see lines 1002-1005):

“For the low mixture all metabolites were tested at the low concentrations measured in faeces, for the average mixture all metabolites were tested at the average concentrations measured in faeces, and for the high mixture all metabolites were tested at the high concentrations measured in faeces.”

The low, average, and high concentrations found in human faeces are specified in the x-axis for in **Figs. 3, S9-S14**. For example, in **Fig. 3** the concentration of propionate tested was 5 mM for low (specified as “Low (5)” on the x-axis), 15 mM or average (specified as “Average (15)” on the x-axis), and 35 mM for high (specified as “High (35)” on the x-axis).

However, to be clearer we have added a list of the concentrations tested as **Table S5** rather than just referencing the previous paper where we measured these concentrations.

We also tested a wide range of metabolite concentrations get a better idea of the range of concentrations that may be inhibitory and to calculate IC₅₀ values. In the specific example of isovalerate mentioned in the reviewer comment above, 128 mM would be supraphysiological as the maximum concentration measured in healthy human faeces is ~ 5mM.

Reviewer 4 Comment 7:

• Figure 5 – What I the main conclusion from this result? Obviously adding micronutrient to minimal media will boost growth of any other bacteria? Again, the use of a suitable control would be advisable.

Author response:

As outlined in our manuscript, the purpose of the data outlined in **Fig. 5** was to demonstrate that the mixture of nutrients that were enriched with antibiotic treatment were capable of supporting VRE growth, before moving on to individual carbon and nitrogen testing that identified the specific nutrients that VRE could utilise.

We do expect that this mixture of nutrients would also support the growth of gut commensals, as it was the killing of gut commensals that resulted in this enrichment of nutrients. As mentioned above, there is no “control” strain for the gut microbiome. However, below we show examples of gut commensal strains that can also growth in this mixture of nutrients where these species belong to some of the bacterial families that we found were decreased with these antibiotics (**Fig. R7**). Again, this data was collected as part of a separate manuscript, and so we would prefer not to include this data in the manuscript.

Figure R7: Examples of gut commensals cultured in basal minimal medium supplemented with a mixture of nutrients that were increased with antibiotic treatment. Each strain was cultured under anaerobic conditions overnight. Data shown as mean \pm SD, with 10 replicates from 2 independent experiments. Unpaired t-test, **** = $P \leq 0.0001$. OD₆₀₀ = optical density at 600 nm.

Reviewer 4 Comment 8:

• Figure 6 Did the antibiotic-naïve faecal microbiome possess the same “strain” of *E. faecium* you inoculated for the other conditions (spent supernatant)? Or are they *E. faecium* free?

Author response:

The presence of VRE in the healthy donor stool was an exclusion criterion for this study. We screened the healthy donors used in this experiment for VRE by assessing growth on Brilliance VRE plates, and we did not find any growth. Therefore, the healthy donor faeces used to inoculate the faecal cultures shown in Fig. 6 did not contain the same VRE strains that were tested in this assay.

Reviewer 4 Comment 9:

What if you were to inoculate with any other bacteria a spent supernatant (generated as in “faeces supernatant”): would they not stop growing and show “suppression” due to lack of nutrients?

Author response:

It would depend on the bacteria of interest. If the exogenous bacteria consume nutrients that are normally depleted in a healthy gut microbiome, then we would not expect them to grow.

Reviewer 4 Comment 10:

- Please clarify what the results in Fig 5 and 8 are normalized against; medium without the nutrients?

Author response:

This was described in the AMiGA methods section of the manuscript (see lines 1117-1119):

“The carbon source utilisation growth curves and mixed nutrient growth curves were normalised to the no carbon control using the subtraction method. The nitrogen utilisation growth curves were normalised to the no nitrogen control using the subtraction method.”

Reviewer 4 Comment 11:

- The authors investigated preferred nutrients for VRE, with the claim that “This information could help inform the design of new microbiome therapeutics composed of gut commensals that can outcompete the VRE strains for use of these specific nutrients”. However, as shown in Fig 7-9 the VRE can utilize several very common carbon and nitrogen sources, which are also utilized by many other strains, and it would be crucial to investigate how therapeutic depletion of these nutrient sources would affect the remaining gut microbiota.

Author response:

In antibiotic-treated faecal cultures we showed that the remaining gut bacteria did not deplete these nutrients, which is why they were enriched compared to water-treated faecal cultures. Healthy gut microbiota contain gut commensals that can deplete these nutrients, and it is killing of these gut commensals that results in an increase in the amounts of these nutrients.

The idea of designing a new microbiome therapeutic composed of gut commensals would be to include common healthy gut commensals that are normally found in a healthy gut microbiome to speed up recovery of the gut microbiome, and to restore a state of nutrient depletion and inhibitory metabolite production that restores colonisation resistance to restrict VRE growth. Therefore, our goal would be to develop a microbiome therapeutic to restore these gut commensals that can use these nutrients to restore nutrient depletion. As this is what is normally found in a healthy gut microbiome, this shouldn't be an issue for a microbiome therapeutic. However, this is something that would be investigated in future studies of a new microbiome therapeutic.

Reviewer 4 Comment 12:

- Figure 10 a – What makes you conclude that there is a low inhibition? How could you co-culture of *E. faecium* and *E. faecalis* together and be able to separately assess their CFU ? Please explain the methodology used to differentiate this.

Author response:

We concluded that there was low inhibition as there was a small but statistically significant decrease in growth, however the overall growth was still high for both *E. faecium* and *E. faecalis* when grown in co-culture. We are not sure if the reviewer does not like the use of the term “inhibition” of growth (as mentioned in Reviewer 4 Comment 3), so we have reworded the results section to read as follows (see lines 468-469):

“However, for *E. faecium* there was a small but significant suppression of growth in the co-culture compared to growth in monoculture.”

As described in the methods section and in the **Fig. 10** legend, our co-culture experiments with vancomycin-resistant *E. faecium* and vancomycin-resistant *E. faecalis* used Brilliance VRE plates to quantify the growth of each species. Brilliance VRE plates are chromogenic plates that can differentiate these two species based on the colour of their colonies, where vancomycin-resistant *E. faecium* grow as purple colonies, while vancomycin-resistant *E. faecalis* grow as light blue colonies.

Reviewer 4 Comment 13:

• In this study, strains with vanA-type glycopeptide resistance gene is utilized. Do the authors expect similar results in VRE strains with other resistance genes?

Author response:

The most frequently isolated and clinically relevant vancomycin resistance determinants for *E. faecium* and *E. faecalis* are *vanA* and *vanB*³⁴. We have only tested *vanA* type VRE strains in this study as the *vanA* type resistance is the most dominant genotype worldwide and shows higher levels of overall antibiotic-resistance compared to *vanB*³⁵⁻³⁷. We had previously searched for *vanB* strains to purchase, but there were very few strains available – e.g. we could only find one *vanB E. faecium* strain to purchase and it was vancomycin-dependent. As such, we did not perform this comparison of *vanA* and *vanB* as we could not find enough *vanB* strains to make a definitive conclusion. However, we have added this as a limitation of our study and mentioned this in our discussion (see lines 742-747):

“Moreover, we have only tested *vanA* VRE strains in this study as *vanA* type resistance is the most dominant genotype worldwide, *vanA* shows higher levels of overall antibiotic-resistance compared to *vanB*, and *vanA* strains were the most commercially available stains to test⁵²⁻⁵⁴. However, future studies should investigate whether there is a difference in nutrient utilisation and metabolite inhibition of other non-*vanA* type resistant VRE.”

Discussion:

Reviewer 4 Comment 14:

• Overall, the discussion is long and repetitive and could benefit from more concise language avoiding repetitions. It also lacks connections to the larger context of other antibiotic-resistant pathobionts.

Author response:

As the reviewer did not indicate which parts of the discussion should be cut, we read through the discussion and edited/cut out parts that we felt could be cut or could be expressed in a more concise manner.

It is not feasible to discuss all the literature on nutrient competition and inhibitory metabolite production for other antibiotic-resistant pathogens, and so we had highlighted the relevant literature that focused on other multidrug-resistant pathogens. We have thoroughly discussed results from carbapenem-resistant *Enterobacteriaceae* (CRE) - another antibiotic-resistant pathogen that grows in a nutrient-enriched and metabolite-depleted intestinal environment.

The reviewer did not mention specific literature that they thought was missing from our discussion. If there is a specific relevant study that the reviewer believes is missing, we are happy to add it in. But as this revision of the manuscript has also added in additional text to the discussion based on reviewer feedback, we did not add further discussion about other pathobionts to keep the discussion a reasonable length.

Reviewer 4 Comment 15:

• Please make sure to differentiate between colonization and expansion of pathobionts that already exist in the microbiome

Author response:

We have read through our discussion and ensured we used the correct terminology throughout. VRE are not normally found in the healthy gut microbiome and are often acquired in healthcare settings, and so we have used the term colonisation not expansion, as expansion implies a bloom of a bacteria that is already part of the gut microbiome³⁸⁻⁴⁰.

Reviewer 4 Comment 16:

• **Faecal metabolites are produced by a combination of interactions within the microbiome, but also between the microbiome and the human host. The connection to the human host should be elaborated.**

Author response:

This was part of the reason we also wanted to test mice in this study. We measured the faecal metabolome of water-treated and antibiotic-treated mice, where water-treated mice had intact colonisation resistance. Moreover, our gut growth medium used in the *ex vivo* human faecal cultures contained host-derived substances such as bile acids and mucin. In our *ex vivo* faecal culture experiments we showed that water-treated faecal cultures have intact colonisation resistance against VRE in the absence of additional host-microbe derived metabolites.

Reviewer 4 Comment 17:

• **L .631 –Wouldn't a bloom be associated with VRE > 10% of gut microbiome composition, as in *C. difficile* infections? The experiments shows log CFU/mL of 8 to 9 for VRE. Which is way lower than the 12 that can be found in a healthy gut microbiome (colon). If we unduly compare, it would be a very tiny portion of a microbiome composition, perhaps <0.1%.**

Author response:

VRE growth is promoted by broad-spectrum antibiotics, which reduce the abundance of many bacterial families (as shown in **Fig. 1a**). As such, gut commensal abundances of 10^{11} CFU/g found in the healthy large intestine are not representative of gut commensal abundances in an antibiotic-treated gut microbiome⁴¹.

Moreover, previous studies have shown that VRE can dominate the antibiotic-treated gut microbiome. For example, Taur *et al.* (2012) investigated changes in the gut microbiome of patients undergoing allogeneic hematopoietic stem cell transplantation, where these patients frequently become colonised by VRE and develop invasive infections, and are often exposed to multiple antibiotics (including vancomycin, metronidazole, fluoroquinolones, and beta-lactams)⁴². They showed that *Enterococcus* was the most frequently dominating genus, where it composed more than 30% of the gut microbiota in 40.4% of patients. Of those patients, 92% had the *vanA* gene suggesting the presence of VRE.

A “bloom” of a bacterial taxa corresponding to >10% abundance is not a standard definition used in our field. Typically, a bloom in a bacterial population refers to a significant increase in its abundance. However, to avoid confusion we have reworded this sentence to say the following (see lines 707-710):

“This highlights that VRE have different strategies for utilising nutrients to colonise an antibiotic-treated gut microbiome compared to other multidrug-resistant pathogens, which may have implications for the design of new microbiome therapeutics to decolonise pathogens from the intestine.”

Introduction:

The manuscript would benefit from additional introduction on the following topics:

Reviewer 4 Comment 18:

- Since the aim is to understand mechanisms for resistance, it Who are the Enterococcus genus, and what vancomycin antibiotics do on a molecular level?

Author response:

We aimed to keep our introduction concise and relevant to the clinical problem, and as such we focused our introduction on VRE. However, in response to Reviewer 4 Comment 13, Reviewer 4 Comment 19, and Reviewer 4 Comment 20, we have provided additional background information on *E. faecium*, *E. faecalis*, and VRE.

The aim of our paper is to investigate the mechanisms of colonisation resistance (defined as the ways in which the gut microbiome protects us against the intestinal colonisation with pathogens), not mechanisms of antibiotic resistance. As such, we have not added the details about the molecular mechanism of vancomycin resistance to this paper.

Reviewer 4 Comment 19:

- Please describe why *E. faecalis* is relevant. *E. faecalis* is not mentioned in the introduction and only in passing in most of the paper.

Author response:

We have edited the introduction to give a more balanced introduction to both pathogens. We have added the following text to the introduction section (see lines 48-58):

“*E. faecium* and *E. faecalis* are responsible for 75% of enterococcal infections and are commonly associated with hospital outbreaks of invasive infections such as bloodstream infections, urinary tract infections, and endocarditis^{2,4,5}. *E. faecalis* was more virulent and more prevalent in healthcare-associated infections than *E. faecium*, however, the prevalence of *E. faecium* is increasing due to the rise of vancomycin-resistant and lactam-resistant *E. faecium* strains⁶. *E. faecium* has higher levels of intrinsic and acquired resistance than *E. faecalis*, where 80% of *E. faecium* isolates are vancomycin-resistant compared to 10% of *E. faecalis* isolates⁴. Vancomycin-resistant *E. faecium* has gained attention as the World Health Organisation classified it as a high priority pathogen due to its unfavourable rankings in several criteria, including mortality, trends of resistance, transmissibility, preventability, treatability, and pipelines for new medicines and diagnostics⁷.”

Reviewer 4 Comment 20:

- Are *E. faecalis* and *E. faecium* always present in the human gut, or is a de-novo colonization necessary to create VRE infection?

Author response:

VRE are hospital-acquired pathogens, where patients acquire VRE from other patients, hospital rooms, or from healthcare workers in acute and long-term healthcare facilities³⁸⁻⁴⁰. We have added this text to the introduction (see lines 60-62):

“VRE can be acquired from other patients, hospital rooms, or healthcare workers in acute and long-term healthcare facilities⁹⁻¹¹.”

Methods:

Reviewer 4 Comment 21:

- Add cat number and company that supplied the materials. Some are missing.

Author response:

We have added **Table S4** to this manuscript which list the reagents used alongside the supplier and catalogue numbers.

Reviewer 4 Comment 22:

- **Line 687 – Please, briefly mention how you assessed the MIC.**

Author response:

We have added the following text (see lines 810-814):

“Briefly, MIC assays were carried out using Mueller-Hinton broth in 96-well plates, where the broth was supplemented with no antibiotics or with 2-fold dilutions of each antibiotic ranging from 0.125-128 mg/L. Each VRE strain was inoculated into the supplemented broth at 5×10^5 CFU/mL and plates were incubated aerobically at 37°C for 18 hours. Growth was assessed by taking OD⁶⁰⁰ measurements.”

Reviewer 4 Comment 23:

- **Line 694 – What is “distal gut growth medium”? Formulation? Or company ?**

Author response:

We have added the following text in addition to the reference (see lines 822-828):

“The gut growth medium was composed of the following: unmodified starch (5 g/L), casein (3 g/L), inulin (1 g/L), sodium chloride (0.1 g/L), peptone water (2 g/L), yeast extract (2 g/L), sodium bicarbonate (2 g/L), pectin (2 g/L), xylan (2 g/L), arabinogalactan (2 g/L), calcium chloride (0.01 g/L), porcine gastric mucin type II (4 g/L), potassium phosphate monobasic (0.04 g/L), potassium phosphate dibasic (0.04 g/L), magnesium sulphate heptahydrate (0.01 g/L), hemin (0.005 g/L), menadione (0.001 g/L), bile salts (0.5 g/L), and L-cysteine hydrochloride (0.5 g/L)⁶⁰.”

Reviewer 4 Comment 24:

- **Line 698 – How was “VRE growth” was assessed? OD or CFU?**

Author response:

This was stated in the methods section (see lines 833-834):

“VRE growth was quantified by plating on Columbia agar and incubating the plates at 37°C for 24 hours.”

Reviewer 4 Comment 25:

- **Line 715 – What is a “faecal slurry”? A mixture of liquified feces and PBS?**

Author response:

The faecal slurry was faeces homogenised in 0.9% saline. We have edited the methods section to clarify this (see lines 854-856):

“Faecal slurries were prepared by homogenising fresh faeces in degassed 0.9% saline. Each faecal slurry was inoculated into the gut growth medium at a final concentration of 2% (w/v).”

Reviewer 4 Comment 26:

- Line 773 – Please add which is elongation, annealing, etc.

Author response:

We have edited this sentence as follows (see lines 913-915):

“The PCR cycle conditions for qPCR were as follows: 50°C for 3 min for UNG treatment, 95°C for 10 min for *Taq* polymerase activation, and 40 cycles of 95°C for 15 s for denaturation and 60°C for 1 min for annealing and extension.”

Reviewer 4 Comment 27:

- Line 845 to 847 – How do you distinguish and assess *E. faecalis*?

Author response:

This part of the manuscript describes the methods for the mouse experiment. Mice in this experiment were administered vancomycin-resistant *E. faecium*, not vancomycin-resistant *E. faecalis*. As stated in the methods, vancomycin-resistant *E. faecium* growth was assessed using Brilliance VRE plates, and we did not detect the presence of any vancomycin-resistant *E. faecalis*.

Reviewer 4 Comment 28:

- Line 966 – Please explain what Credible intervals are?

Author response:

The 95% credible interval is another name for the Bayesian 95% confidence interval. So in this case, this means that there is a 95% probability that the growth curves fall within the intervals shown in the AMiGA growth curve models.

Reviewer 4 Comment 29:

- Line 985 to 989– How did you quantify pathogen growth?

Author response:

This was stated in the methods section of the manuscript (see lines 1143-1146):

“Pathogen growth was quantified by plating samples on Brilliance VRE plates to quantify *E. faecium* or *E. faecalis* growth and on Brilliance CRE plates to quantify *K. pneumoniae*, *E. coli*, or *E. hormaechei* growth.”

REVIEWER #5 (REMARKS TO THE AUTHOR):

Reviewer 5 Comment 1:

Author response:

Thank you for taking the time to co-review this manuscript.

Authors note for Fig. 10: In the previous version of the manuscript **Fig. 10** had a large amount of unused white space. Therefore, for this version of the manuscript we reconfigured the positioning of each panel to reduce this white space, without altering the data that was presented in the previous version of the manuscript.

REFERENCES

1. Donskey, C. J. *et al.* Effect of antibiotic therapy on the density of vancomycin-resistant enterococci in the stool of colonized patients. *N Engl J Med* **343**, 1925-32 (2000).
2. Stiefel, U., Pultz, N. J., Helfand, M. S. & Donskey, C. J. Increased susceptibility to vancomycin-resistant *Enterococcus* intestinal colonization persists after completion of anti-anaerobic antibiotic treatment in mice. *Infect Control Hosp Epidemiol* **25**, 373-9 (2004).
3. Isaac, S. *et al.* Microbiome-mediated fructose depletion restricts murine gut colonization by vancomycin-resistant *Enterococcus*. *Nat Commun* **13**, 7718 (2022).
4. Al-Nassir, W. N. *et al.* Both oral metronidazole and oral vancomycin promote persistent overgrowth of vancomycin-resistant enterococci during treatment of *Clostridium difficile*-associated disease. *Antimicrob Agents Chemother* **52**, 2403-6 (2008).
5. Banerjee, T., Anupurba, S., Filgona, J. & Singh, D. K. Vancomycin-resistance enterococcal colonization in hospitalized patients in relation to antibiotic usage in a tertiary care hospital of North India. *J Lab Physicians* **7**, 108-111 (2015).
6. Rice, L. B., Hutton-Thomas, R., Lakticova, V., Helfand, M. S. & Donskey, C. J. Beta-lactam antibiotics and gastrointestinal colonization with vancomycin-resistant enterococci. *J Infect Dis* **189**, 1113-8 (2004).
7. Isaac, S. *et al.* Short- and long-term effects of oral vancomycin on the human intestinal microbiota. *J Antimicrob Chemother* **72**, 128-136 (2017).
8. Brandl, K. *et al.* Vancomycin-resistant enterococci exploit antibiotic-induced innate immune deficits. *Nature* **455**, 804-7 (2008).
9. Gu, S. *et al.* Bacterial community mapping of the mouse gastrointestinal tract. *PLoS One* **8**, e74957 (2013).
10. Archambaud, C., Derré-Bobillot, A., Lapaque, N., Rigottier-Gois, L. & Serror, P. Intestinal translocation of enterococci requires a threshold level of enterococcal overgrowth in the lumen. *Sci Rep* **9**, 8926 (2019).
11. Winston, J. A. & Theriot, C. M. Diversification of host bile acids by members of the gut microbiota. *Gut Microbes* **11**, 158-171 (2020).
12. Cummings, J. H., Pomare, E. W., Branch, W. J., Naylor, C. P. & Macfarlane, G. T. Short chain fatty acids in human large intestine, portal, hepatic and venous blood. *Gut* **28**, 1221-1227 (1987).
13. Fabich, A. J. *et al.* Comparison of carbon nutrition for pathogenic and commensal *Escherichia coli* strains in the mouse intestine. *Infect Immun* **76**, 1143-52 (2008).
14. Yip, A. Y. *et al.* Antibiotics promote intestinal growth of carbapenem-resistant *Enterobacteriaceae* by enriching nutrients and depleting microbial metabolites. *Nat Commun* **14**, 5094 (2023).
15. Nguyen, T. L., Vieira-Silva, S., Liston, A. & Raes, J. How informative is the mouse for human gut microbiota research? *Dis. Model. Mech.* **8**, 1-16 (2015).

16. Sorbara, M. T. *et al.* Inhibiting antibiotic-resistant *Enterobacteriaceae* by microbiota-mediated intracellular acidification. *J. Exp. Med.* **216**, 84-98 (2019).
17. Theriot, C. M. *et al.* Antibiotic-induced shifts in the mouse gut microbiome and metabolome increase susceptibility to *Clostridium difficile* infection. *Nat. Commun.* **5**, 3114 (2014).
18. Papadimitriou-Olivgeris, M. *et al.* Risk factors for enterococcal infection and colonization by vancomycin-resistant enterococci in critically ill patients. *Infection* **42**, 1013-22 (2014).
19. Stiefel, U. *et al.* Suppression of gastric acid production by proton pump inhibitor treatment facilitates colonization of the large intestine by vancomycin-resistant *Enterococcus* spp. and *Klebsiella pneumoniae* in clindamycin-treated mice. *Antimicrob Agents Chemother* **50**, 3905-7 (2006).
20. Emwas, A. H. *et al.* NMR spectroscopy for metabolomics research. *Metabolites* **9**, 123 (2019).
21. Nagana Gowda, G. A. & Raftery, D. in *Metabolomics and Its Impact on Health and Diseases* (eds Ghini, V., Stringer, K. A. & Luchinat, C.) (Springer, 2022).
22. McDonald, J. A. *et al.* Evaluation of microbial community reproducibility, stability and composition in a human distal gut chemostat model. *J. Microbiol. Methods* **95**, 167-174 (2013).
23. Silverman, J. D., Durand, H. K., Bloom, R. J., Mukherjee, S. & David, L. A. Dynamic linear models guide design and analysis of microbiota studies within artificial human guts. *Microbiome* **6**, 202 (2018).
24. Aguirre, M. *et al.* Diet drives quick changes in the metabolic activity and composition of human gut microbiota in a validated *in vitro* gut model. *Res Microbiol* **167**, 114-25 (2016).
25. Gao, G. *et al.* Effects of valerate on intestinal barrier function in cultured Caco-2 epithelial cell monolayers. *Mol Biol Rep* **49**, 1817-1825 (2022).
26. Yamaguchi, M. *et al.* Increased intestinal ethanol following consumption of fructooligosaccharides in rats. *Biomed Rep* **9**, 427-432 (2018).
27. Barlow, J. T., Bogatyrev, S. R. & Ismagilov, R. F. A quantitative sequencing framework for absolute abundance measurements of mucosal and luminal microbial communities. *Nat Commun* **11**, 2590 (2020).
28. Stokes, J. M., Lopatkin, A. J., Lobritz, M. A. & Collins, J. J. Bacterial metabolism and antibiotic efficacy. *Cell Metab* **30**, 251-259 (2019).
29. Cabral, D. J. *et al.* Microbial metabolism modulates antibiotic susceptibility within the murine gut microbiome. *Cell Metab* **30**, 800-823 (2019).
30. Perry, E. K., Meirelles, L. A. & Newman, D. K. From the soil to the clinic: the impact of microbial secondary metabolites on antibiotic tolerance and resistance. *Nat Rev Microbiol* **20**, 129-142 (2022).
31. Yang, J. H. *et al.* Antibiotic-induced changes to the host metabolic environment inhibit drug efficacy and alter immune function. *Cell Host Microbe* **22**, 757-765 (2017).
32. Shimizu, K., Seiki, I., Goto, Y. & Murata, T. Measurement of the intestinal pH in mice under various conditions reveals alkalization induced by antibiotics. *Antibiotics (Basel)* **10**, 180 (2021).
33. Shelton, C. D. *et al.* *Salmonella enterica* serovar Typhimurium uses anaerobic respiration to overcome propionate-mediated colonization resistance. *Cell Rep* **38**, 110180 (2022).
34. Guzman Prieto, A. M. *et al.* Global emergence and dissemination of enterococci as nosocomial pathogens: attack of the clones? *Front Microbiol* **7**, 788 (2016).

35. Werner, G. *et al.* Vancomycin-resistant *vanB*-type *Enterococcus faecium* isolates expressing varying levels of vancomycin resistance and being highly prevalent among neonatal patients in a single ICU. *Antimicrob Resist Infect Control* **1**, 21 (2012).
36. Gold, H. S. Vancomycin-resistant enterococci: mechanisms and clinical observations. *Clin Infect Dis* **33**, 210-9 (2001).
37. Faron, M. L., Ledebroer, N. A. & Buchan, B. W. Resistance mechanisms, epidemiology, and approaches to screening for vancomycin-resistant *Enterococcus* in the health care setting. *J Clin Microbiol* **54**, 2436-47 (2016).
38. Hayden, M. K. Insights into the epidemiology and control of infection with vancomycin-resistant enterococci. *Clin Infect Dis* . **31**, 1058-65 (2000).
39. Chavers, L. S. *et al.* Vancomycin-resistant enterococci: 15 years and counting. *J Hosp Infect* **53**, 159-71 (2003).
40. Bonten, M. J. *et al.* Epidemiology of colonisation of patients and environment with vancomycin-resistant enterococci. *Lancet* **348**, 1615-9 (1996).
41. Sender, R., Fuchs, S. & Milo, R. Revised estimates for the number of human and bacteria cells in the body. *PLoS Biol* **14**, e1002533 (2016).
42. Taur, Y. *et al.* Intestinal domination and the risk of bacteremia in patients undergoing allogeneic hematopoietic stem cell transplantation. *Clin. Infect. Dis.* **55**, 905-914 (2012).

RESPONSE TO REVIEWER COMMENTS

We would like to again thank the reviewers for giving their time to review our manuscript. Please see our responses to the reviewer comments below.

REVIEWER #1 (REMARKS TO THE AUTHOR):

The authors have thoughtfully addressed the issues raised in my primary review.

Author response:

Thank you.

REVIEWER #2 (REMARKS TO THE AUTHOR):

Thank you for taking my comments into account and for thoughtfully incorporating them into the revised manuscript. I am satisfied with the changes made and your answers to my questions. I agree with the adjustments that were made and I have no further requests.

Author response:

Thank you.

REVIEWER #3 (REMARKS TO THE AUTHOR):

All the points I raised were answered sufficiently.

Author response:

Thank you.

REVIEWER #4 (REMARKS TO THE AUTHOR):

The authors have to a large degree given thorough answers to answer the raised concerns, which have improved and clarified the manuscript. However, a few aspects remain:

Reviewer 4 Comment 1:

- Regarding comment 1: Yes, NMR spectroscopy provides absolute quantification when using a reference signal (usually TSP or DSS or ERETIC signal). Please clarify in the manuscript which reference has been used.

Author response:

We believe the reviewer is referring to the comment numbered as "Reviewer 4 Comment 2" in the previous response to reviewer comments file.

The reference signal used was TSP, as specified in the previous versions of our manuscript. This can be found on lines 958-960:

"NMR spectra pre-processing was performed in Topspin (v3.2.6), with an exponential window function (line broadened by 0.3 Hz), zero-order auto phasing (command apk0), baseline correction (command abs), and referencing TSP (command sref)."

We also mentioned that TSP was included in the NMR buffers used in this study (see lines 936-939 and lines 945-947).

Reviewer 4 Comment 2:

- Regarding comment 4: I remain skeptical that the concentration used and shown is representative of a healthy colon. In fact, a previous study (Higher total faecal short-chain fatty acid concentrations correlate with increasing proportions of butyrate and decreasing proportions of branched-chain fatty acids across multiple human studies; PMID: 39295782) show that the concentrations used in this draft and the table in the rebuttal, are borderline of the «healthy» spectrum. As stated by the author, it is «not unreasonable to find these metabolite concentrations». That is true, but being “not unreasonable” does not mean such concentrations are representative of the average gut environment; they seem to be a rather extreme case. I recommend the authors to discuss clearly this limitation. Lastly, I think the additional figure R6 should indeed be added to paper. It is an important control and reference necessary to properly interpret the data of Fig 4.

Author response:

In Figure 4 of this manuscript, we tested a mixture of propionate, butyrate, and valerate (PBV mixture) and a mixture of acetate, propionate, butyrate, and valerate (APBV mixture) at their high concentrations found in healthy human faeces, not their average concentrations found in healthy human faeces.

The metabolite concentrations tested in this manuscript were based on concentrations measured from the faeces of healthy human donors in our previous study¹. The metabolite concentrations measured in healthy participants (not consuming a dietary intervention) by LaBouyer *et al.* (2022) are comparable to the concentrations measured in our previous study:

Table R1: Average and high concentrations of short chain fatty acids measured in the faeces of healthy human donors that did not receive a dietary intervention.

Metabolite	Average concentration (mM)		High concentration (mM)	
	From our previous study ¹	From LaBouyer et al. (2022)	From our previous study ¹	From LaBouyer et al. (2022)
Acetate	64.1	54.3	122.7	122.4
Propionate	16.1	17.3	34.7	45.0
Butyrate	16.4	12.9	38.8	40.9
Valerate	3.7	2.6	12.0	6.6

Some differences in the concentrations of short chain fatty acids measured in these studies are expected, as these donors consumed their own uncontrolled baseline diet, may have had differences in the time that they last consumed antibiotics (minimum of 3 vs 6 months prior to the start of the study), and may have had differences in the consumption of probiotics, prebiotics, or medications. Moreover, these studies contained different numbers of donors and used different metabolite profiling techniques to quantify the concentrations of the metabolites.

Moreover, as mentioned in our previous response to reviewer comment file, in our previous study we found that the highest concentrations of acetate, propionate, and butyrate were all found in the same donor faeces, while the highest valerate concentration was found in the donor faeces with the second highest concentration of propionate and butyrate and the third highest concentration of acetate. As mentioned previously, this data demonstrates that it is not unreasonable to find these high metabolite concentrations for propionate, butyrate, and valerate within the same healthy gut microbiome. We were not implying that these concentrations were the average concentrations found in healthy human faeces.

We have not included Figure R6 as part of the manuscript main text files or supplementary files, but Figure R6 will be published as part of the response to reviewers file along with the manuscript. As stated in our previous response to reviewer comment file, no such “control” bacteria exist for the gut microbiome. We were reluctant to include Figure R6 as part of the manuscript main text files or supplementary files as we did not want to look like we were introducing bias by selecting these specific gut commensal strains to test in this assay. However, we ended up including the data in the rebuttal to demonstrate feasibility that some gut commensal strains can grow in the presence of the propionate, butyrate, and valerate (PBV) mixture. In addition, Figure 4 clearly demonstrates that the PBV mixture provides full or near full inhibition of VRE growth in the absence of the data presented in Figure R6.

Reviewer 4 Comment 3:

- Regarding comment 10: Please explain what “the subtraction method” is? It remains unclear how results in Fig 5 and 8 are normalized.

Author response:

As stated in the previous version of the manuscript, we used the subtraction method to normalise the growth curve data from the carbon and nitrogen assays. We have clarified how the subtraction method works on lines 1120-1125:

“The carbon source utilisation growth curves and mixed nutrient growth curves were normalised to the no carbon control using the subtraction method, where AMiGA subtracted the growth parameters of the no carbon control from the carbon test samples. The nitrogen utilisation growth curves were also normalised to the no nitrogen control using the subtraction method, where AMiGA subtracted the growth parameters of the no nitrogen control from the nitrogen test samples.”

Reviewer 4 Comment 4:

- Regarding comment 12: It is well explained in the rebuttal. Please, state more explicitly in the results and methods, shortly, how you were able to select *E. faecium* combining vancomycin-resistant and Brilliance VRE plates.

Author response:

We have added the following sentence to the methods section at the first mention of Brilliance VRE plates (see lines 882-885):

“Brilliance VRE agar is a selective chromogenic agar plate that can differentiate vancomycin-resistant *E. faecium* and *E. faecalis* based on the colour of their colonies (vancomycin-resistant *E. faecium* grow as purple colonies, while vancomycin-resistant *E. faecalis* grow as light blue colonies).”

Reviewer 4 Comment 5:

- Regarding comment 24: Please describe exactly what was quantified by plating on Columbia agar? This remains unclear.

Author response:

This experiment quantified *E. faecium* and *E. faecalis* growth in gut growth medium broth supplemented with different antibiotics. As stated previously, VRE growth was quantified by plating the culture on Columbia agar plates. We have slightly edited this sentence to make this clearer (see lines 834-835):

“VRE growth was quantified in VRE cultures by plating samples on Columbia agar plates and incubating the plates at 37°C for 24 hours.”

This technique allows us to quantify VRE growth as plate counts (CFU/ml), as shown in Figure S6.

REVIEWER #5 (REMARKS TO THE AUTHOR):

Author response:

Thank you.

REFERENCES:

1. Yip, A. Y. *et al.* Antibiotics promote intestinal growth of carbapenem-resistant *Enterobacteriaceae* by enriching nutrients and depleting microbial metabolites. *Nat Commun* **14**, 5094 (2023).